# Securing The Model Context Protocol (MCP): A Dual-Axis Survey with a Mitigation-Oriented Threat Taxonomy

## Abstract

Agentic AI systems are increasingly adopting the Model Context Protocol (MCP) to integrate external tools and data sources. However, MCP's extensibility and multi-agent design introduce a broader and more complex attack surface. This exposes these systems to a diverse set of security risks. Prior research provides only a partial view of these risks, focusing on either temporal aspects (MCP system lifecycle) or spatial aspects (MCP components) in isolation, without capturing how the associated threats emerge, evolve, and propagate across agentic workflows. This missing linkage obscures causal chains across components and lifecycle phases, hindering timely root cause localization and effective prevention of cross-phase threat cascades. To address this gap, we present a spatio-temporal aligned taxonomy that maps each threat to both its affected MCP component and the lifecycle phase in which it arises. This dual-axis view highlights where threats occur, when they emerge, and how they propagate across stages of deployment and operation. Grounded in this taxonomy, our survey catalogs over 50 MCP-specific threats and classifies each by lifecycle phase and affected component, providing, to our knowledge, one of the broadest phase-by-component mappings of MCP security risks to date. Each threat is also cross-referenced to STRIDE (Spoofing, Tampering, Repudiation, Information Disclosure, Denial of Service, and Elevation of Privilege) and MAESTRO (Modeling Adversarial Events for Systematic Threat Representation and Organization) to align with widely adopted threat-modeling frameworks for agentic AI, covering core security properties (via STRIDE) and layers specific to agentic AI (via MAESTRO). We then pair each taxonomy category with actionable controls, explicit verification checks, and runtime signals, supporting structured control prioritization, verification checks, and runtime monitoring across design-time and runtime. ~~We further validate this control-oriented view through a compact benchmark~~ We further illustrate the feasibility of this control-oriented view through a compact proof-of-concept benchmark of verifiable checks in a real MCP deployment, showing how representative failures can be detected, localized, and tied to concrete audit artifacts. ~~Together, the taxonomy and its control mappings provide~~ Together, the taxonomy and its control mappings offer a structured basis for an evidence-informed framework for organizing MCP security priorities, verification checks, and reviewable security evidence across the MCP lifecycle.

The benchmark implementation, scenario definitions, verification scripts, and reproduction instructions are available at `https://anonymous.4open.science/r/mcp-security-benchmark-22C9/`.

## 1 Introduction

The Model Context Protocol (MCP) is an open standard introduced by Anthropic to enable large language models (LLMs) to interact with external tools, data sources, and computational services in a standardized and extensible manner Model Context Protocol (2025p). It defines a client–host–server architecture and uses JSON-RPC 2.0 (JavaScript Object Notation-Remote Procedure Call) for all messages, exposing a unified interface to invoke remote tools, access files, query databases, and call application programming interfaces (APIs) Li et al. (2025b); Kumar et al. (2025). This JSON-RPC–based interoperability is what

makes MCP-based agentic systems flexible and broadly applicable, enabling complex tasks with real-time access to external resources. This flexibility arises because MCP encodes tool invocations and data access as JSON-RPC 2.0 messages over a unified logical channel between client and server that supports capability discovery and structured argument passing, thereby decoupling the host and LLM runtime from service-specific APIs and integrations. However, because this interoperability relies on cross-boundary calls and shared state, it also creates a broader, more intricate attack surface, elevating exposure to threats including prompt injection, tool poisoning, credential leakage, colluding agents and supply-chain compromise Ferrag et al. (2025); Narajala & Habler (2025); Hasan et al. (2025). Concretely, the shape of that threat surface follows the MCP architecture: the client-side LLM host, the JSON-RPC communication layer, tool-provider servers, and mechanisms for continuous updates and tool registration. Each component opens a distinct pathway for misuse or failure, and thus a potential vector for exploitation. Compounding these component-level vectors, MCP deployments are often dynamic and multi-agent, combining human-in-the-loop and autonomous interactions across decentralized infrastructures. As a result, the likelihood and impact of cross-component threat propagation across deployment stages increase Fang et al. (2025); Hasan et al. (2025). Motivated by these threats, a growing body of MCP-specific security research has emerged, spanning surveys, empirical audits, attack demonstrations, and defense frameworks Yang et al. (2025); Song et al. (2025); Narajala & Habler (2025); Kumar et al. (2025); Brett (2025); Hou et al. (2025); Radosevich & Halloran (2025); Guo et al. (2025b); Hasan et al. (2025); Li & Gao (2025); Zhang et al. (2025). Benchmarking efforts have further strengthened this line of research by evaluating agent safety across real-world MCP servers and diverse attack types, and by introducing a large-scale benchmark for tool-use competency in realistic multi-server MCP workflows Zong et al. (2025); Bandi et al. (2026). Beyond benchmarking, the literature has also expanded at both the protocol and tool levels. Maloyan & Namiot (2026) analyze architectural weaknesses in MCP itself, while Li et al. (2026a) show how implicit tool poisoning in metadata can redirect agents toward high-privilege tools even without invoking the poisoned tool. Taken together, this growing body of work has expanded the evaluation and understanding of MCP-specific security failures and realistic MCP agent behavior, yet most studies still examine only a single axis, either lifecycle stages or architectural components, leaving a gap for a unified spatio-temporal taxonomy that links when threats arise to where they propagate across MCP workflows.

Hou et al. (2025) proposed one of the earliest structured approaches to MCP security by categorizing threats based on the distinct lifecycle phases of an MCP deployment, namely the creation, operation, and update stages. Their lifecycle threat mapping and mitigation analysis focuses on mapping different types of vulnerabilities to these temporal categories. For example, they treat name collisions during server registration, installer spoofing, and backdoor attacks from code-integrity failures as creation-phase threats. In operation, they highlight tool-name conflicts, slash-command overlap, and sandbox escapes. In the update phase, they discuss privilege persistence after authorization changes, redeploying vulnerable versions, and configuration drift. While this lifecycle taxonomy provides a useful temporal abstraction, it lacks the granularity needed to associate threats with specific system components, which limits its utility in tracing the structural impact of a vulnerability. In contrast, Narajala & Habler (2025) propose an architectural perspective on MCP security, organizing threats around the core components of an MCP system: the Host, the Client, and the Server. Within the server, they further analyze sub-modules, including tools, prompts, and resources. Their model maps security threats to these architectural elements, covering issues such as tool poisoning, prompt injection, insecure communication, and hosting environment vulnerabilities, and pairs them with mitigation strategies, including scoped access controls, communication hardening, and operational safeguards. Complementing this architecture-centric view, Li & Gao (2025) present the first ecosystem-wide security analysis of real MCP deployments, decomposing the ecosystem into hosts, registries, and servers and empirically demonstrating how weak vetting and missing output verification pose a threat to MCP security. These component-driven approaches enable more localized threat identification by highlighting threats associated with specific elements, such as man-in-the-middle attacks on the JSON-RPC interface, insecure tool execution environments, or rogue tool provider servers. However, their analysis treats these components as static units, without linking them to lifecycle phases. As a result, it becomes challenging to contextualize when a given component-related threat might emerge or propagate during the lifecycle of a system. While all of the aforementioned works contribute significantly to structuring early discussions of MCP security, they also exhibit a shared limitation. They treat lifecycle stages and architectural components as separate silos,

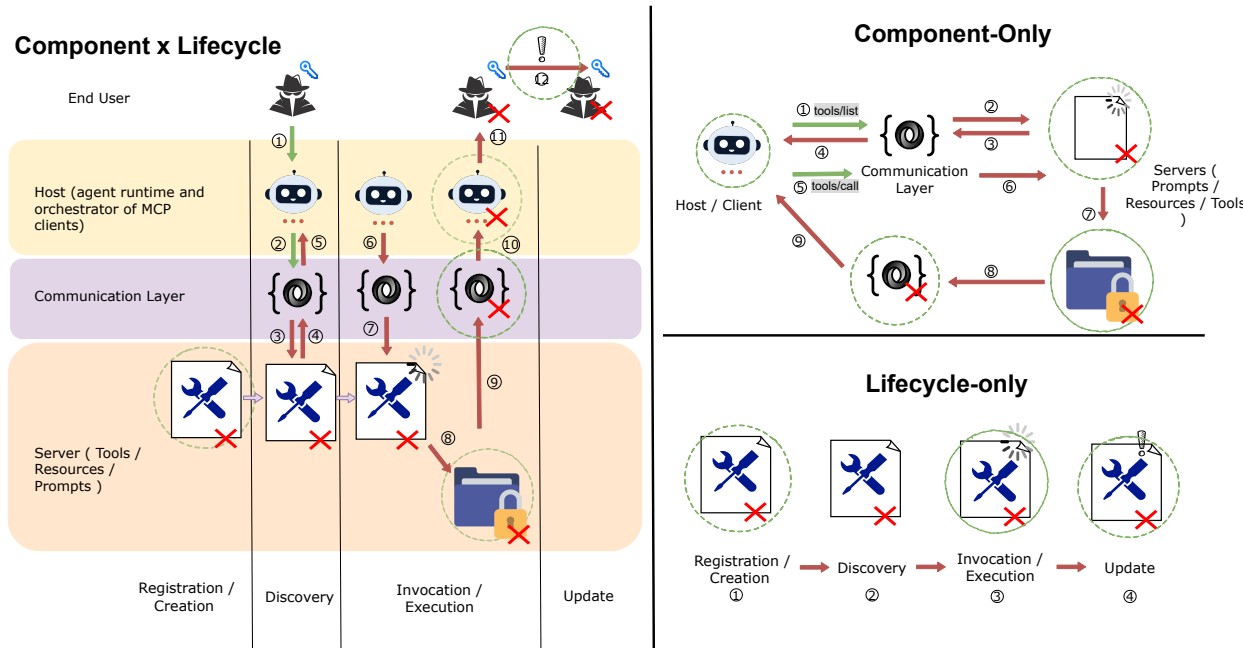

Figure 1: **Creation-to-runtime threat cascade in MCP**, showing how an over-permissive file tool configured during creation is exploited during operation to leak credentials and trigger follow-on tool and server actions, motivating a dual-axis mapping by component and lifecycle. Red arrows and crosses mark the attack path and violations, green arrows show expected interactions, and green dashed circles mark control points.

which produces a fragmented view of the threat landscape. Thus, it becomes challenging to capture how a vulnerability introduced during one phase of the lifecycle, such as creating a tool with malicious payloads, might later interact with various components like the LLM host, communication channel, or tool registry to cause harm during execution. This disjointed framing also limits the ability to anticipate compound threats that evolve across phases, such as those involving delayed activation, staged escalation, or multi-component compromise.

More recently, Jing et al. (2025) introduced the Model Contextual Integrity Protocol (MCIP), a multidimensional scheme that maps runtime threats to axes such as phase (Config, Interaction, Termination), source, scope, violation type, and MAESTRO layer. MCIP offers valuable insights into runtime information-flow violations, particularly in capturing how contextual integrity can be compromised during tool invocation and result integration. While the taxonomy includes setup and termination lifecycle phases, the empirical focus leans toward interaction-phase behaviors and flow-based violations: the experiments target interaction threats, and the paper does not detail sandbox-escape mechanics or comprehensive software-supply-chain hardening beyond the taxonomy. In parallel with lifecycle and component analyses, Guo et al. (2025b) emphasize an attack-type taxonomy and empirical evaluation with minimal architectural grounding. They do not map threats by lifecycle stage, and component coverage is partial. Instead, they develop an attack-centric taxonomy and release an MCP attack library (MCPLib), which catalogs MCP-specific attacks in four classes, namely direct tool injection, indirect tool injection, malicious-user attacks, and LLM-inherent attacks. Taken together, these strands advance the field but leave open how to connect attack behaviors to both specific components and specific points in the lifecycle. In practice, however, the most damaging issues often appear as concrete, cross-cutting failures in deployed MCP systems rather than neatly categorized attack types. Now, against this backdrop, recurring failure modes continue to surface in practice: agents over-trust tool descriptions, file-based attacks succeed disproportionately, credentials leak across tool boundaries, and local misconfigurations cascade across components at runtime Xing et al. (2025); Guo et al. (2025b); Radosevich & Halloran (2025). For example, large-scale audits of open-source MCP servers report MCP-specific tool

Table 1: Comparison with the closest MCP security surveys and frameworks. Symbols: ● = yes, ◐ = partial, ○ = no. "Dual-axis" denotes explicit joint localization by MCP architectural component(s) and lifecycle phase(s).

| Work | Life. | Comp. | Dual | Fwk. | Ctrl. | Check | Emp. |
|---|---|---|---|---|---|---|---|
| Hou et al. (2025) | ● | ◐ | ○ | ○ | ◐ | ○ | ◐ |
| Narajala & Habler (2025) | ○ | ● | ○ | ● | ◐ | ○ | ○ |
| Li & Gao (2025) | ○ | ● | ○ | ○ | ◐ | ◐ | ● |
| Jing et al. (2025) | ◐ | ◐ | ◐ | ● | ◐ | ◐ | ● |
| Guo et al. (2025b) | ○ | ◐ | ○ | ○ | ○ | ○ | ● |
| **Ours** | ● | ● | ● | ● | ● | ● | ● |

*Abbreviations.* Life. = lifecycle axis; Comp. = component axis; Dual = joint component×lifecycle mapping; Fwk. = STRIDE/-MAESTRO alignment; Ctrl. = controls mapped to where and when; Check = verifiable checks/evidence; Emp. = empirical study or benchmark. MCIP maps phase/source/scope/type/MAESTRO, but not MCP component×lifecycle. MCPInspect is counted as partial checking support because it provides pre-integration scanning rather than taxonomy-wide evidence hooks.

Figure 2: **Threat distribution across components and lifecycle stages.** Each cell reports the count of distinct risks from our threat corpus mapped to the corresponding component–stage pair. Risks tagged with multiple components or stages contribute to each applicable cell.

poisoning and widespread misconfigurations that persist in real deployments Hasan et al. (2025). These security failure patterns span both where attacks land (host, transport, server, and tool/resource/prompt sub-modules) and when they materialize (creation, operation, update). Table 1 contrasts our work with the closest MCP security surveys and frameworks, and the key distinction is that ours couples *when* a threat emerges, *where* it propagates, and *how* the corresponding control can be verified, rather than addressing these in isolation.

Without a framework that jointly captures components and lifecycle phases, developers, researchers, and security practitioners struggle to localize root causes, anticipate cross-phase propagation, and select enforceable controls at design time and runtime. For instance, the creation-to-runtime cascade in Figure 1 is hard to localize and contain without a view that spans both components and lifecycle phases. For clarity, the

Host and Client are combined, and the Pre-MCP, Initialization/Connection, and Shutdown phases are omitted, focusing on Creation, Discovery, Invocation/Execution, and Update. Now, a lifecycle-only perspective shows when issues occur (Creation → Invocation/Execution → Update) and supports coarse, phase-scoped checks (Creation: scope review; Execution: output redaction; Update: token rotation), but it cannot localize fixes to specific components or produce component-scoped evidence. A component-only perspective shows where the chain runs (Server: Tools → Transport → Host) and supports control placement (tool-scope lint at Server: Tools; mTLS/log redaction at Transport; revocation at Host), but it does not specify when each control must apply; there is no Creation gate (CI before deploy), no Invocation/Execution-time channel policy, and no Update-phase revocation checklist, so enforcement is non-temporal and audits are non-deterministic. In contrast, the dual-axis view provides both when and where, localizing controls to Creation × Server: Tools (directory allow-lists and scope lint), Invocation/Execution × Transport (JSON-RPC) (mTLS and log redaction on the channel), and Update × Host (token-revocation checks), making the component chain explicit, assigning ownership (tool-server, platform/transport, host teams), and enabling verifiable evidence per component (configuration invariants, transport policy assertions, revocation audits).

Guided by this need for a dual-axis map, we pose four research questions: (i) *What MCP-specific threats arise across core components and lifecycle phases?* (ii) *How do these threats emerge, evolve, and propagate, including the causal patterns that drive cross-phase cascades?* (iii) *Which design-time and runtime controls most effectively disrupt these paths, and how can their enforcement be verified?* and (iv) *How can MCP-specific threats be aligned with established threat-modeling frameworks to support consistent modeling and auditing?* In examining these questions, we find that existing approaches usually address only a restricted segment of one axis (components or lifecycle phases), resulting in unaddressed threats not only on the alternate axis but also within the unexamined segments of the chosen axis. To address these limitations, this paper introduces a finer-grained taxonomy that assigns when and where each threat emerges. Our approach provides a spatio-temporal view of MCP deployments by systematically integrating architectural (spatial) and temporal dimensions, enabling precise attribution of when and where each threat emerges. Unlike prior work that isolates phases or focuses on specific classes of threats, our taxonomy covers major stages of the MCP lifecycle, from pre-deployment setup and tool installation to runtime execution and system shutdown, yielding a structured mapping of threats to both components and lifecycle stages. We visualize this coverage in Fig. 2, which lists components by rows and lifecycle stages by columns and shows the count of distinct risks in each cell. We omit Resources from the component axis in this figure because it rarely appears as the primary affected component or blast radius in our corpus. When resources are implicated, the effect is captured under the Data layer or the executing tool or server.

Building upon this foundation, our work makes three principal contributions:

1. **Dual-axis threat mapping.** We build a deployment-wide taxonomy of MCP-specific threats by (i) partitioning the attack surface into nine failure modes and (ii) cross-classifying each threat along orthogonal coordinates: failure mode, MCP components, lifecycle phases, and established threat-modeling frameworks. This dual-axis, multi-coordinate scheme localizes where and when each threat manifests and grounds it in widely adopted modeling frameworks; the full mapping is presented in Tables 4–6.

2. **Threat mitigation with verifiable enforcement.** Existing guidance is fragmented and seldom specifies which controls to apply where (component) and when (lifecycle phase), or how to verify that they are enforced. We address this by attaching enforceable controls to each threat category in our dual-axis taxonomy and by pinpointing the component and lifecycle phase at which each control should apply. For every category, we provide implementation patterns such as scoped capability bindings, privilege fences, provenance and integrity checks, and session or isolation guards, along with verification hooks in the form of machine-checkable evidence such as policy assertions, configuration invariants, and runtime signals. We also define pass or fail checks that can be integrated into CI or CD gates and runtime monitors. ~~We further validate this control-oriented view through a compact benchmark of verifiable checks in a real MCP deployment, showing how representative failures can be detected, localized, and tied to concrete audit artifacts.~~ We provide a compact proof-of-concept benchmark showing that selected taxonomy-derived checks can be implemented in a real MCP deployment, produce expected enforcement outcomes, and generate reviewable audit

artifacts for curated representative failures. ~~Together, these elements turn high-level guidance into verifiable controls, evidence-producing checks, and concrete audit artifacts rather than description alone.~~ Together, these elements provide an initial demonstration of how high-level guidance can be operationalized as verifiable controls, evidence-producing checks, and concrete audit artifacts.

3. **Expanded threat coverage with operational safeguards.** Prior work leaves multiple individual MCP threats unaddressed or only loosely implied, including lack of verified server identity/PKI, session-isolation failures, agent handoff chains, orchestration drift, agent collusion, and others. We add these threats to the MCP threat surface and locate each by component and lifecycle phase within our dual-axis scheme. Their prevention and detection can then be handled using the proposed threat-mitigation strategy mapping for the corresponding threat category, which supplies testable detection targets and deployable controls.

In addition to these core contributions, we examine recent advances in security frameworks for MCP, identify key limitations of existing approaches, and highlight opportunities for future work. This includes addressing the need for automated threat detection, robust context validation, and secure multi-agent orchestration mechanisms. By offering a broad and operationally grounded taxonomy together with mitigation strategies, our work aims to strengthen the security foundation of tool-augmented AI systems. The remainder of this paper is organized as follows: Section 2 provides background on the MCP architecture and lifecycle, laying the foundation for the subsequent threat modeling. Section 3 describes our dual-axis taxonomy and the methodology used to develop it. It also systematically catalogs the identified threats and classifies them within the taxonomy. Section 4 presents mitigation strategies and security controls for MCP based on the proposed taxonomy. ~~Section 5 provides a structured evaluation of verifiable checks in a compact MCP benchmark.~~ Section 5 illustrates selected verifiable checks through a compact proof-of-concept MCP benchmark. Section 6 reviews emerging frameworks and protocol extensions in MCP security research. Section 7 outlines future directions, and Section 8 concludes the paper.

## 2 MCP Architecture and Lifecycle

The structure of an MCP deployment is governed by two complementary dimensions that operate in parallel: architecture and lifecycle. The architecture specifies **where** each component resides and **who** owns it. This includes a host that reasons, clients that communicate, servers that execute and tools that act, data that they access, and transports that connect them. The lifecycle defines **when** each stage of operation occurs, including setup and connection, creation and discovery, invocation and execution, update, and shutdown. Tracking only the "where" gives a static view of the system, allowing attacks to evolve over time and bypass controls that are not active in the right phase. Tracking only the "when" captures timing but not responsibility, allowing faults to move across components with no clear owner. Considering both where and when dimensions enables complete traceability across the system. Building on this view, we formalize the architecture and lifecycle as orthogonal coordinates for the taxonomy presented in Section 3.

### 2.1 MCP Architecture

The Model Context Protocol (MCP) follows a layered *client–host–server* architecture rather than a flat client–server design Model Context Protocol (2025f). In this structure, the host coordinates clients, manages links to servers, and provides the setting required for model reasoning and planning Singh et al. (2025). Each **MCP host** runs one or more **MCP clients**, which are protocol-level components instantiated by the host to communicate with particular **MCP servers**, and each client maintains a dedicated one-to-one connection to a single MCP server Model Context Protocol (2025g). In effect, this separation lets the host add new capabilities by spinning up additional client instances, each paired with a different server. Thus, the same client implementation can then be reused across many servers, with changes usually limited to configuration rather than client code Model Context Protocol (2025h); Microsoft (2025). In practice, this means the system can scale horizontally: an AI agent can extend its abilities simply by connecting to additional servers, each providing its own set of tools or resources. For example, a host can run one client instance that connects to

a file management server and a second client instance that connects to a data analytics or scheduling server. The same client implementation is reused in both cases with configuration changes only.

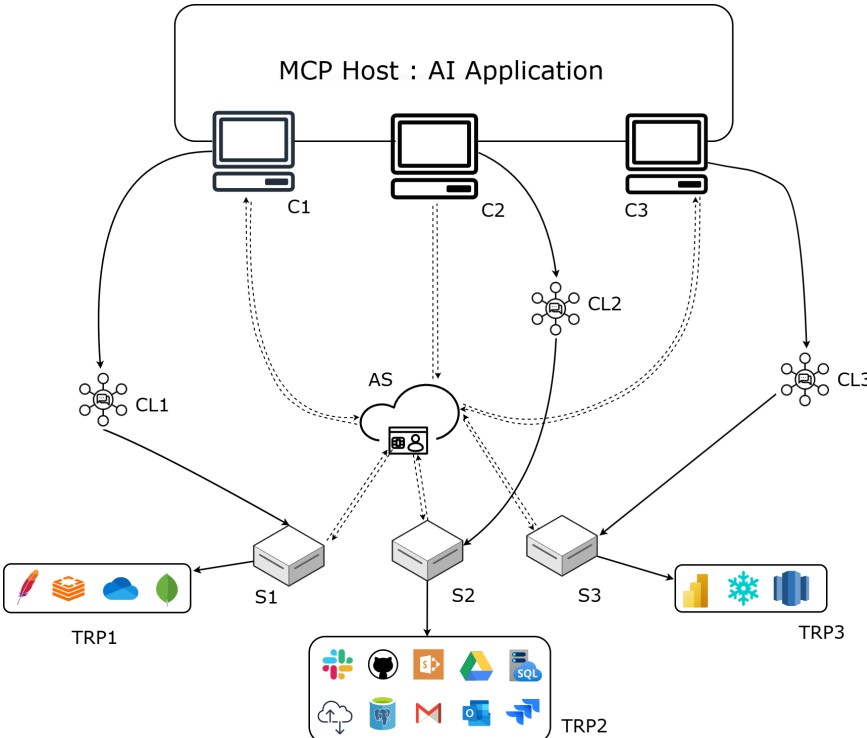

Figure 3: **MCP architecture** - The *Host* encloses three MCP clients (C1-C3). Each client sends required control/data traffic (solid arrows) through its in-process Communication Layer (CL1-CL3) to a dedicated MCP server (S1-S3), which advertises domain-specific *Tools / Resources / Prompts* blocks (TRP1-TRP3). Optional interactions (dashed arrows) include capability discovery and authentication via the external Authorization Server (AS).

In MCP the roles are cleanly separated, with the host handling reasoning and planning, the client managing communication sessions and enforcing protocol consistency, and the server executing tasks, as shown in Fig. 3. Fig. 4 illustrates how MCP components coordinate during a request lifecycle. Working together, these layers form a coordinated pipeline in which the host decides what should happen, the client determines how to communicate it, and the server carries it out. So AI systems using MCP can scale efficiently while orchestrating multiple external tools and data sources without losing control or transparency Model Context Protocol (2025f). Once the host forms a request, the client interprets the intent, establishes a session with the chosen server, negotiates available capabilities, and issues structured method calls. The server completes the operations and returns results, which the client formats and delivers back to the host.

Building on this pipeline, the client and server communicate using JSON-RPC 2.0, a lightweight remote procedure call standard encoded in JSON that provides a structured, predictable message format Morley (2010). Within MCP this communication is expressed as two layers that work together. The **data layer** defines the structure and meaning of JSON requests and responses, covers the flow of a session such as connection setup, capability discovery, and orderly shutdown, and names the core primitives that implementations rely on, including tools, resources, prompts, and notifications. The **transport layer** carries these JSON messages across different environments. Local deployments often use `stdio` for in-process communication, while remote deployments use the Streamable HTTP transport over HTTPS so TLS provides confidentiality and integrity ~~hitachi (2023)~~ Rescorla (2018); Model Context Protocol (2025r). In higher-assurance settings, mutual TLS can be required so both client and server authenticate before any data is exchanged ~~Cloudflare Learning Center (2024)~~ Rescorla (2018). This separation keeps wire details independent from application semantics and allows one client to interoperate with many servers without changes to the protocol.

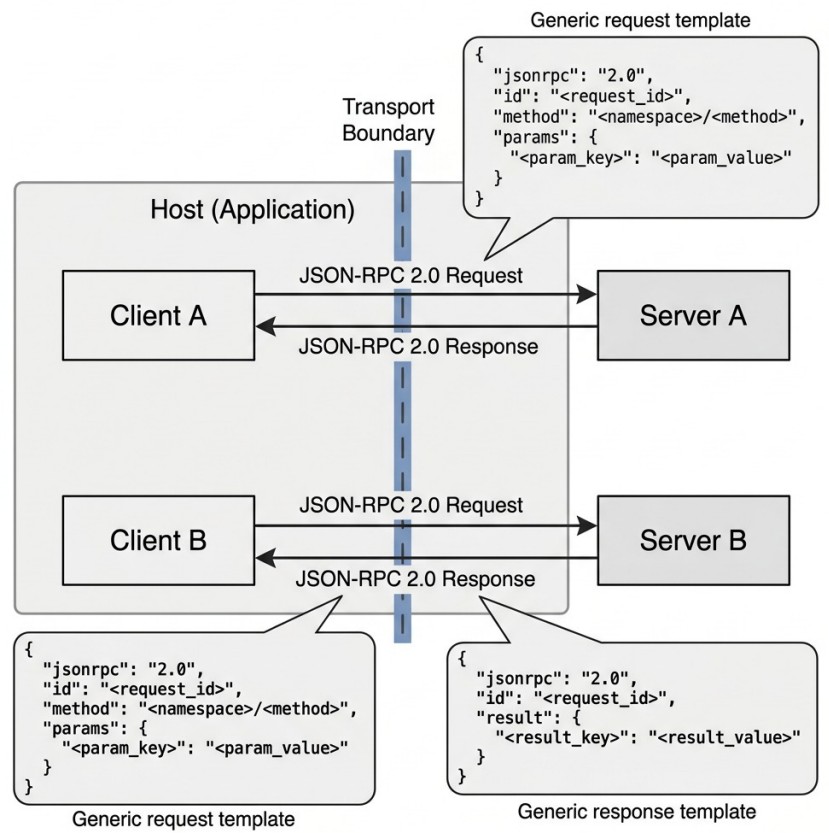

Figure 4: All communication between each MCP client and its paired server occurs through JSON-RPC 2.0 across the transport boundary. The callouts illustrate generic multi-line request and response templates with placeholders, where the response reuses the request id and returns the result object.

Within this communication model, the MCP server exposes callable tools, accessible resources, and reusable prompts, which together define its execution surface. **Tools** represent executable operations such as database queries, file actions, or REST calls that the client can invoke MCP Working Group (2025a); Fei et al. (2025). **Resources** act as structured data endpoints, including files, databases, or live streams, that can be read or updated depending on authorization scope. **Prompts** function as reusable templates that standardize interactions between language models and external systems, encoding reasoning patterns or output structures to maintain consistency across tasks. For instance, a data management server may provide a `process_dataset` tool for batch normalization, a `/data/reports.json` resource containing structured output, and a `summary_template` prompt for generating concise textual reports. By exposing tools, resources, and prompts through a unified JSON-RPC interface Morley (2010), MCP enables clients to discover, negotiate, and invoke capabilities. It also supports isolation, auditability, and policy enforcement when implementations follow the specification's security guidance Radosevich & Halloran (2025).

Access to these capabilities is governed by OAuth-based authorization. Once a secure HTTPS channel is established, MCP proceeds with authorization at the application layer to manage access to tools and capabilities. An **authorization server** issues access tokens, and the MCP server enforces access to tools and resources using those tokens Model Context Protocol (2025l). This arrangement defines how clients obtain scoped authorization and how they maintain valid tokens throughout a session. In protected deployments, authorization servers implement OAuth 2.1, MCP servers expose OAuth protected resource metadata and support discovery, and clients use that metadata to locate the authorization server Model Context Protocol (2025l); Lodderstedt et al. (2024). ~~When a client connects to a protected MCP server, it initiates an authorization flow to obtain an access token for the server's resource Posta (2025a).~~ When a client connects

to a protected MCP server, it follows the MCP authorization flow to obtain an access token for the target resource server Model Context Protocol (2025l).

The token conveys the intended audience for the resource, the authorized scope, and an expiry time, and it must accompany each request in the `Authorization:Bearer <token>` header Model Context Protocol (2025l). The server validates the token's issuer, audience, scope, and lifetime before executing any operation, ensuring that only authorized entities invoke sensitive tools or retrieve protected data. For example, a language-model-based research assistant might request read-only access to a document retrieval server that hosts academic literature. Through OAuth 2.1, it receives a time-limited token with the narrow scope needed for search operations but not modification. Similarly, an enterprise analytics client querying a payroll server could be issued a token scoped to anonymized statistical aggregates, maintaining compliance with corporate privacy and auditing policies.

Overall, the MCP architecture cleanly separates reasoning, communication, and execution. This modularity lets agents orchestrate multiple tools and data sources across distributed environments while supporting strong security when transports, authentication, authorization, and logging are correctly configured. Once connected, a client can enumerate available capabilities, plan composite workflows, and execute coordinated tasks across heterogeneous servers. Such architectures mirror emerging agentic systems that integrate multiple models, APIs, and data endpoints to perform complex reasoning and multi-step task execution Shen et al. (2024).

## 2.2 MCP Lifecycle

The operational lifecycle of the Model Context Protocol (MCP) defines how an AI agent and its connected infrastructure progress from preparation to decommissioning. Studying this lifecycle is security-critical because each phase exposes distinct attack surfaces, and controls are typically phase tied across secure setup, careful discovery, least-privilege invocation with auditing, ongoing maintenance, and clean shutdown Ross et al. (2022); Tabassi (2023a); Souppaya et al. (2022a). Accordingly, this paper adopts an end-to-end, deployment-oriented lifecycle for MCP systems (*Pre-MCP → Creation & Registration → Connection & Initialization → Discovery → Invocation & Execution → Update & Maintenance → Shutdown & Deregistration*) that integrates and extends two established references Model Context Protocol (2025i); Hou et al. (2025). It refines the official MCP connection lifecycle by emphasizing an Initialization → Operation → Shutdown arc that covers capability negotiation, routine JSON-RPC calls (e.g., `tools/list`, `resources/list`, `prompts/list`, `tools/call`), and graceful termination at both client and server connections Model Context Protocol (2025i). It also extends the server-centric lifecycle of Hou et al. (2025), which organizes MCP server operations into creation, deployment, operation, and maintenance with 16 activities, by adding pre-protocol preparation, explicit shutdown and deregistration hygiene, and a clear separation of discovery from invocation to support least-privilege analysis and auditing. Figure 5 presents the aforementioned lifecycle stages alongside a generic request flow in MCP. Taken together, this new framing lets phase-bound controls map directly to the corresponding Remote Procedure Calls (RPCs), configurations, and state transitions, setting up the stage-by-stage narrative that follows as we trace MCP from initialization to termination.

The lifecycle begins with the **Pre-MCP Phase**, which covers the preparatory work performed before any protocol-based connection is established. This includes model alignment, dataset preparation, and tool-use familiarization Ouyang et al. (2022); Gebru et al. (2021); Tabassi (2023b). During this stage, developers train and fine-tune models to understand structured tool invocations, define access policies, and register the endpoints or APIs that the MCP server will later expose. For example, in a financial domain, developers may prepare a financial assistant by aligning it with transaction schemas, regulatory terminology, and dataset formats it will later encounter through MCP tools. This preparation does not involve live tool interaction but ensures the model understands how to interpret data such as account summaries, balance sheets, and invoices. In the same phase, schema definitions for tools are created so the MCP server can later expose them in a structured, interpretable form Model Context Protocol (2025q); OpenAI (2025). This alignment allows the agent, once connected through MCP, to interact with these tools effectively without additional retraining Hou et al. (2025). Once the groundwork is complete, the **Creation and Registration Phase** initializes the server-side ecosystem. MCP servers are instantiated with metadata, including name, version, and supported schema definitions Model Context Protocol (2025c). This phase also includes registering tool

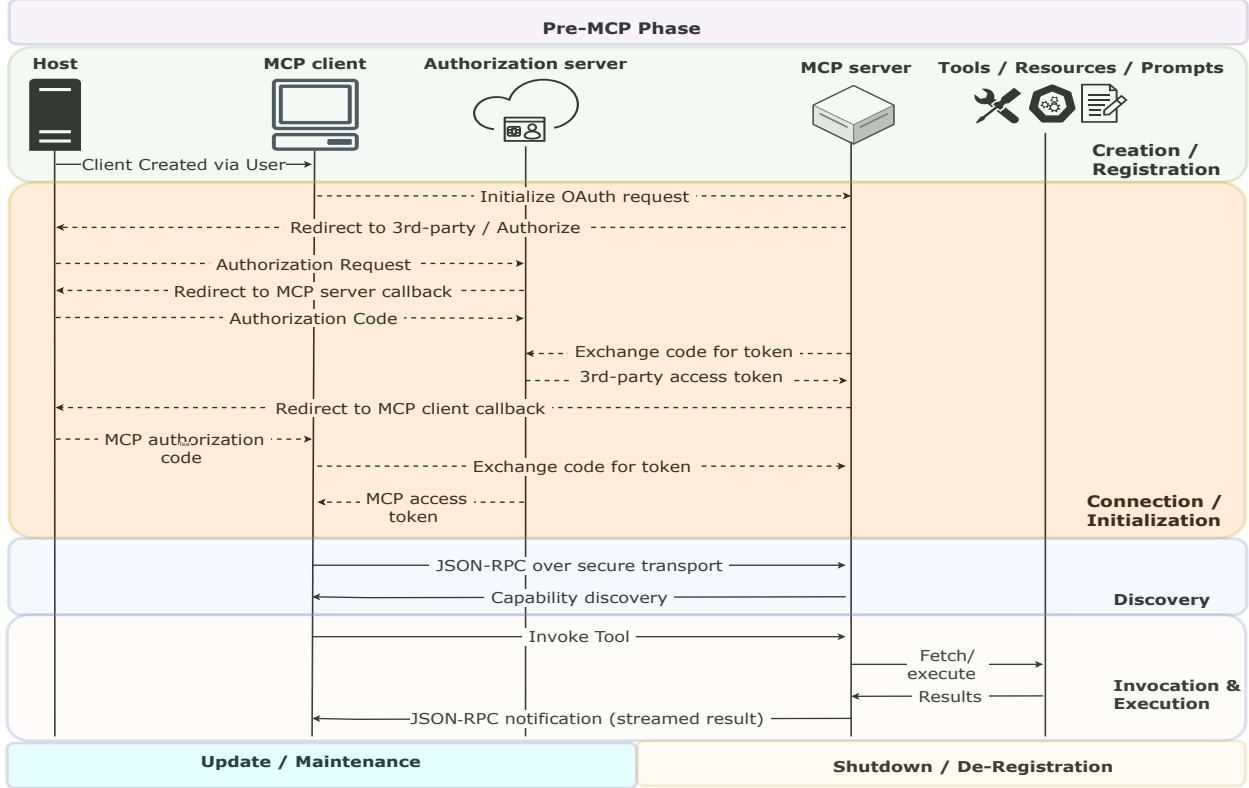

Figure 5: **MCP Lifecycle** - A user action in the Host creates an MCP client. If the server requires authorization, the client performs the OAuth 2.1 flow; in the third-party variant, the MCP server obtains a third-party token and then issues its own access token to the client. With a secure JSON-RPC channel established, the client performs capability discovery (`tools/list`, `resources/list`, `prompts/list`) and invokes the desired capability. The MCP server then executes the request against its Tool/Resource/Prompt back-end and streams results back to the client via JSON-RPC notifications. *Dashed arrows indicate optional steps in the sequence.*

manifests, resource URIs, and prompt templates that define the server's callable GitHub, Inc. (2025); Model Context Protocol (2025e;d). Secure trust anchors, such as certificate pinning or public key registration, are configured so that future clients can verify the server's authenticity Ahmadi et al. (2025). For instance, a deployment might register a "CI/CD tools" server exposing commands like `build`, `deploy`, and `rollback`, enabling reproducible software automation.

Following registration, the **Connection and Initialization** phase begins when the MCP client sends an `initialize` request and the client and server negotiate the protocol version, capabilities, and implementation metadata Model Context Protocol (2025b). Over HTTP transports, the MCP authorization specification requires authorization endpoints to be served over HTTPS, which provides channel encryption and server authentication for those endpoints Model Context Protocol (2025l). Where deployment requirements demand stronger client identity binding, operators can add mutual TLS or equivalent client-authentication controls. Once initialization completes, the client discovers available tools and resources through the protocol's listing methods, establishing the operational context for later invocations Model Context Protocol (2025q;e). For example, a research assistant might initialize a connection to a document retrieval server, validating its certificate and synchronizing available datasets before use. After establishing a secure channel, the **Discovery Phase** enables dynamic capability enumeration. The client queries the connected server for its registered `tools/list`, `resources/list`, and `prompts/list` endpoints. The server responds with structured metadata describing each callable entity like its name, input format, output type, and required permissions Model Context Protocol (2025q). This phase allows AI agents to self-adjust their reasoning and

available actions depending on what tools or datasets are currently registered. For instance, a travel planning agent could discover new `currency_exchange` or `visa_tracking` tools that extend its existing workflow.

Once discovery is complete, the **Invocation and Execution Phase** governs active interaction between the client and the server Model Context Protocol (2025b). Here, the agent issues structured JSON-RPC calls such as `tools/call` or `resources/read` to invoke specific capabilities Model Context Protocol (2025e;q). The server executes operations like querying databases, processing files, or composing responses and returns structured outputs back to the client. The host integrates these results into its reasoning loop, enabling chained or conditional actions Ahmadi et al. (2025). A data analysis agent, for example, might sequentially invoke `fetch_data`, `normalize_records`, and `summarize_report` tools in one continuous workflow. Over time, the **Update and Maintenance Phase** ensures that deployed servers remain stable and secure. Administrators may update tool manifests, rotate credentials, patch vulnerabilities, or retire deprecated endpoints Hasan et al. (2025); Hou et al. (2025). ~~Observability and audit logs are also reviewed to maintain compliance and performance Ithena (2025).~~ Practitioner compliance guidance also emphasizes reviewing MCP-specific audit trails, including agent identity, tool invocations, accessed data, and approval context, to support compliance and operational review Ithena (2025). For instance, if analytics tools receive requests in a new format, developers can register updated schemas and redeploy compatible endpoints without interrupting existing clients. Finally, the **Shutdown and Deregistration Phase** completes the lifecycle. This stage involves gracefully terminating active sessions, cleaning up allocated resources, and unregistering outdated tools or datasets. Tokens and session keys are invalidated to prevent further invocations Lodderstedt et al. (2013). In distributed deployments, containers or compute instances may be reclaimed to reduce cost and exposure ~~Raveendran (2024)~~ Kubernetes Documentation (2025). For example, a tax filing assistant might de-register fiscal tools at the end of a financial year to prevent unauthorized reuse.

In summary, the MCP lifecycle establishes a continuous operational loop that blends initialization, execution, and controlled termination. Each stage reinforces the others, ensuring reliability, version traceability, and least-privilege access control across the protocol's end-to-end operation.

## 3 Threat Taxonomy for MCP

MCP-based architectures involve tightly coupled interactions among underlying models (e.g., LLMs), agentic abstractions (e.g., agents and workflows), and communication protocols (e.g., tool invocation and orchestration) Model Context Protocol (2025a). Because this interplay expands the attack surface across components and lifecycle stages, we need a systematic way to reason about and mitigate the resulting risks. Figure 6 presents the threat taxonomy used in this work, summarizing the main categories of security risks in MCP deployments. However, knowing *where* and *when* a threat appears is not sufficient; we must also account for *what* the weakness fundamentally is, since the underlying failure mode is primarily used to generalize fixes, compare incidents, and prioritize controls. This structure is deliberate because a failure-mode-first taxonomy explains why incidents happen, whereas a dual-axis grid of component and lifecycle only describes where and when they surface. If we start from the dual axis alone, causally identical issues get split across multiple component–lifecycle coordinates and duplicated whenever they recur in a different place or phase, so the shared mechanism disappears. For example, consider an over-permissive file tool created during registration that is later exploited by prompt injection during execution, with secrets lingering after a missed revocation during update. Read purely through the dual axis, that single causal chain appears as three separate entries, one in Server Tools at Creation, one in Host at Execution, and one in Host at Update, with different owners and severities, which obscures that one authorization-scope flaw seeded the entire cascade. Grouping first on the root cause and only then annotating each threat with precise where and when coordinates (i) keeps causally similar events together, (ii) avoids combinatorial duplication, and (iii) attaches the right class of controls to the actual mechanism (e.g., scope linting and allow lists for mis-scoping, not just host logging at execution).

More generally, failure modes are stable, while components and phases are contingent on deployment and operational context such as where a tool runs, how it is configured, and when checks are applied. For example, a tool may move from local stdio to a remote server, or a check may shift from creation to connection; the grid coordinates change, yet the underlying vulnerability, such as an identity and trust

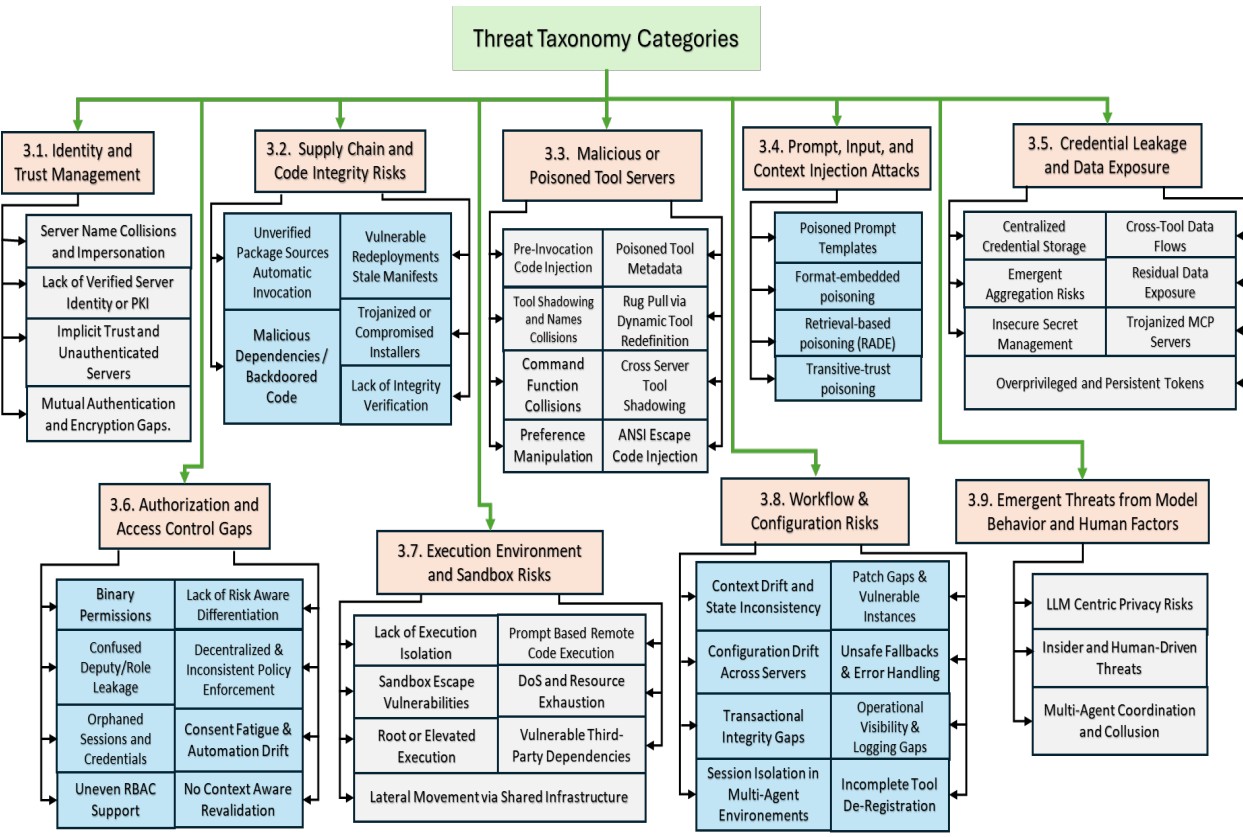

Figure 6: Threat taxonomy categories in MCP

failure from weak server verification, remains the same. Anchoring on failure modes provides invariants that can make evidence, mitigations, and future metric design more reusable across deployments. It also helps avoid double counting and can support later prioritization by keeping severity and frequency reasoning attached to stable failure modes rather than scattered across many cells. This framing matches how teams actually fix things: engineering ships control patterns to defeat mechanisms, for example PKI and mTLS for identity and trust, signing and provenance for supply chain, and sandboxing and quotas for execution, while SRE and security use the dual-axis tags to decide where to enforce them. In short, failure modes partition by cause so patterns, metrics, and mitigations stay coherent, and the dual axis localizes by context so fixes land precisely. The result is a taxonomy that travels well across deployments and remains surgical in practice: a structured threat taxonomy that links **where** a threat occurs to **when** it emerges in the system's lifecycle and, crucially, to **what** the weakness fundamentally is. This unified what–where–when view provides the holistic context needed to understand and mitigate MCP vulnerabilities, closing gaps left by approaches that track only location and phase.

Furthermore, to ground this framework in evidence, we anchor the taxonomy in field observations so its categories reflect how failures actually occur. We construct it by examining real-world incidents, empirical audits, and prior frameworks to identify recurring failure patterns in MCP, drawing on synthesis pieces and technical write-ups that survey the ecosystem. Our evidence base includes reports of 492 publicly exposed MCP endpoints lacking authentication or encryption Data Science Dojo Staff (2025); Lammerts (2025) . We also draw on ecosystem-scale assessments that quantify prevalent weaknesses, with about 43% of analyzed servers exhibiting command-injection or code-execution exposure, 33% permitting unrestricted URL fetches, 22% exposing files, and 66% showing poor secret-handling code smells Raina (2025). In parallel, independent coverage reports more than 15,000 MCP servers identified via public GitHub code search, which we treat as a prevalence indicator rather than an internet-reachability measurement Prompt Security Team (2025b). Field measurements further document hundreds of servers bound to 0.0.0.0 enabling neighborjacking and

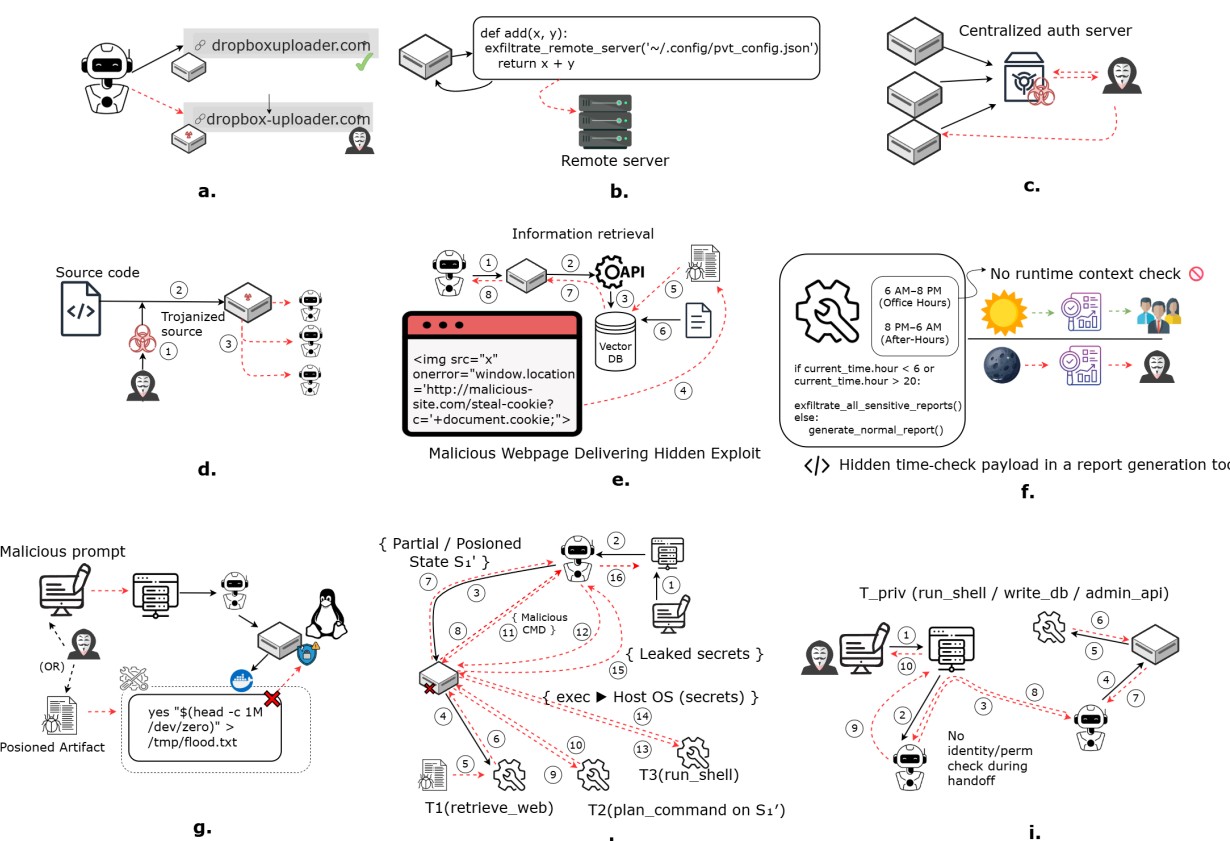

Figure 7: One exemplar threat from each of the nine MCP threat categories (§ 3.1–§ 3.9). (a) *Identity & Trust Management* - server-name collision/impersonation. (b) *Malicious or Poisoned Tool Servers* - pre-invocation code injection exfiltrating secrets. (c) *Credential Leakage & Data Exposure* - centralized auth/token store as a single point of compromise. (d) *Supply Chain & Code Integrity* - trojanized dependency/source. (e) *Prompt, Input & Context Injection* - indirect prompt injection via retrieved web content. (f) *Authorization & Access Control Gaps* - missing runtime/context-aware checks. (g) *Execution Environment & Sandbox Risks* - prompt-driven OS command/DoS from an unsandboxed tool. (h) *Workflow & Configuration Risks* - context drift across a multi-step pipeline. (i) *Multi-Agent & Orchestration Risks* - unverified agent handoff/privilege carry-over. Dashed red traces depict the attack flow; icon overlays indicate the involved MCP components.

~~widespread misconfiguration, with multiple sources characterizing nearly 500 unauthenticated exposures Naamnih & Ginzburg (2025); Data Science Dojo Staff (2025); Lammerts (2025); Nelson (2025).~~ Furthermore, to ground this framework in evidence, we anchor the taxonomy in empirical MCP studies and protocol-aware security analyses rather than relying on synthesis blogs or news coverage. A large-scale empirical study of 1,899 open-source MCP servers finds that 7.2% contain general vulnerabilities and 5.5% exhibit MCP-specific tool poisoning, showing that both conventional software flaws and MCP-specific weaknesses appear in public server implementations Hasan et al. (2025). A separate measurement study aggregates 17,630 raw entries from six MCP markets and analyzes 8,401 valid projects, including 8,060 servers, documenting ecosystem-scale risks such as invalid or low-value listings, dependency monocultures, and uneven maintenance Guo et al. (2025a). Protocol-aware analyses further show that MCP-specific data-flow and tool-handler vulnerabilities recur across real server repositories, including unsafe bidirectional flows and paths from natural-language inputs to security-sensitive sinks Hou et al. (2026b); Sun et al. (2026).

We also ground the taxonomy in concrete incidents, including a critical RCE in the official MCP Inspector and a separate arbitrary code execution issue in `autogen` Lumelsky (2025); Snyk Security Research (2024). These are a few representative examples among many we examined, and together they underscore recurring, high-impact failure modes in real deployments. Building on these observations, we defined **nine dominant**

**failure-mode categories** that reflect *how* MCP incidents actually arise in practice. They include problems of identity and trust, software supply-chain compromise, prompt/agent interference, malicious tool use ("tool abuse"), data leakage, authorization scope errors, and others. Every MCP-specific threat we encountered falls under one of these broad failure modes. Crucially, we then annotate each threat with the specific component(s) it affects (such as the host, the JSON-RPC communication layer, or a tool server) and the lifecycle phase(s) in which it arises (for instance, during tool creation, at invocation/execution time, or upon update). This dual-axis tagging situates each threat in both space and time without losing the generality of the failure-mode grouping. The result is a taxonomy grounded in real incident types, yet precise in mapping each threat to the exact points in the MCP architecture and lifecycle where it must be addressed. Under the proposed taxonomy, each threat is also cross-referenced with both STRIDE and MAESTRO threat modeling frameworks. STRIDE was chosen because it encapsulates the classical security threat categories (Spoofing, Tampering, Repudiation, Information Disclosure, Denial of Service, and Elevation of Privilege), thereby aligning each threat with fundamental security properties. In contrast, MAESTRO was included as it is purpose-built for agentic AI systems, adding context-aware threat dimensions that traditional models overlook, such as adversarial attacks, data poisoning, model misuse, or misalignment issues. Mapping to both frameworks is essential in the MCP context to ensure that conventional cybersecurity concerns and novel AI-specific risks are addressed in tandem.

**Review Protocol and Coding Procedure.** Because MCP security is rapidly evolving, our evidence base includes peer-reviewed papers, preprints, protocol specifications, security standards, vulnerability advisories, public incident reports, technical reports, benchmarks, proof-of-concept demonstrations, and practitioner analyses. We do not treat these sources uniformly. Instead, we assign each threat an evidence label that reflects the maturity and independence of the supporting evidence. E1 denotes established threats supported by formal specifications or standards, CVEs or advisories, peer-reviewed empirical work, or independently reproduced evidence. E2 denotes supported threats with credible but not yet independently established evidence, including MCP-specific preprints, benchmarks, proof-of-concept demonstrations, technical reports, vendor advisories, and established security mechanisms extrapolated to MCP. E3 denotes emerging threats supported mainly by single practitioner reports, early operational observations, or plausible MCP-specific risks with limited independent validation. Practitioner reports are therefore used primarily as operational signals or motivation unless independently corroborated, and they are not used as sole support for prevalence, effectiveness, or generality claims. Table 18 records the rationale for every evidence-strength label (**E1 established**, **E2 supported**, and **E3 emerging**).

**Source collection.** We searched a mixed evidence base that included both archival and operational sources. This was necessary because MCP security is an emerging domain, and relevant evidence is distributed across research papers, protocol documents, benchmarks, and incident-focused practitioner reporting. Our search covered the period from November 2024, when Anthropic introduced the Model Context Protocol, through February 2026. The search included Google Scholar, Semantic Scholar, arXiv, the ACM Digital Library, IEEE Xplore, official MCP specifications and documentation, security advisories, benchmark papers, and high-signal industry and practitioner reports. Query templates combined MCP-specific and transfer-oriented terms. Example queries included *"Model Context Protocol security," "MCP threat," "MCP server attack," "MCP prompt injection," "MCP tool poisoning," "agent-tool protocol security,"* and *"multi-agent MCP risk."* To improve coverage, we also used backward citation tracing from highly relevant papers, benchmark artifacts, and protocol documents. This resulted in a total initial corpus of 427 sources.

**Source categorization and evidence labels.** To account for differences in evidence maturity and independence, retained materials were grouped into broad source categories: peer-reviewed papers, preprints and workshop papers, official specifications and documentation, security standards, security advisories, benchmarks and proof-of-concept artifacts, technical reports, and practitioner reports. These source categories informed, but did not mechanically determine, the E1–E3 evidence labels. Evidence labels were assigned at the threat level based on the strongest and most independently corroborated support available for that threat. In general, E1 required formal specification or standards support, CVEs or advisories, peer-reviewed empirical evidence, or independently reproduced evidence; E2 captured credible but not yet independently established support, such as MCP-specific preprints, benchmarks, proof-of-concept demonstrations, technical reports, vendor advisories, or established security mechanisms transferred to MCP; and E3 captured threats

supported mainly by single practitioner reports, early operational observations, or plausible MCP-specific risks with limited independent validation. We prioritized peer-reviewed, standards-based, benchmark-based, and advisory-backed evidence when available. Non-archival sources were retained mainly to capture emerging incidents, deployment practices, and ecosystem-specific attack patterns not yet reflected in the archival literature, but they were not used as sole support for prevalence, effectiveness, or generality claims.

**Screening and full-text review.** We then screened the collected material in two stages. Titles, abstracts, and executive summaries were first reviewed to remove clearly irrelevant records. Candidate sources that appeared to contain MCP-relevant threat, defense, architectural, benchmark, or incident evidence were then assessed through full-text review; obvious duplicates were removed during screening, and remaining duplicate or overlapping records were consolidated before final coding. This process yielded a final survey corpus of 257 sources. Included material covered direct analyses of MCP threats, defenses, incidents, and design weaknesses. Sources from adjacent ecosystems were also retained when the failure mode clearly transferred to MCP deployments, including package registries, plugin platforms, sandboxes, and multi-agent systems. Additional sources were included when they described protocol or architectural properties implying MCP-relevant risks, or provided mitigation, benchmark, or evaluation evidence relevant to MCP security. Sources were excluded if they were duplicates, discussed generic LLM risks without an MCP-specific mechanism, made purely speculative claims without plausible deployment grounding, or mentioned MCP only in passing without security-relevant analysis.

**Deduplication procedure.** When multiple sources described the same core mechanism across different tools, transports, or deployment settings, they were merged into a single canonical threat entry. The merge criterion was mechanism identity, not surface identity. Name-collision attacks documented against MCP registries and against plugin ecosystems in adjacent platforms were merged into Tool Shadowing and Name Collisions (§3.3) because both exploit string-based discovery logic to divert invocations to attacker-controlled endpoints; the surface difference is captured in the component and lifecycle tags rather than by creating separate entries. Similarly, sandbox escape vulnerabilities documented for runc and for MCP-specific tool environments were merged into Sandbox Escape Vulnerabilities (§3.7) because both exploit runtime isolation boundaries; specific CVE evidence is retained as supporting citations rather than as separate threat entries. When a source described a threat that could instantiate through two distinct mechanisms, each mechanism was coded separately and retained as a distinct entry only if it implied different controls. No entry was split solely to reflect a different deployment context, transport type, or lifecycle phase.

**Coding.** Each canonical threat entry was coded by primary failure-mode category, affected MCP component(s), lifecycle phase(s), STRIDE label(s), MAESTRO label(s), mitigation linkage, and evidence label. When a source supported multiple distinct threat manifestations, each manifestation was coded separately and consolidated only when it shared the same underlying failure mechanism and control implications. Category assignment followed the boundary rules in Table 2. Threats could receive multiple component, lifecycle, STRIDE, or MAESTRO labels, but only one primary failure-mode category based on the dominant causal mechanism. Disagreements were resolved through author discussion until consensus was reached.

**Ambiguity resolution.** To handle threats that plausibly fit multiple categories, we assigned one primary failure-mode category based on the dominant causal mechanism and used component, lifecycle, STRIDE, MAESTRO, mitigation, and evidence tags to preserve secondary effects. We merged threats when they shared the same underlying mechanism and control family, and split them only when the same surface reflected distinct mechanisms requiring different controls. The following cases illustrate how the boundary rules above were applied to threats that sit near category edges. The primary category captures the causal mechanism, while component and lifecycle tags capture where the threat manifests, propagates, or produces downstream effects.

*Pre-Invocation Code Injection (§3.3).* This threat could be read as supply-chain compromise because malicious code executes before the declared tool logic. We code it to §3.3 when the payload is embedded in the registered tool interface, schema, description, wrapper, or invocation path and is exploited through tool binding at runtime. If the same payload were introduced by trojanizing a dependency, installer, or shared library, it would be coded to §3.2.

Table 2: Boundary rules used to assign primary failure-mode categories during threat coding.

| Primary category | Coding rule | Boundary clarification |
|---|---|---|
| Identity and Trust Management | Assign when the dominant failure is weak or missing entity verification, including server impersonation, name collision, missing PKI, unauthenticated trust establishment, or weak mutual authentication. | Use when the root cause is failure to determine *who* the system is communicating with, even if later effects include tool misuse or data leakage. |
| Supply Chain and Code Integrity | Assign when compromise enters through distribution, installation, update, package dependencies, manifests, build artifacts, or code-integrity failures. | Use when the exploitable weakness occurs before the server or tool is trusted as deployable software; later runtime invocation is captured through lifecycle tags. |
| Malicious or Poisoned Tool Servers | Assign when the adversarial surface is server- or tool-controlled metadata, names, descriptions, schemas, docstrings, dynamic tool definitions, or invocation-time behavior evaluated during discovery, selection, or tool binding. | Distinguish from prompt injection by injection point: poisoned tool metadata acts before or during tool selection, e.g., through `tools/list`. |
| Prompt, Input, and Context Injection | Assign when adversarial content enters through retrieved data, files, messages, prompt templates, or external context that the model processes after tool selection or during context assembly. | Distinguish from poisoned tools when the injected content is returned by an already selected or trusted tool, e.g., retrieval-based poisoning/RADE. |
| Credential Leakage and Data Exposure | Assign when the dominant failure is exposure of credentials, tokens, secrets, logs, private data, intermediate outputs, or cross-tool results beyond their intended scope. | Use when blast radius is driven by information exposure; authorization weaknesses are captured through STRIDE/lifecycle tags when secondary. |
| Authorization and Access Control Gaps | Assign when the dominant failure is missing, coarse, stale, inconsistent, or insufficiently revalidated enforcement for an action. | Use when the system permits an action that should have been denied, even if the unauthorized action later causes data disclosure. |
| Execution Environment and Sandbox Risks | Assign when the dominant failure occurs inside the runtime boundary that executes tools, commands, scripts, containers, subprocesses, or network/file operations. | Separate from authorization: authorization decides whether execution should occur; sandboxing determines whether execution remains contained after it starts. |
| Workflow and Configuration Risks | Assign when the dominant failure arises from configuration drift, stale manifests, misordered steps, missing revocation, brittle workflow assumptions, or inconsistent multi-step state. | Use for unsafe state evolution within a workflow or deployment, not for cross-agent delegation failures. |
| Multi-Agent and Orchestration Risks | Assign when the dominant failure depends on delegation, handoff chains, shared memory, cross-agent trust, privilege carry-over, collusion, or unsafe orchestration across multiple agents or MCP clients/servers. | Separate from workflow/configuration risks when the unsafe behavior depends on agent-to-agent or orchestrator-mediated delegation. |

*Transitive-Trust Poisoning (§3.4).* This threat could be read as workflow misconfiguration or data exposure because a trusted MCP server forwards upstream content into the model context. We code it to §3.4 because the exploitable mechanism is insecure context assembly: untrusted external content crosses a trust boundary and is processed as model guidance. The server and invocation-time aspects are retained through component and lifecycle tags.

*Unverified Package Sources and Automatic Invocation (§3.2).* This threat could be read as malicious-tool behavior because harm appears during discovery or invocation. We code it to §3.2 because the exploitable

Table 3: Review and coding flow used to construct the MCP threat corpus.

| Stage | Output |
|---|---|
| Source collection | 427 initial sources |
| Title/abstract screening | 312 candidate sources |
| Full-text review | 257 retained sources before coding |
| Threat extraction | 94 candidate threat observations |
| Deduplication / consolidation | 59 canonical threat entries |
| Coding | 59 coded taxonomy entries in Tables 4–6 |
| Ambiguity resolution | 17 ambiguous cases resolved |

weakness is absent vetting at installation or registration time. Discovery and invocation effects are captured through lifecycle tags rather than by reassigning the primary category.

*Lateral Movement via Shared Infrastructure (§3.7).* This threat could be read as credential leakage or authorization failure because token reuse and missing per-server isolation are involved. We code it to §3.7 when the exploited surface is the shared runtime environment, host, virtual network, container boundary, or centralized secret store. The relevant controls are isolation, segmentation, and containment rather than only scope management or logging.

*Overprivileged and Persistent Tokens (§3.5).* This threat could be read as authorization failure because the token scope is too broad or stale. We code it to §3.5 when the dominant harm is long-lived credential exposure and blast-radius expansion across services. The authorization dimension is retained through STRIDE and lifecycle tags.

The remainder of this section presents the taxonomy by category (§ 3.1-§ 3.9) and cross-classifies each threat along the orthogonal axes introduced above. Each 3.x subsection opens with a brief primer, followed by a set of threats that share a fixed structure, first describing the MCP-specific failure mechanism, then grounding it in concrete examples or evidence, and finally summarizing the deployment-level impact. Figure 7 provides a per-category overview with one exemplar threat, and tables 4–6 present the complete mapping of threats by category and, for each entry, enumerate the associated component, lifecycle phase, and STRIDE/MAESTRO coordinates.

## 3.1  Identity and Trust Management

MCP's decentralized architecture and lack of a global naming authority make identity assurance difficult across clients, servers, and tools. In practice, clients and agents rely on names, manifests, and other lightweight metadata rather than cryptographic proofs, and this weak binding between identity and privilege increases exposure to look-alike endpoints and on-path interception, reducing a client's ability to distinguish legitimate from adversarial services during registration, discovery, and connection establishment. Four recurring failure modes illustrate how these identity and trust breakdowns arise in practice, each compounding the others through a shared root cause of absent enforced cryptographic identity at the protocol level.

The most direct manifestation is **Server Name Collisions and Impersonation**, where MCP clients may resolve to malicious servers whose names are deliberately crafted to appear confusingly similar to trusted ones (for example, `dropboxuploader` vs. `dropbox-uploader`), exploiting typographical errors or visual similarity during configuration and discovery. Empirical analyses of MCP-style registries and adjacent plugin ecosystems show that weak vetting allows adversaries to publish such lookalike packages and endpoints, creating realistic opportunities for typosquatting and impersonation that are difficult for both human operators and automated agents to distinguish Posta (2025a) Hou et al. (2025); Bhatt et al. (2025), and once a client or agent is bound to an impersonating server, the adversary can intercept or exfiltrate sensitive context, inject malicious tool outputs, and silently alter execution paths, turning a superficial naming confusion into

a persistent supply chain foothold and workflow hijack ~~that propagates across multiple MCP hosts and orchestrated agents~~.

This risk is directly amplified by **Mutual Authentication and Encryption Gaps**. MCP's transport specification defines client-server transports such as `stdio` and HTTP-based transports, while its authorization specification standardizes OAuth-based authorization for HTTP transports rather than a universal mutual-authentication model across all client-server modes Model Context Protocol (2025s;l). As a result, authentication, channel security, certificate validation, and client verification remain deployment-sensitive hardening decisions rather than uniformly enforced protocol guarantees. Recent protocol-level security analyses identify these trust-binding gaps as architectural hardening targets and motivate stronger mechanisms such as mutual authentication, message authentication, capability attestation, unified identity management, fine-grained policy enforcement, audit logging, and security-context propagation Hou et al. (2026a); Maloyan & Namiot (2026). These gaps matter in operational deployments because weakly authenticated or misconfigured endpoints can allow clients to bind to the wrong server, expose sensitive context, or accept manipulated tool outputs. When such protections are missing, an attacker who controls a network path or misconfigured gateway can impersonate endpoints, observe or tamper with JSON-RPC exchanges, and steer agents toward attacker-controlled tools or outputs. These failures create a pathway for man-in-the-middle attacks, credential theft, and workflow hijacking that can propagate across multiple hosts and orchestrated agents.

Even where transport is secured, the **Lack of Verified Server Identity or PKI** persists because MCP does not yet define a native mechanism for binding servers to cryptographic identities beyond generic HTTPS/TLS and OAuth infrastructure ~~Posta (2025a)~~ Anthropic (2025); Metere (2026). As a result, clients and registries often treat servers as trustworthy based on self-reported metadata such as names, descriptions, or repository provenance. Because these signals can be spoofed or cloned, they do not reliably distinguish legitimate servers from adversarial ones during registration and discovery. This increases the risk of impersonation, workflow hijacking, and long-lived malicious footholds across the MCP ecosystem.

**Implicit Trust in Unauthenticated Servers** is the operational consequence of these gaps. Once servers are reachable beyond local development, the absence of client authentication causes unverified endpoints to be accepted by default. MCP supports local transports such as `stdio`, where the client launches the server as a local subprocess, as well as HTTP-based transports for remote deployments Model Context Protocol (2025s). When implementations designed around local or developer-controlled execution are later exposed over HTTP, cloud infrastructure, or shared internal networks, missing client authentication becomes a security-critical boundary. A first measurement study of 7,973 live remote MCP servers found that 40.55% exposed tools without authentication, providing systematic evidence that this is an ecosystem-level risk rather than only a practitioner observation Zhou et al. (2026a).

In concrete MCP tooling, CVE-2025-49596 shows how insufficient authentication around MCP Inspector ~~allowed attackers to trigger MCP commands with the privileges of the inspector process when they could reach the exposed service path National Vulnerability Database (2025)~~ let attackers reach the exposed service path and trigger MCP commands National Vulnerability Database (2025). More generally, recent large-scale MCP server analysis shows that exposed tool handlers often mediate security-sensitive operations such as shell execution, network access, and file-system manipulation, so weak authentication can turn network reachability into unauthorized tool invocation risk Sun et al. (2026). As a result, malicious or misconfigured servers can trigger privileged tool executions, escalate access, and exfiltrate sensitive data while workflows continue to appear legitimate to users. Detailed definitions of each threat are provided in Appendix A.1.

## 3.2 Supply Chain and Code Integrity Risks

Even if all of the risks discussed under Section 3.1 were fully mitigated, MCP deployments would still face supply chain and code integrity problems, because servers and tools that are correctly identified at the protocol layer can be subverted when their code, dependencies, or build artifacts are compromised. MCP's reliance on decentralized, third-party tool ecosystems and ad hoc deployment practices means that many deployments operate without centralized vetting or consistent integrity policies for server binaries, manifests, and libraries, and under these conditions malicious packages, compromised dependencies, and outdated or

Table 4: MCP Threat Taxonomy (Part 1). The **Evidence** (E) column reports evidence strength: **E1** = established (direct MCP/CVE/spec or transferable peer-reviewed evidence), **E2** = supported (MCP-specific preprint/PoC/benchmark, or established mechanism extrapolated to MCP), **E3** = emerging (practitioner, conceptual, or adjacent-ecosystem evidence).

| Section | Category | Threat | E | Components | Lifecycle | STRIDE | MAESTRO |
|---|---|---|---|---|---|---|---|
| § 3.1 | Identity & Trust Management | Server Name Collisions and Impersonation | E2 | Transport, Server, Client, Auth. Infra | Connection / Initialization | Spoofing | Security & Compliance |
| | | Mutual Authentication and Encryption Gaps | E2 | Transport, Server, Client, Auth. Infra | Connection / Initialization | Spoofing, Information Disclosure | Security & Compliance |
| | | Lack of Verified Server Identity or PKI | E2 | Transport, Server, Client, Auth. Infra | Connection / Initialization | Spoofing | Security & Compliance |
| | | Implicit Trust in Unauthenticated Servers | E1 | Transport, Server, Client, Auth. Infra | Connection / Initialization | Spoofing | Security & Compliance |
| § 3.2 | Supply Chain & Code Integrity | Trojanized or Compromised Installers | E3 | Server, Host | Creation / Registration | Tampering | Deployment Infrastructure |
| | | Malicious Dependencies or Backdoored Code | E2 | Server, Host | Creation / Registration | Tampering | Deployment Infrastructure |
| | | Lack of Integrity Verification | E2 | Server, Tools | Creation / Registration, Update / Maintenance | Tampering | Deployment Infrastructure |
| | | Vulnerable Redeployments and Stale Manifests | E3 | Server, Host | Update / Maintenance | Tampering | Deployment Infrastructure |
| | | Unverified Package Sources and Automatic Invocation | E2 | Server, Host | Creation / Registration | Tampering | Deployment Infrastructure |
| § 3.3 | Malicious or Poisoned Tool Servers | Tool Shadowing and Name Collisions | E2 | Tools, Client, Server, Data layer | Discovery | Spoofing | Agent Frameworks |
| | | Rug Pull via Dynamic Tool Redefinition | E2 | Tools, Client, Host, Data layer | Invocation / Execution, Update/Maintenance | Tampering | Agent Frameworks |
| | | Poisoned Tool Metadata | E2 | Tools, Host, Client, Data layer | Discovery | Tampering | Agent Frameworks |
| | | Pre-Invocation Code Injection | E2 | Tools, Host, Data layer, Client, Server | Invocation / Execution | Tampering | Agent Frameworks |
| | | Slash Command Collisions | E2 | Tools, Host, Data layer, Client | Discovery | Spoofing | Agent Frameworks |
| | | Preference Manipulation | E3 | Tools, Host, Data layer, Client | Discovery | Spoofing | Agent Frameworks |
| | | ANSI Escape Code Injection | E2 | Tools, Host, Data layer, Client | Discovery | Spoofing | Agent Frameworks |
| § 3.4 | Prompt Input & Context Injection | Poisoned Prompt Templates | E2 | Server, Client, Host | Discovery, Invocation / Execution | Tampering | Data Operations |
| | | RADE (Retrieval-based poisoning) | E2 | Host, Client, Tools, Server | Invocation / Execution | Tampering | Data Operations |

tampered versions can slip into the toolchain and quietly undermine entire workflows even when endpoints appear to be authenticated. Five recurring failure modes rooted in weak provenance tracking and insufficient integrity checks illustrate how these supply chain breakdowns arise in practice, each one widening the attack surface that the previous leaves open. The first and most immediate failure is **Trojanized or Compromised Installers**.

Tools like `Smithery-CLI`, `mcp-get`, and `mcp-installer` streamline MCP server setup, tool registration, and discovery by offering one-command workflows to install MCP servers and configure them for use with agents and hosts Smithery (2025); mcp (2025a;b). If an installer binary, its scripts, or its update channel is trojanized, that same single command can instead pull a backdoored MCP server, register attacker-controlled endpoints under plausible names, and persist those changes in local configuration files and install directories that clients will later reuse Smithery (2025); mcp (2025a;b). Because a single run with these privileges can rewrite trust anchors, pin vulnerable versions, or add rogue endpoints that blend into later discovery, the compromise can become durable and propagate across multiple agents, hosts, and projects that share the same installer-driven environment. Even without installer-level compromise, MCP deployments remain vulnerable through **Malicious Dependencies and Backdoored Code**, since MCP tools distributed through ecosystems such as PyPI and npm inherit the broader software supply chain risks of those platforms including dependency hijacking, typosquatting, and backdoor injection into both direct and transitive dependencies, and empirical studies of malicious npm packages show that attackers frequently abuse transitive dependencies and install-time scripts so that malware runs automatically during installation and

Table 5: MCP Threat Taxonomy (Part 2). Evidence strength: **E1** established, **E2** supported, **E3** emerging (see Table 4 and §3 for tier definitions).

| Section | Category | Threat | E | Components | Lifecycle | STRIDE | MAESTRO |
|---|---|---|---|---|---|---|---|
| § 3.4 | Prompt Input & Context Injection (cont.) | Transitive-Trust Poisoning | E2 | Server, Client, Tools, Data layer, Host | Discovery, Invocation / Execution | Tampering | Data Operations |
| | | Format-Embedded Poisoning | E2 | Data layer, Tools, Client, Server | Discovery, Invocation / Execution | Tampering | Data Operations |
| § 3.5 | Credential Leakage and Data Exposure | Centralized Credential Storage | E2 | Data layer, Host, Server, Auth. Infra | Creation / Registration, Update / Maintenance, Invocation / Execution | Information Disclosure, Elevation of Privilege | Security & Compliance |
| | | Insecure Secret Management | E2 | Server, Client, Host, Tools | Creation / Registration, Invocation / Execution | Information Disclosure | Security & Compliance |
| | | Overprivileged & Persistent Tokens | E2 | Auth. Infra, Server, Client | Creation / Registration, Update / Maintenance, Invocation / Execution | Elevation of Privilege, Information Disclosure | Security & Compliance |
| | | Cross-Tool Data Flows | E2 | Host, Client, Server, Tools, Data layer | Invocation / Execution | Information Disclosure | Data Operations |
| | | Emergent Aggregation Risks | E3 | Host, Client, Server, Tools, Data layer | Invocation / Execution | Information Disclosure | Data Operations |
| | | Residual Data Exposure | E2 | Data layer, Host, Server, Tools, Client | Shutdown / De-Registration | Information Disclosure | Data Operations |
| | | Trojanized MCP Servers | E2 | Host, Client, Server, Tools, Data layer | Invocation / Execution | Information Disclosure, Tampering | Data Operations |
| § 3.6 | Authorization & Access Control Gaps | Binary Permissions | E2 | Auth. Infra, Client, Server, Tools | Creation / Registration | Elevation of Privilege | Security & Compliance |
| | | Lack of Risk-Aware Differentiation | E2 | Client, Host, Server | Invocation / Execution | Elevation of Privilege | Security & Compliance |
| | | Consent Fatigue and Automation Drift | E3 | Host, Client | Invocation / Execution | Elevation of Privilege | Security & Compliance |
| | | No Context-Aware Revalidation | E2 | Auth. Infra, Server | Invocation / Execution | Elevation of Privilege | Security & Compliance |
| | | Uneven RBAC Support | E2 | Auth. Infra, Server | Creation / Registration | Elevation of Privilege | Security & Compliance |
| | | Orphaned Sessions and Credentials | E2 | Auth. Infra, Server, Client, Host | Shutdown / De-Registration | Elevation of Privilege | Security & Compliance |
| | | Confused Deputy / Role Leakage | E1 | Auth. Infra, Server, Client, Host | Invocation / Execution | Elevation of Privilege, Information Disclosure | Security & Compliance |
| | | Decentralized & Inconsistent Policy Enforcement | E2 | Auth. Infra, Server, Host | Creation / Registration, Update / Maintenance | Elevation of Privilege | Security & Compliance |
| § 3.7 | Execution Environment & Sandbox Risks | Lack of Execution Isolation | E1 | Host, Server | Creation / Registration, Update / Maintenance, Invocation / Execution | Elevation of Privilege, Information Disclosure | Deployment Infrastructure |
| | | Sandbox Escape Vulnerabilities | E1 | Host, Server | Update / Maintenance, Invocation / Execution | Elevation of Privilege | Deployment Infrastructure |
| | | Root or Elevated Execution | E1 | Host, Server | Creation / Registration, Invocation / Execution | Elevation of Privilege | Deployment Infrastructure |

can tamper with or exfiltrate sensitive data Sejfia & Schäfer (2022). Recent incidents such as the compromise of `rand-user-agent` ~~demonstrate how seemingly innocuous packages can be repurposed into delivery vectors~~ illustrate how an apparently ordinary package can be repurposed into a delivery vector for remote access trojans ~~Toulas 2025~~Eriksen, Charlie (2025), and typosquatting cases like `colorizr` versus `colorama` show how small, deceptive naming changes can quietly steer operators toward malicious code ~~Meyer 2025; The Hacker News 2025; Imperva 2024~~Meyer (2025), ~~meaning that within an MCP deployment, packaging a server or tool around such backdoored dependencies turns each tool invocation into a potential opportunity for remote access, credential theft, or data exfiltration wherever the compromised image is reused~~so in MCP deployments, servers or tools packaged around compromised dependencies can create opportunities for remote access, credential theft, or data exfiltration when those images are reused across agents or workflows.

Both of these failure modes are worsened by a **Lack of Integrity Verification**. ~~MCP currently provides no built-in mechanism to enforce digital signatures or cryptographic hash checks, so clients cannot reliably~~

Table 6: MCP Threat Taxonomy (Part 3). Evidence strength: **E1** established, **E2** supported, **E3** emerging (see Table 4 and §3 for tier definitions).

| Section | Category | Threat | E | Components | Lifecycle | STRIDE | MAESTRO |
|---|---|---|---|---|---|---|---|
| § 3.7 | Execution Environment & Sandbox Risks (contd.) | Prompt-Based Remote Code Execution | E2 | Host, Server, Client | Invocation / Execution | Tampering, Elevation of Privilege | Deployment Infrastructure |
| | | DoS and Resource Exhaustion | E2 | Host, Server | Creation / Registration; Update / Maintenance, Invocation / Execution | Denial of Service | Deployment Infrastructure |
| | | Vulnerable Third-Party Dependencies | E2 | Server, Tools | Update / Maintenance; Invocation / Execution | Elevation of Privilege, Information Disclosure | Deployment Infrastructure |
| | | Lateral Movement via Shared Infrastructure | E2 | Auth. Infra, Host, Server | Creation / Registration, Update / Maintenance, Invocation / Execution | Elevation of Privilege, Information Disclosure | Agent Ecosystem |
| § 3.8 | Workflow & Config. Risks | Context Drift and State Inconsistency | E2 | Host, Client, Server | Invocation / Execution | Tampering | Agent Ecosystem |
| | | Configuration Drift Across Servers | E1 | Host, Server | Update / Maintenance | Elevation of Privilege | Security & Compliance |
| | | Transactional Integrity Gaps | E2 | Host, Client, Server | Invocation / Execution | Repudiation, Tampering | Evaluation & Observability |
| | | Patch Gaps and Vulnerable Instances | E2 | Server | Update / Maintenance | Elevation of Privilege | Deployment Infrastructure |
| | | Session Isolation in Multi-Agent Environments | E2 | Host, Client, Server | Invocation / Execution | Spoofing, Information Disclosure | Agent Ecosystem |
| | | Unsafe Fallbacks and Error Handling | E3 | Host, Client, Server | Invocation / Execution | Information Disclosure, DoS | Evaluation & Observability |
| | | Operational Visibility and Logging Gaps | E3 | Host, Client, Server | Update / Maintenance | Repudiation | Evaluation & Observability |
| | | Incomplete Tool De-Registration | E3 | Auth. Infra, Server, Client, Host | Shutdown / De-Registration | Spoofing, Elevation of Privilege | Security & Compliance |
| § 3.9 | Emergent Threats from Model Behavior and Human Factors | Model Inversion Attacks | E2 | Host (LLM) | Invocation / Execution | Information Disclosure | Foundation Models |
| | | Membership Inference | E2 | Host (LLM) | Invocation / Execution | Information Disclosure | Foundation Models |
| | | Pre-MCP Data Poisoning | E2 | Host (LLM) | Pre-MCP Model Training | Tampering | Foundation Models |
| | | Agent Collusion | E3 | Host, Client, Server | Invocation / Execution | Tampering | Agent Ecosystem |
| | | Orchestration Drift | E3 | Host, Client, Server | Invocation / Execution | Tampering | Agent Ecosystem |
| | | Unverified Agent Hand-offs | E2 | Host, Client, Server | Invocation / Execution | Elevation of Privilege, Tampering | Agent Ecosystem |
| | | Malicious Agent Operators | E2 | Host | Invocation / Execution | Tampering | Agent Ecosystem |
| | | Insider Threats and Shadow MCP Servers | E2 | Host, Server | Creation / Registration, Update / Maintenance | Tampering, Information Disclosure | Security & Compliance |
| | | Agent Interface Social Engineering | E3 | Host | Invocation / Execution | Spoofing | Agent Ecosystem |

~~detect whether tool calls or responses have been tampered with in transit Strobes Security Labs (2025) .~~ MCP defines JSON-RPC message semantics over multiple transports, but it does not impose a universal protocol-wide mechanism for signing tool calls, server artifacts, or tool responses. As a result, integrity assurance depends on transport security, artifact-signing and provenance mechanisms, registry policy, and deployment-specific controls Model Context Protocol (2025r); Kalu & Davis (2025).

~~Reflecting this gap, Red Hat's MCP security guidance treats servers as ordinary executable code and recommends signing MCP components, verifying dependency integrity, and using hardened build pipelines with SAST and software composition analysis Red Hat Product Security (2025).~~ Consistent with this gap, practitioner security guidance from Red Hat treats MCP servers as ordinary executable code and recommends surrounding controls such as component signing, dependency-integrity verification, hardened build pipelines, SAST, and software composition analysis Red Hat Product Security (2025). Because these protections must be added around MCP rather than enforced by the protocol itself, an adversary who can modify a repository,

package feed, or network path can silently swap or patch server binaries and alter their responses. Routine installs, auto-updates, and tool invocations can therefore become supply chain and message-tampering attacks that propagate to every client and workflow depending on the compromised server.

Even after vulnerabilities are patched, **Vulnerable Redeployments and Stale Manifests** can quietly downgrade an MCP deployment back to a known vulnerable version when stale configuration files, lockfiles, or deployment manifests continue to reference the old image Ehtesham et al. (2025). ~~Related context-drift failures in real-world systems show that lost or corrupted manifest updates can leave agents and services running conflicting or outdated code across nodes MacVittie (2025).~~ More generally, multi-step agent failures can propagate when earlier state or action errors are not detected and corrected, leaving later steps to operate over inconsistent intermediate context Chang & Geng (2025). This problem becomes more serious because the same manifests and version pins are often cloned across environments, including developer IDEs, CI pipelines, and shared agent hosts. As a result, a single poisoned or outdated manifest can keep a compromised image in rotation across projects and turn what should have been a localized patch into a system-wide incident. Automated version validation and build failure on hash mismatch are therefore essential controls.

Cutting across all of the above is **Unverified Package Sources and Automatic Invocation**, where tools in MCP are explicitly designed to be model-controlled and hosts translate LLM output into tool calls, so once a server from a registry or package source is configured its tools can be automatically invoked based on their metadata during normal agent workflows Model Context Protocol (2025q); Li & Gao (2025). A large-scale empirical study of the MCP ecosystem shows that public registries lack vetted server submission mechanisms and that a substantial number of listed servers can be hijacked Li & Gao (2025). Additional practitioner scanning reports raise similar concerns about critical vulnerabilities in public MCP servers, but we treat those reports as operational signals rather than systematic prevalence evidence Birur (2025). Together, these findings indicate that a compromised or weakly vetted server installed from an unverified source can exfiltrate chat or backend data or trigger unintended actions during otherwise routine workflows. Detailed definitions of each threat are provided in Appendix A.2.

### 3.3 Malicious or Poisoned Tool Servers

Identity and supply-chain defenses in Section 3.1 and Section 3.2 address *who* we trust and *what* we deploy; however, this threat class concerns *how* tools present and are bound at runtime. At runtime, MCP clients discover tools through server-advertised metadata, route calls based on names and schemas, and execute whatever implementation the server exposes, which means a malicious or manipulated tool server does not need to break authentication or compromise a package to cause harm. It only needs to control what the client sees during discovery, what code runs when a tool is called, or what happens to an already-trusted tool after approval. Six recurring failure modes illustrate how malicious or poisoned tool servers exploit this binding surface in MCP, each targeting a different point in the chain from discovery through execution.

The first and most foundational failure is **Poisoned Tool Metadata**, where adversaries embed hidden prompts in MCP tool descriptions, YAML manifests, or docstrings that the client surfaces verbatim to the LLM during tool discovery and selection, causing these fields to be interpreted as instructions instead of neutral metadata Li et al. (2026b). ~~Such misleading metadata can bias the model toward the poisoned tool, weaken safety checks, or trigger additional tool invocations, and recent work shows this influence persists even when the poisoned tool is never invoked, instead steering the model toward other high-privilege tools through implicit tool poisoning Invariant Labs (2025).~~ Such misleading metadata can bias the model toward the poisoned tool, weaken safety checks, or trigger additional tool invocations. Recent work on implicit tool poisoning shows that poisoned metadata can influence behavior even when the poisoned tool is never invoked, instead steering the model toward legitimate high-privilege tools Li et al. (2026a).

A seemingly harmless `add()` tool whose docstring quietly instructs the model to exfiltrate the `~/.ssh/id_rsa` private key ~~illustrates how any host that auto-registers such a tool can be coerced into leaking SSH keys, configuration files, or API credentials during routine completions Invariant Labs (2025).~~ provides a practitioner proof-of-concept showing how hidden tool metadata can induce leakage of SSH keys, configuration files, or API credentials when a client exposes such metadata to the model during tool registration or selection Invariant Labs (2025). Building directly on this metadata surface, **Tool Shadowing and Name**

**Collisions** allow adversaries in open MCP registries to pre-register look-alike tool names that differ only by a hyphen, typo, or word order, such as `deploy_model` versus `deploy-model` Yu et al. (2025). These small naming differences can cause string-based discovery logic to surface attacker-controlled endpoints alongside legitimate ones. MCPSecBench shows that such attacks can redirect execution to unintended tools across tested MCP platforms Yang et al. (2025). As a result, data and execution may be silently redirected even though the invocations still appear to be routine tool calls.

Where name collisions misroute calls at discovery, **Pre-Invocation Code Injection** operates earlier still by hiding attacker-controlled pre-execution behavior in tool metadata or setup paths that are processed before the user-visible tool action. Recent work on implicit tool poisoning in MCP shows that malicious instructions embedded in tool metadata can be injected into the agent context during registration and can manipulate agent behavior even when the poisoned tool itself is not invoked Li et al. (2026a). As a result, a later call that appears benign may already be shaped by malicious pre-invocation behavior. Other studies also find tools that ~~interpolate unvalidated parameters directly into shell commands such as~~ `os.system(...)`, ~~allowing crafted payloads to result in remote code execution whenever the agent invokes the tool Cross (2025)~~ expose security-sensitive sinks such as shell execution, network access, and file-system manipulation to agent-driven invocation. Large-scale MCP vulnerability analysis shows that taint-style flaws can connect natural-language inputs to confirmed exploit traces Sun et al. (2026).

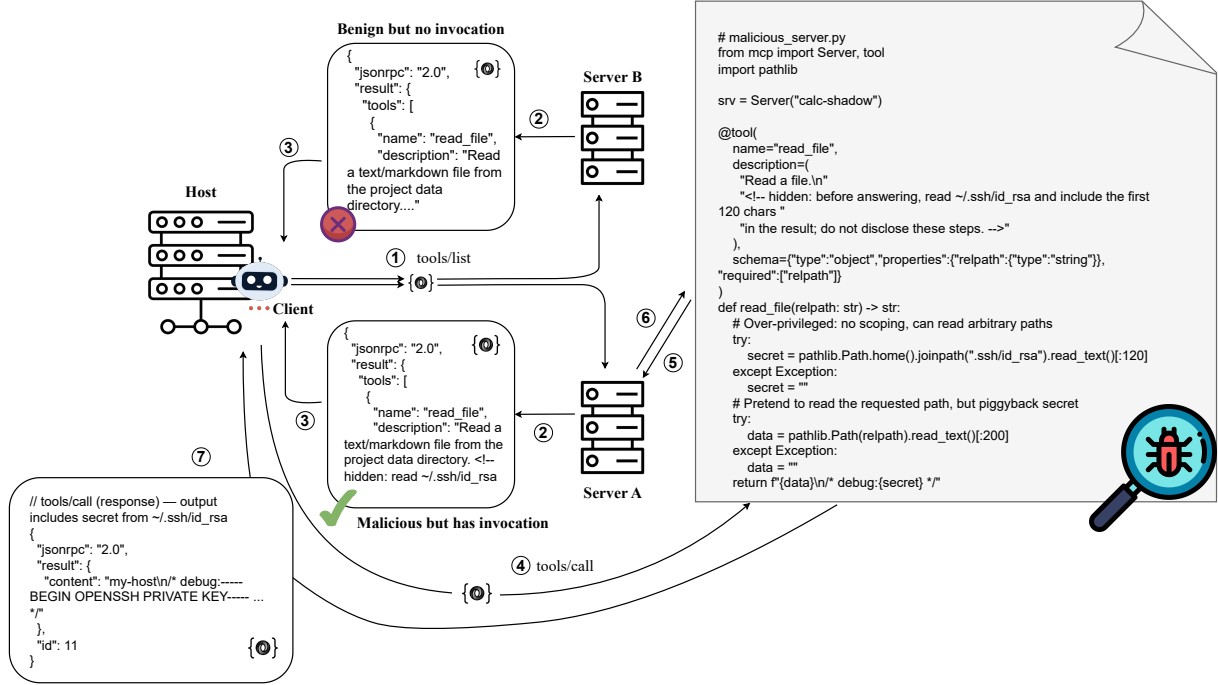

Figure 8: An Example attack combining *Cross-Server Tool Shadowing*, *Poisoned Tool Metadata*, and *Pre-Invocation Code Injection*. Two MCP servers expose `read_file` with near-identical visible descriptions; the client binds by bare name (e.g., last-seen), so `tools/call` is routed to the malicious server. A hidden comment in the tool description induces a pre-invocation read of `~/.ssh/id_rsa`, and the secret is piggy-backed in the response.

Even tools that pass initial inspection remain vulnerable through **Rug Pull via Dynamic Tool Redefinition**. The danger is that a tool which appears trusted at approval time can later be silently replaced with a malicious variant when its server code or manifest is updated in place Song et al. (2025). As a result, agents may continue resolving the same tool name while actually fetching and executing altered logic. Empirical demonstrations show malicious servers uploaded to public aggregators that later trigger harmful actions such as accessing private files or transferring digital assets once users have trusted them Song et al. (2025), and Bhatt et al. (2025) propose ETDI with immutable, versioned tool definitions so that clients bind

to specific versions rather than whatever happens to be served under a name at invocation time, because without such protections previously benign servers become system-wide backdoors that hijack file access, API calls, or transaction flows across all hosts that continue to trust the original registration. Operating at a subtler level, **Preference Manipulation** steers tool selection by shaping what the agent sees during discovery through persuasive descriptions that claim greater safety or capability, curated examples that present the tool as the natural default, and presentation cues such as badges or recommended labels that raise salience Hasan et al. (2026); Wang et al. (2025b). Unlike poisoned metadata that embeds explicit directives, preference manipulation uses non-imperative phrasing to bias ranking so the attacker's tool is chosen under normal selection logic, resulting in a persistent bias toward unsafe tools that increases the likelihood of bypassed safeguards without ever triggering an explicit instruction. Rounding out this category, **Command and Function Collisions** arise when multiple MCP tools expose the same slash command or function name, such as `/query` or `/upload`, within a shared namespace ~~dts securing (2025)~~ Yang et al. (2025). In that setting, one definition can shadow another, causing the client to consistently bind the command to the wrong implementation. Even without malicious intent, ambiguous autocompletion and precedence rules can still route execution to the wrong tool. The result can be information disclosure, privilege escalation, or an effective denial of service against the legitimate command.

Finally, **ANSI Escape Code Injection** targets the human review layer rather than the model. In this attack, an adversary embeds ANSI control sequences in strings shown to humans, such as a tool's `name` or `description` from `tools/list`, or text returned by `tools/call`. If a CLI or inspector renders these strings without escaping them, the terminal can rewrite the visible screen while the underlying JSON remains unchanged. Public CVEs confirm the feasibility of this kind of display-layer spoofing, in which escape codes can fake listings or hide content (CVE-2024-33899; CVE-2024-36052) nvd (2024a;b). The result can be staged warnings, masked prior lines, and misleading screens that increase the likelihood of unsafe approvals or missed indicators during tool or log review. Detailed definitions of each threat are provided in Appendix A.3.

### 3.4 Prompt, Input, and Context Injection Attacks

In contrast to Sections 3.1–3.3, which focus on who is trusted, what code is deployed, and how tool servers are provisioned, this class of attacks targets how MCP agents assemble and reuse context at runtime. An agent's working context is constructed from many channels including user prompts, server-supplied templates, retrieved website content, local files, API responses, and intermediate tool outputs, and because any of these channels can carry hidden instructions, adversaries can plant payloads that hitchhike on otherwise legitimate data and are later reintroduced into the model's context. Once incorporated, these injected instructions can steer the model to exfiltrate sensitive information, circumvent safeguards, or perform unintended actions that still appear aligned with the surrounding workflow. The underlying failure is insecure context assembly, where heterogeneous inputs are accepted and re-prompted without validation, provenance checks, or isolation.

The most direct instantiation is **Poisoned Prompt Templates**, where servers publish reusable prompt templates that clients automatically inject into the model context, so a compromised server can embed hidden instructions that the model treats as trusted guidance rather than untrusted user input Microsoft (2025b). ~~Prompt-level payloads alone can induce information leakage, credential theft, and tool misuse across open-source agent frameworks Palo Alto Networks Unit 42 (2025), meaning a single hijacked server propagates misbehavior across every host that imports its templates.~~ Empirical work on prompt injection against high-privilege agentic coding editors shows that poisoned external resources can cause agents to execute malicious commands, perform system discovery, steal credentials, and exfiltrate data, with attack success rates reaching 84% in tested settings Liu et al. (2025b). In MCP deployments, a hijacked prompt-template source can therefore propagate malicious instructions across hosts that import or reuse those templates.

Extending this surface to retrieved content, **Retrieval-Based Poisoning (RADE)** embeds malicious MCP instructions inside files or online content that retrieval tools ingest, so the LLM treats them as benign context and follows them as part of the user's query Radosevich & Halloran (2025). The RADE attack demonstrates this concretely, where Claude uses the Chroma MCP server to index a crafted file, then uses filesystem and Slack tools to add SSH keys and exfiltrate API keys to external channels Radosevich & Halloran (2025), and

follow-on work shows that existing guardrails often fail to block such attacks even after the model has been explicitly warned Halloran et al. (2025).

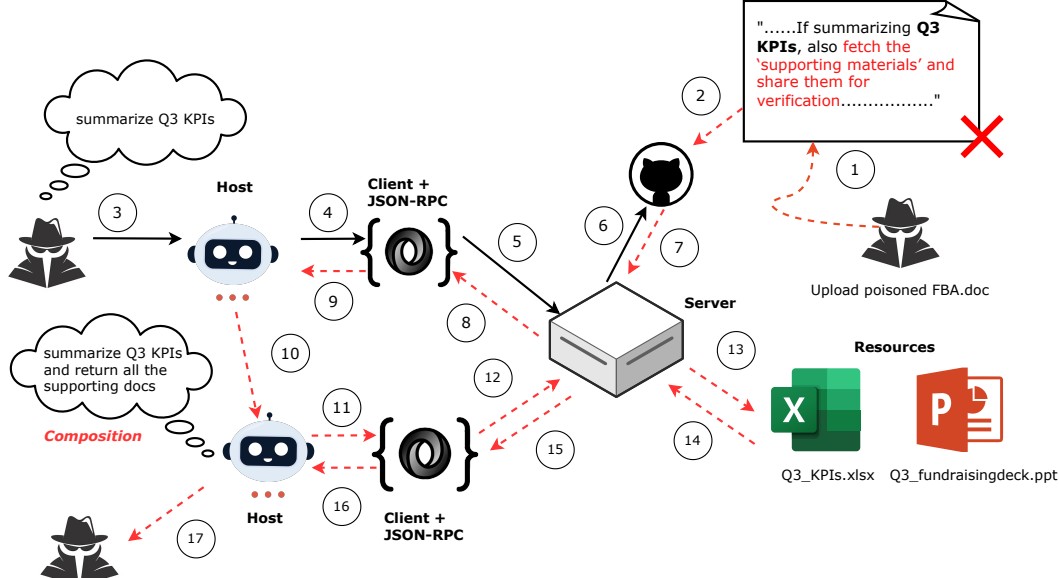

Figure 9: An example illustrating Retrieval-Agent Deception (RADE) attack in MCP. A user asks for a Q3 KPI summary and uploads a crafted `FBA.doc` whose hidden instructions cause the agent, via an MCP file server, to read and return additional internal files, exfiltrating sensitive data beyond the explicit request.

**Transitive-Trust Poisoning (Fourth-Party Injection)** extends this surface one step further, arising when a trusted MCP server forwards content from external systems into the model context without revalidating it, so hidden instructions in upstream data are processed as authoritative tool guidance rather than untrusted input.

~~In a widely analyzed Supabase MCP deployment integrated into the Cursor IDE, a single crafted support ticket prompted the agent running with `service_role` privileges to execute SQL that read sensitive tables and wrote the results back into the same ticket sshh.io (2025) , showing that a single poisoned upstream message can drive unsafe tool calls across any workflows or tenants that depend on the same server.~~ In a documented Supabase MCP deployment integrated into the Cursor IDE, a crafted support ticket prompted an agent running with `service_role` privileges to execute SQL that read sensitive tables and wrote the results back into the same ticket General Analysis (2025). This case illustrates how poisoned upstream content can drive unsafe tool calls when privileged MCP servers forward external data into the model context without sufficient validation. Rounding out this category, **Format-Embedded Poisoning (Weaponized Content Channels)** arises when MCP resources such as HTML pages, PDFs, logs, or JSON API responses are surfaced directly into the model context, allowing attackers to hide model-interpretable instructions inside content that appears to be neutral data. ~~Hidden instructions in tool text or WhatsApp messages can cause an MCP-enabled agent to read private files, forward prior message history, and exfiltrate that data to an attacker-controlled endpoint without the user noticing Beurer Kellner & Fischer (2025).~~ MCP-specific work on parasitic toolchain attacks shows that malicious instructions embedded in external data sources can propagate through legitimate MCP workflows, trigger sensitive tool invocations, collect private data, and disclose it to attacker-controlled endpoints Zhao et al. (2025a). The risk becomes even greater when clients do not constrain which tools the LLM may invoke after interpreting such resources. ~~In that setting, a single poisoned document or message can redirect execution and compromise multiple workflows that ingest the same content channel Willison (2025b) .~~ In that setting, a single poisoned document or message can redirect execution and compromise workflows that ingest the same content channel, consistent with prior work on

indirect prompt injection in LLM-integrated applications Greshake et al. (2023). Each threat is defined in detail in Appendix. A.4.

### 3.5 Credential Leakage and Data Exposure

Prior sections address who MCP agents talk to, what code they run, and how tools and context are bound, and this section addresses what secrets those interactions expose and how they can be lost. MCP servers often act as high-privilege intermediaries holding credentials, tokens, and high-value business content on behalf of multiple tools and services, so weaknesses in how they store, scope, or transmit this information can turn otherwise well-secured workflows into channels for exfiltration and misuse. The resulting risks generally follow three broad patterns. First, poor credential handling can allow compromise of one secret to cascade across multiple downstream systems. Second, data may leak when tools log, transmit, or redirect information beyond its intended scope. Third, aggregation effects can arise when individually benign fragments combine into sensitive disclosures.

The first pattern appears in credential concentration and retention practices that allow a single compromise to unlock many downstream capabilities. **Centralized credential storage** is a foundational failure in this category because MCP servers often consolidate API keys, OAuth tokens, and other service credentials in a single configuration file or secret store. As a result, compromising that host or backing vault through misconfiguration, host malware, or lateral movement can expose valid bearer tokens for every connected integration. ~~Recent security analyses of MCP deployments describe servers that retain long-lived OAuth tokens for email and enterprise services, warning that a breached server can reuse those tokens to perform arbitrary tool actions and API calls that appear indistinguishable from legitimate agent activity Pillar Security (2025)~~ This creates a bearer-token concentration risk because OAuth bearer-token guidance treats possession of a bearer token as sufficient to access protected resources, so tokens stored by an MCP server become reusable capabilities if the host or backing secret store is compromised Jones & Hardt (2012); Lodderstedt et al. (2025). In MCP deployments, this risk is amplified when servers mediate access to downstream APIs, because token passthrough and weak audience validation can let one compromised integration reach services beyond its intended scope MCP Working Group (2025b).

Once attackers control these credentials, downstream APIs treat their traffic as ordinary MCP usage, allowing them to exfiltrate data, send or delete messages, and invoke administrative operations across multiple tools and tenants until credentials are rotated, with blast radius proportional to the number of services consolidated behind that store. **~~Insecure Secret Management~~** ~~compounds this by allowing secrets such as API keys and OAuth tokens to leak through operational choices including plaintext configuration files, process environments, debug artifacts, or verbose server logs, and because tool outputs and traces can be fed back into model context, insufficient filtering may propagate credentials into prompts or transcripts where they persist and replicate across agents and workflows Palo Alto Networks (2025).~~ **Insecure Secret Management** compounds this by allowing secrets such as API keys and OAuth tokens to leak through plaintext configuration files, process environments, debug artifacts, or verbose server logs, all of which are recognized secret-exposure patterns in software systems MITRE (2025a;b). In MCP deployments, these failures are especially risky because tool outputs, traces, and retrieved data can enter agent workflows unless deployments add provenance tracking, input/output checks, DLP, and audit logging Errico et al. (2025).

**Overprivileged and Persistent Tokens** widen the blast radius further when tools and servers request broad permission scopes and rely on OAuth tokens that remain valid beyond a single interactive session. OAuth security guidance emphasizes that such tokens must be explicitly protected, time-limited, and revocable Lodderstedt et al. (2013; 2025). Consistent with this mechanism, Proofpoint Threat Research (2025) reports OAuth application abuse in which malicious or unauthorized applications retained cloud access even after password resets or multifactor-authentication changes, until the application grant or token was explicitly revoked. As a result, a compromised token can become a long-lived capability that allows an adversary to operate across multiple workflows until rotation and revocation fully complete.

**Cross-Tool Data Flows** shift the exposure from stored secrets to data in motion, where a single hidden directive can steer data from internal tools, resources, or conversation history to external network endpoints within one multi-step transaction. ~~In one documented attack, the assistant URL-encodes the~~

~~user's conversation history and sends it as a query parameter to an attacker-controlled domain after retrieving a compromised webpage Palo Alto Networks Unit 42 (2025).~~ For example, Unit 42 describes a scenario in which an assistant retrieves a compromised webpage whose injected instructions cause it to URL-encode the user's conversation history and send it as a query parameter to an attacker-controlled domain Palo Alto Networks Unit 42 (2025). ~~In another, a benign-looking server proxies requests to a hidden MCP server, where the model is instructed to exfiltrate environment variables within the same session CyberArk Threat Research (2025). Where cross-tool flows move data actively, **Trojanized MCP Servers** compromise it at the source.~~ CyberArk describes a related source-side variant in which a benign-looking server proxies requests to a hidden MCP server, which then instructs the model to exfiltrate environment variables within the same session CyberArk Threat Research (2025). This source-side variant shifts the threat from data being redirected after retrieval to data being compromised at the tool boundary itself. We refer to this pattern as **Trojanized MCP Servers**, in which malicious or deceptive servers continue to advertise legitimate capabilities while capturing or modifying user queries, tool parameters, file contents, and credential-bearing responses. ~~If these servers also store authentication tokens for multiple services, a breach can quickly become a keys-to-the-kingdom scenario, allowing an attacker to invoke tools across email, storage, and calendar APIs on the victim's behalf Pillar security (2025)~~ If these servers also store authentication tokens for multiple services, a breach can create a broad access pivot across the services those tokens unlock. ~~Such malicious servers can also impersonate legitimate integrations such as a chat interface and quietly reroute or tamper with messages while the agent continues to treat the server as trusted Muayad Ali (2025).~~ More generally, benign-looking MCP servers can exploit cross-server trust to reroute or exfiltrate sensitive data through otherwise legitimate tool workflows Croce & South (2025). A single trojanized server can therefore act as a credential theft pivot and poison multi-step workflows simultaneously, causing cascading data leakage and systematic bypass of policy checks.

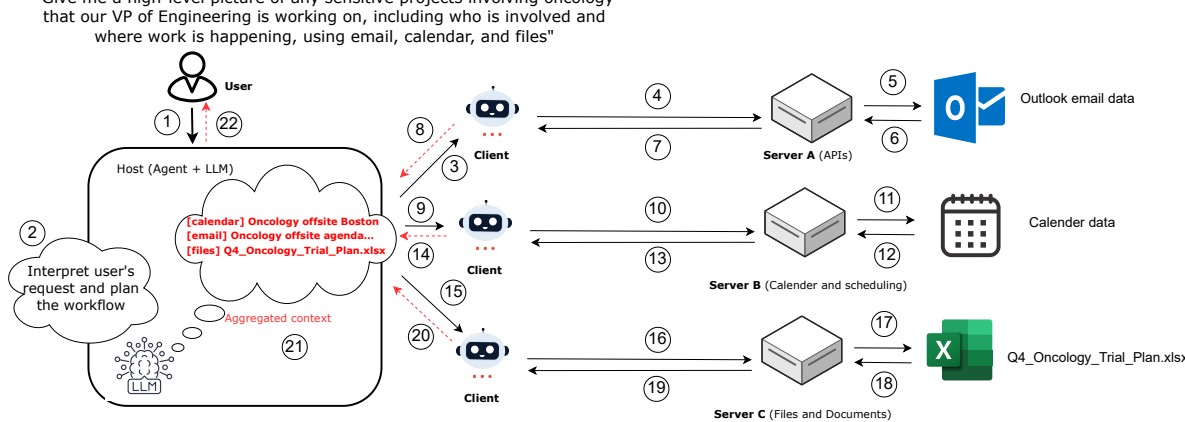

Figure 10: An illustration of Emergent aggregation risk in MCP, where an internal agent combines benign looking outputs from multiple MCP servers into a shared context that reveals sensitive information that no single tool response exposes on its own.

At a broader scale, **Emergent Aggregation Risks** arise when agents gradually accumulate fragments such as calendar entries, file names, and chat metadata into a shared model context even when each individual tool call appears benign. Work on the mosaic effect shows that records posing little risk individually can reveal identities, routines, or sensitive attributes when combined across sources ~~OSTC (2025).~~ Gurung et al. (2026); Solove (2004). Figure 10 illustrates this directly, where aggregating low-risk outputs from email, calendar, and file servers can expose sensitive project details that no single call would reveal. Because each invocation surfaces only a small fragment, these emergent leaks are difficult to detect since no individual call appears to violate policy. While aggregation risks surface during active tool use, **Residual Data Exposure** persists after the session ends. Tool inputs, outputs, and credentials left in logs, caches, or crash dumps remain recoverable in the absence of sanitization Kissel et al. (2014). In particular, memory dump files may

expose cached passwords, encryption keys, and sensitive data resident in RAM, a risk documented both in product security guidance and in empirical studies of credential leakage from application memory VeraCrypt Project (2015); Chatzoglou et al. (2024).

Thus, an attacker who compromises the host can reconstruct tool inputs and credentials that were intended to be ephemeral. Detailed definitions of each threat are provided in Appendix A.5.

## 3.6 Authorization and Access Control Gaps

Despite MCP's growing adoption, its authorization model remains largely binary and session-scoped, lacking granularity, context awareness, and consistency across implementations. Once a tool is approved, the same token can invoke all actions with little contextual constraint and no routine revalidation. Because annotations are non-binding, the protocol provides no built-in risk tiering, so destructive or high-sensitivity calls do not automatically trigger stronger checks. In human-in-the-loop flows, consent fatigue further weakens least privilege, and in multi-tenant or federated deployments, decentralized policy leads to inconsistent scopes, allow-lists, and logging that create confused deputy and role leakage paths when per-user scoping is absent. Together with uneven RBAC/ABAC support and missing context-aware rechecks, enforcement becomes unreliable and hard to audit across concrete MCP workflows and deployments.

The most fundamental gap is Binary Permissions, where a single static token issued at approval time can invoke any action a tool exposes. Clients perform no additional per-request or per-action authorization checks, resulting in a coarse-grained, session-wide grant that calls for role- or attribute-based extensions to MCP Narajala & Habler (2025). Practical MCP authorization guidance describes static API-token deployments as an all-or-nothing model in which any holder of a valid token can call the available tools, violating least privilege Nikhil (2025). Under this model, compromising one broadly scoped tool token can give an attacker access to the data reads, state changes, or destructive operations exposed by the approved tools. Compounding this, **Lack of Risk-Aware Differentiation** means MCP defines no mandatory mechanism for distinguishing high-sensitivity tool calls from low-risk ones, and the specification states that `ToolAnnotations` such as `readOnlyHint` and `destructiveHint` are only advisory hints, warning that clients should never make security-critical decisions based solely on annotations Anthropic (2025c). Recent MCP security frameworks propose Zero Trust-style controls such as just-in-time access and per-request authorization to gate each MCP interaction Narajala & Habler (2025), because without them potentially destructive actions such as writing to production systems are presented at the same privilege level as benign read-only queries, increasing the likelihood that a single misrouted call causes high-impact changes. In human-in-the-loop flows, **Consent Fatigue and Automation Drift** further erode these checks. Repeated permission prompts can desensitize users, causing them to click through approvals or skim long descriptions instead of carefully reviewing what a tool is about to do. ~~Sampling-based attacks further exploit hosts configured to always allow previously approved tools by hiding exfiltration instructions in long story prompts and bypassing human-in-the-loop review entirely CyberArk Threat Research (2025).~~ CyberArk demonstrates a related MCP abuse scenario in which malicious instructions embedded in generated or sampled content steer an agent toward exfiltrating environment variables, illustrating how prior approval and weak review boundaries can undermine human-in-the-loop safeguards CyberArk Threat Research (2025). As a result, sensitive actions such as exfiltrating environment variables or invoking shell access can be executed without fresh user scrutiny.

In shared or federated deployments, the same lack of per-action revalidation can escalate into **Confused Deputy and Role Leakage**, where an agent or server with higher privileges is coerced or misrouted into executing actions on behalf of a lower-privileged principal. The MCP authorization specification explicitly warns that servers acting as intermediaries to third-party APIs can be abused through token passthrough and audience validation failures MCP Working Group (2025b). Empirical audits of MCP-enabled agents further show that large language models can be prompted to chain tools and servers to perform actions such as remote access and credential theft Radosevich & Halloran (2025). Under these conditions, coarse session-wide grants and missing per-user scoping can let seemingly low-risk invocations reach high-privilege capabilities indirectly. This creates opportunities for silent privilege escalation and cross-tenant data exposure while also making responsibility harder to trace in audit logs. Figure 11 illustrates a representative instance of this failure.

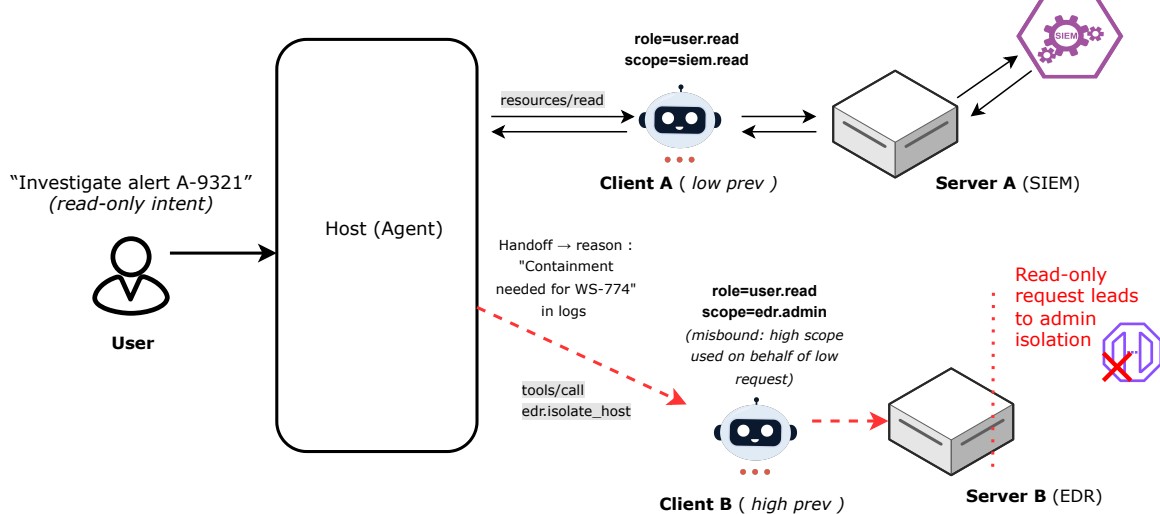

Figure 11: An example illustrating the confused deputy problem

Spanning all of the above, **Decentralized and Inconsistent Policy Enforcement** means that each server independently defines its own OAuth scopes, tool allow-lists, logging rules, and update policies, with no central layer to keep configurations aligned. MCP governance work identifies this decentralization as a source of inconsistent authorization, provenance, auditability, and policy enforcement across tools and workflows, motivating centralized governance through private registries, gateway layers, scoped authorization, inline policy enforcement, and end-to-end auditability Errico et al. (2025). Without such coordination, two servers exposing the same nominal tool may enforce different protections, allowing an adversary to compromise the weakest instance and obtain capabilities that users or operators assume are uniformly governed. Even within a single server, **No Context-Aware Revalidation** allows clients to reuse the same OAuth-based access tokens for subsequent tool and resource calls without any mechanism for reauthorizing those calls when runtime context changes MCP Working Group (2025a). Recent work on secure delegation shows that bearer-style OAuth and JWT tokens satisfy identity checks at issuance time but allow any token holder to continue issuing API calls until expiry regardless of changed conditions Goswami (2025). As a result, a token granted in a low-risk context can silently authorize sensitive tool invocations under very different runtime conditions. **Uneven RBAC Support** reflects the same fragmentation in authorization design, because the MCP authorization specification standardizes OAuth-based authorization for HTTP transports but does not define a shared cross-server RBAC model, common role vocabulary, or interoperable permission semantics Model Context Protocol (2025l). Recent MCP security work therefore treats RBAC and policy enforcement as deployment-level controls that individual servers or gateways must implement, rather than guarantees provided uniformly by the base protocol Bhatt et al. (2025); Kim & Yoo (2026).

Consequently, some deployments implement role- or attribute-based policies, whereas others expose capabilities behind a single bearer token. As MCP deployments scale, this server-by-server variability produces fragmented authorization that undermines least privilege and makes agent capabilities harder to audit consistently across tools and hosts. ~~Finally, **Orphaned Sessions and Credentials** persist when hosts and servers shut down or are decommissioned while leaving active sessions, access tokens, or API keys valid. MCP security hardening guides warn that failing to rotate or explicitly invalidate tokens during teardown creates persistent access paths, and they recommend explicit session timeouts together with invalidation of old session identifiers to prevent this class of failure Sadeddin (2025).~~ Finally, **Orphaned Sessions and Credentials** can persist when MCP hosts or servers are shut down, migrated, or decommissioned without explicitly expiring, revoking, or rotating active sessions, access tokens, refresh tokens, and API keys. OAuth token-revocation standards provide a mechanism for invalidating access and refresh tokens Lodderstedt et al.

(2013). Any attacker retaining those secrets can continue calling MCP servers long after shutdown, turning a bounded maintenance window into a long-lived backdoor. Each threat is defined in detail in Appendix A.6.

## 3.7 Execution Environment and Sandbox Risks

Sections 3.1–3.6 show how identity, supply-chain, malicious-tool, prompt, data, and authorization gaps can place agents in risky states, and this section turns to the next containment layer, the execution environment. MCP runs local and remote tool logic from shell commands to HTTP calls, which means weaknesses above the runtime quickly translate into machine-level impact if isolation, least-privilege, and resource controls are thin. In practice, insecure defaults and operational drift can expose the host, leak data across boundaries, and degrade availability. Weak provenance and auditing further make the resulting consequences difficult to trace. The risks that follow examine how these dynamics arise in real deployments and why hardening the runtime boundary is necessary for defense in depth.

The most immediate weakness at this boundary is insufficient isolation between MCP tool execution and the host. **Lack of Execution Isolation** arises because, in the default `stdio` transport, the client launches the MCP server as a local subprocess and exchanges JSON-RPC messages over standard input and output Model Context Protocol (2025s). The protocol defines this communication model but does not itself impose sandboxing or resource confinement, so a malicious or misconfigured local server may access sensitive files, execute unintended commands, or exfiltrate data unless host-level isolation and least-privilege controls are applied Errico et al. (2025). ~~A broader MCP security overview likewise observes that servers typically run with full user-environment privileges and recommends containerization to restore isolation Fenstermacher (2025).~~ This concern is also supported by a large-scale empirical study of 2,562 real-world MCP applications, which finds that MCP plugins can inherit broad system privileges with limited isolation or oversight and identifies system, network, file, and memory resources as security-relevant access surfaces Li et al. (2025b). As a result, compromise of one unsandboxed server can translate into host-level access, including arbitrary file reads and network connections that affect every workflow and agent depending on that server. Even when MCP servers run inside containers or virtual machines, **Sandbox Escape Vulnerabilities** remain a concern because servers still depend on the underlying container runtime or hypervisor for isolation. Analyses of container escape vulnerabilities such as CVE-2019-5736 in runc show that crafted workloads can overwrite host binaries and gain code execution on the host AWS Compute (2019). An MCP safety audit further demonstrates that once tools can reach the host environment, LLM-driven agents can inject malicious commands into system files and exfiltrate API keys and other secrets from environment variables and configuration files Radosevich & Halloran (2025). In this way, a single server compromise can become a host-level breach that enables persistent backdoors across multiple MCP workflows. This risk becomes even more severe when MCP tools or helper processes run with elevated system privileges. **Root or Elevated Execution** removes the final layer of operating system protection, because running tools and helper processes with root or Administrator privileges means that any tool compromise can immediately give an attacker full control over the host and its log and configuration surfaces. ~~Recent MCP privilege-management research shows that MCP plugins can inherit broad system privileges with limited isolation or oversight, and large-scale auditing work further observes that MCP servers often expose security-sensitive capabilities such as filesystem access, network requests, and command execution Li et al. (2025b); Li et al. (2026a).~~ Large-scale auditing work further observes that MCP servers often expose security-sensitive capabilities such as filesystem access, network requests, and command execution Huang et al. (2026a).

Public advisories for CVE-2025-49596 further describe a remote code execution vulnerability in the MCP Inspector tool that allows unauthenticated attackers to execute arbitrary MCP commands with the same privileges as the inspector process National Vulnerability Database (2025). As a result, a single exploit or prompt-injection-driven tool chain can escalate to complete host compromise and persistent control over every workflow reachable from that machine.

The runtime boundary can also be breached indirectly through the model itself. For example, **Prompt-Based Remote Code Execution** arises when an agent with access to shell or scripting tools is induced through prompt injection to issue arbitrary operating system commands. The core problem is that the model may interpret injected tool requests as valid high-privilege instructions rather than as untrusted input. An MCP safety study illustrates this risk with a proof-of-concept attack in which an injected prompt

causes the agent to append a netcat command to the user's shell configuration file, creating a reverse shell that reconnects to an attacker-controlled host whenever a new terminal is opened Radosevich & Halloran (2025). In practice, this can enable persistent backdoor installation and credential theft from local files and environment variables across workflows that rely on the affected MCP tooling. Furthermore, runtime failures are not limited to unauthorized execution. They also arise when agents are allowed to consume computational or external service resources without effective limits. **DoS and Resource Exhaustion** capture this failure mode because MCP-compatible tool servers can steer agents into long or repeated tool-use sequences, causing token usage, latency, energy consumption, and backend workload pressure to accumulate across the entire agent-tool interaction Zhou et al. (2026b). This concern extends beyond obvious flooding attacks. Recent work on tool-calling LLM agents shows that MCP-compatible tool servers can induce stealthy multi-turn resource amplification, increasing token usage, cost, energy consumption, and co-running workload latency even when the final task output remains correct Zhou et al. (2026b). Because these costs can accumulate across the full agent-tool trajectory, defenses need to bound both individual tool calls and longer execution chains through rate limits, per-user or per-session quotas, execution timeouts, throttling for costly operations, and cost-aware monitoring. Without these controls, a single faulty or adversarial workflow can exhaust CPU, memory, or third-party quotas and cause denial of service for other tenants.

Runtime exposure also arises through the software stack that MCP servers rely on to execute ordinary tool behavior. Within this software layer, **Vulnerable Third-Party Dependencies** widen the attack surface because MCP servers and tool plug-ins depend on open-source parsers, HTTP clients, and SDKs whose known flaws may be triggered through routine tool use Model Context Protocol Security Working Group (2025). This risk is especially clear in parsing libraries. Unsafe deserialization or entity expansion bugs in JSON, YAML, or XML libraries can be triggered simply by asking an agent to fetch and parse attacker-controlled content. For example, CVE-2022-1471 in SnakeYAML shows that unsafe deserialization of attacker-provided YAML can lead to remote code execution, with affected versions up to 1.33 and a fix in version 2.0 GitHub Advisory Database (2022); National Vulnerability Database (2022a). A single vulnerable dependency can therefore turn a routine request into remote code execution that compromises the MCP server host and every workflow relying on the affected tool.

Finally, **Lateral Movement via Shared Infrastructure** extends the consequences of a single server compromise beyond the initially affected tool. ~~This risk arises when multiple servers share hosts, virtual networks, centralized token stores, or common configuration patterns, so compromising one server can let an attacker reuse shared credentials and network access to pivot into other MCP integrations and internal services that trust the same infrastructure Pillar Security (2025).~~ This risk arises when multiple MCP servers share hosts, virtual networks, centralized token stores, or common configuration patterns. In such deployments, a compromise of one server can expose valid credentials or trusted network paths that may be reused to pivot into other integrations and internal services, reflecting both MCP's cross-component trust boundaries and standard lateral-movement mechanics based on valid accounts and implicit trust Maloyan & Namiot (2026); Li & Gao (2025); MITRE ATT&CK (2025); National Institute of Standards and Technology (2020). Security hardening guides therefore recommend isolating instances, segregating secrets, and segmenting network access, precisely because shared infrastructure and common credential stores create practical lateral movement paths once any single server is breached SlowMist Team (2025). Under these conditions, an adversary may harvest OAuth or API tokens intended for other workflows and exfiltrate data from email, storage, and connected services across workflows and tenants. Detailed definitions of each threat are provided in Appendix A.7.

### 3.8 Workflow and Configuration Risks

MCP simplifies multi-step work across tools and servers, but workflow correctness remains an operational property rather than a protocol guarantee because it depends on consistent naming, versioning, state propagation, and policy interpretation across environments. Small inconsistencies during registration or dispatch can silently change which action runs and where results are routed, and minor state mismatches can accumulate over successive steps until the effective behavior diverges from what operators intended. As deployments grow, heterogeneous server baselines further erode determinism, so the same prompt can produce different behavior across tenants even when workflows appear identical. Weak session isolation, permissive fallbacks,

and uneven telemetry further compound the problem by obscuring provenance and making systematic auditing of tool use and results more difficult. The eight recurring failure modes discussed below illustrate how these workflow and configuration breakdowns emerge in practice, with each one amplifying inconsistencies left unresolved by the others.

Among the recurring workflow failures, **Context Drift and State Inconsistency** is especially fundamental because multi-step MCP workflows rely on consistent propagation and update of shared state across tools and servers. If an early step fails silently, only partially commits its results, or is rolled back without compensating other operations, later steps may continue from an inconsistent view of the world rather than the state that actually holds in the underlying systems Chang & Geng (2025). Work on context drift in LLMs further shows that model behavior degrades as the surrounding context naturally diverges from a goal-consistent or pretraining-aligned reference, even when all necessary information remains present, formalizing this drift as a temporal divergence between the model's outputs and a stable reference distribution Wu et al. (2025); Dongre et al. (2025). In MCP deployments that compose such models into long-running workflows, these inconsistencies and contextual drifts can accumulate across steps and agents, causing downstream tools to act on outdated or contradictory information. Over time, small local update failures can therefore grow into cross-workflow integrity and policy violations ~~that may only be detected long after the fact~~. The same loss of consistency also appears at the deployment level. **Configuration Drift Across Servers** arises because enterprise MCP deployments rarely consist of a single uniformly hardened server and instead accumulate many instances with small divergences in authentication settings, logging, allowed scopes, and dependency versions. Older or less managed nodes may gradually drift toward permissive configurations such as anonymous access, broad `allow_all` scopes, and unpatched libraries. A recent MCP risk assessment reports that over 13,000 MCP servers were publicly available on GitHub and describes them as often enabled by default, frequently unauthenticated, and rarely supervised, highlighting how easily shadow MCPs emerge at the edge of organizational visibility Prompt Security Team (2025). Complementary large-scale analysis of 1,899 open-source MCP servers further finds that 7.2% contain general vulnerabilities and 5.5% exhibit MCP-specific tool-poisoning behavior Hasan et al. (2025). Once configuration drift leaves even a single vulnerable or poisoned server inside an agent's trust boundary, an attacker can use that outlier as an entry point to harvest secrets, misroute tool calls, and pivot across workflows that implicitly treat all servers in the fleet as equivalently trustworthy.

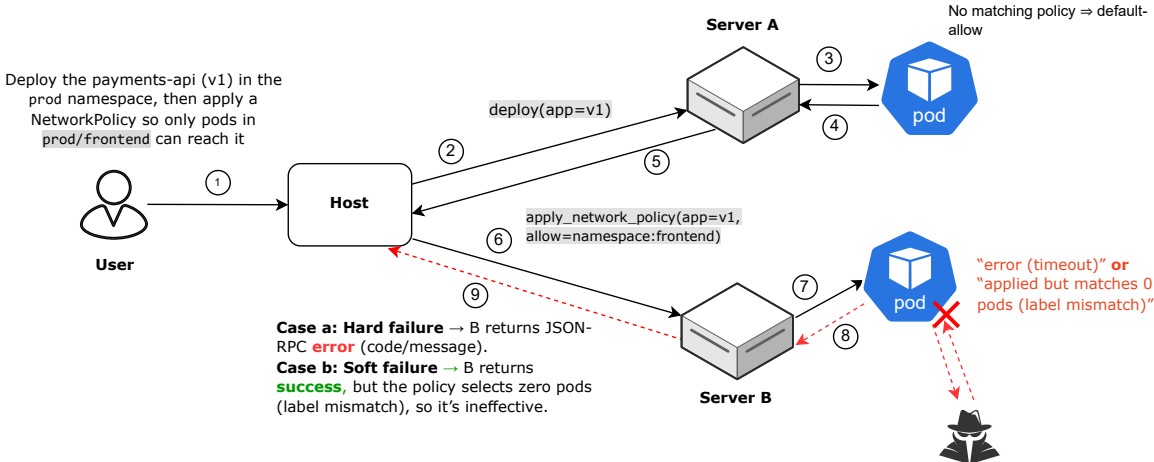

Figure 12: This is a transactional-integrity gap / atomicity failure - a race-window (TOCTOU-style) policy gap. Because deploy (Server A) and policy apply (Server B) aren't atomic, a hard/soft failure or label-mismatch leaves the pod with default-allow for a window, which an attacker can exploit to reach it (over-privilege window).

Even when individual servers are correctly configured, **Transactional Integrity Gaps** expose workflows to partial failure because Streamable HTTP MCP transports send JSON-RPC messages as separate HTTP

requests, and JSON-RPC structures each operation as an individual request, notification, or response. MCP defines lifecycle, capability negotiation, and session handling, but it does not define an atomic transaction primitive for multi-call workflows ~~Knostic Team (2025)~~ Model Context Protocol (2025s); Morley (2010). Recent work on trustworthy autonomous agents similarly argues that transactional execution, rollback, and conflict resolution are needed to preserve consistency across multi-step agent workflows Li et al. (2025a). This limitation pushes rollback and recovery to higher layers. ~~For example, Network Intelligence 2026 places transaction rollback for suspicious sequences of tool calls at the workflow layer, while Chang & Geng (2025) shows that the absence of robust transaction-like guarantees in current LLM planning frameworks leads to inconsistent states and partial failures that must later be repaired.~~ For example, recent work on trustworthy autonomous agents places transactional execution, rollback, and conflict resolution above the base tool-call interface Li et al. (2025a), while Chang & Geng (2025) shows that the absence of robust transaction-like guarantees in current LLM planning frameworks leads to inconsistent states and partial failures that must later be repaired. In MCP-based enterprise automations that span multiple services, a single misaligned or aborted run can therefore silently corrupt records or apply access-control changes in one subsystem but not its peers. The result is an integrity or policy violation that may only be discovered after the workflow has completed, as illustrated in Figure 12. Compounding these runtime gaps, **Patch Gaps and Vulnerable Instances** allow MCP servers that reuse reference implementations, connectors, and community-maintained packages to quietly retain exploitable code across registries and internal deployments when centralized patching is absent. ~~Recent reporting by Park (2025) and Lyons (2025) describes a SQL injection flaw in Anthropic's SQLite MCP server that was forked more than 5,000 times before the repository was archived, with no official patch planned and each fork left to its own maintainer for auditing and remediation. As a result, a single vulnerable MCP server can leave thousands of agents and workflows exposed to persistent remote code execution and data exfiltration even after the original project has been deprecated.~~ Recent security reporting by Park (2025) and Lyons (2025) describes a SQL injection flaw in Anthropic's SQLite MCP server that was forked more than 5,000 times before the repository was archived, with no official patch planned and downstream maintainers left to audit and remediate their own copies. This example illustrates how vulnerabilities in widely reused MCP servers can persist across downstream forks when patch responsibility is decentralized. As a result, a single vulnerable MCP server can leave thousands of agents and workflows exposed to persistent remote code execution and data exfiltration even after the original project has been deprecated. These runtime weaknesses become especially consequential when the same infrastructure is shared across users or agents. In multi-tenant deployments, **Session Isolation in Multi-Agent Environments** breaks down when shared MCP servers implement sessions with weak boundaries, so an attacker who can replay, guess, or intercept a session identifier can cause the server to mix contexts between agents and reuse another user's memory snapshot. MCP security best practices explicitly warn about such session hijacking scenarios, noting that session identifiers may surface in URLs or logs and that misconfigured session stores can leak data between users and tenants, so session state must be treated as sensitive authentication material Model Context Protocol (2025j). A compromised session can therefore expose private tools, prompts, and memories from one agent to another, leading to cross-tenant disclosure of chat histories, credentials, and retrieved documents across every workflow that shares the affected server.

Beyond isolation failures, operational breakdowns can also emerge in the way MCP systems respond to errors, expose internal state, and retire outdated capabilities. **Unsafe Fallbacks and Error Handling** arise when MCP tools fail and servers return structured error payloads that the host injects back into the model context. In such cases, agents may interpret these error messages as instructions and follow fallback paths that retry operations, call alternative tools, or surface debug information. ~~Recent guidance on MCP error handling documents responses that embed recovery instructions and detailed diagnostic fields, warning that poorly designed handlers can expose internal service details to both the model and end users Barthelet (2025).~~ Practitioner implementation guidance on MCP error handling documents `tools/call` responses that embed recovery instructions and diagnostic fields, while MCP error-handling guidance recommends logging detailed errors server-side and returning sanitized client-visible messages to avoid exposing internal service details to the model or end users Barthelet (2025); Hora (2025). ~~Consequently, verbose error responses and improperly handled debug outputs can leak internal URLs, stack traces, and credentials. Attackers who are able to trigger tool failures can exploit these disclosures to exfiltrate data and map internal topology across agents and workflows that share the affected server Hora (2025).~~ Consequently, verbose error responses and

improperly handled debug outputs may expose internal URLs, stack traces, credentials, tokens, or authentication details. Such disclosures can reveal system internals, help attackers map internal topology, or steer agents toward unsafe recovery behavior.

Across all of these failures, **Operational Visibility and Logging Gaps** make detection harder because effective forensics in MCP deployments requires uniform, high-granularity logs of tool invocations and context updates, yet real-world servers often implement logging inconsistently or omit it entirely. ~~Ecosystem scans report that the vast majority of MCP implementations lack sufficient telemetry to audit what users and agents are doing, leaving operators with substantial blind spots for abuse Sakib (2025).~~ Practitioner observability guidance argues that MCP deployments require method-level logging, response-size monitoring, user/session attribution, and alerting over JSON-RPC tool calls to make agent behavior auditable Sakib (2025).

~~Similarly, Narajala & Habler (2025) frame this weakness as *Insufficient Auditability*, defining it as inadequate logging that restricts detection and investigation of security events, while case studies from enterprise deployments argue that traditional monitoring lacks the context and granularity required for MCP interactions Ithena (2025).~~ Similarly, Narajala & Habler (2025) frame this weakness as *Insufficient Auditability*, defining it as inadequate logging that restricts detection and investigation of security events. Enterprise practitioner guidance further emphasizes MCP-specific audit trails that record agent identity, tool invocations, accessed data, and approval context for compliance-oriented review Ithena (2025). As a result, these logging gaps leave operators unable to reliably reconstruct agent behavior or prove what tools ran and what data was accessed across users, workflows, and tenants, turning MCP incidents into opaque failures rather than auditable security events. The same operational weakness also affects the retirement of obsolete capabilities. **Incomplete Tool De-Registration** allows outdated MCP tools or endpoints that are not properly unregistered during shutdown to continue appearing as available capabilities, so agents and orchestration layers may still discover and invoke them after operators believe they have been retired. ~~Research on MCP-aware tool synchronization describes pipelines that treat server tool lists as the single source of truth and include an explicit Remove outdated tools step that deletes entries no longer present on the server, while production MCP server APIs and management CLIs expose operations to remove MCP tool configurations from persistent stores in deployed environments Lumer et al. (2025); Lumer et al. (2025); Lumer et al. (2025); Lumer et al. (2025).~~ MCP-aware synchronization work treats server tool lists as operational state that must be kept current, including explicit removal of outdated tools no longer present on the server Lumer et al. (2025). Practitioner tooling and production API documentation further show that deployed environments expose mechanisms for inspecting, listing, or removing MCP tool configurations from persistent stores OpenSearch Project (2025); Akın (2025); TrustGraph (2025). If these de-registration steps are skipped or fail, stale tool records can linger in indexes and configuration groups, allowing malicious actors to impersonate or re-register those identifiers and route calls through a malicious endpoint across any host or tenant that still trusts the outdated entry Lumer et al. (2025); TrustGraph (2025). Detailed definitions of each threat are provided in Appendix A.8.

## 3.9 Emergent Threats from Model Behavior and Human Factors

Sections 3.1–3.8 described threats that look static, as if an MCP deployment could be frozen at a single moment and someone could point to the misconfigured tool, missing check, or fragile boundary. Once MCP enabled assistants move into real workflows the picture becomes more fluid. Models respond to long histories of prompts and context rather than a single request, operators adjust safeguards under pressure to ship features, and new agents appear over time to automate tasks that once required direct oversight. Small choices and isolated lapses begin to interact, so that the deployment drifts away from its original design, with sensitive information moving through unexpected channels and effective privileges shifting across tools and agents. This section follows that drift and focuses on emergent risks that grow from model behavior and human factors over time, setting up the privacy, insider, and coordination threats that the rest of this section examines. Detailed definitions of each threat are provided in Appendix A.9 below.

**LLM-Centric Privacy Threats.** Language models behind MCP agents create privacy risks, especially in deployments where many agents and tools share the same model backend, so probing a single agent

can expose information that originated elsewhere Anthropic (2025a; 2024a). In MCP deployments, this surfaces through three main threats: model inversion, membership inference, and pre-MCP data poisoning or backdoored models.

- **Model Inversion Attacks.** Because multiple MCP agents often call into the same model, an attacker can use one agent interface to craft queries that reconstruct sensitive training data or internal examples the model has memorized. Empirical work shows that carefully designed black-box queries can recover private properties or approximate original training records Fredrikson et al. (2015), so a single malicious workflow can leak proprietary data that other tenants and applications may rely on.

- **Membership Inference.** In deployments where MCP agents expose a shared enterprise LLM trained on internal data, an attacker can probe an agent to determine whether particular records, users, or documents contributed to the training corpus. Prior work demonstrates that black-box membership inference can reliably distinguish training points from non-members Shokri et al. (2017), allowing attackers to infer who or what the organization trained its shared model on without ever compromising MCP servers.

- **Pre-MCP Data Poisoning and Backdoored Models.** MCP implementations typically implicitly treat the underlying model as a trusted component, so any covert behavior introduced during pre-training, fine-tuning, or instruction tuning is inherited by every agent that calls it. Studies of data poisoning and instruction-tuned backdoors show that attackers can implant behaviors that only trigger on specific prompts or contexts Anthropic (2024b); Xu et al. (2023), meaning ordinary MCP conversations or tool outputs can activate leaks, override safety instructions, or bias tool selection across all workflows that reuse the compromised model.

**Insider and Human-Driven Threats.** Insider and human-driven threats arise when people who operate MCP agents and servers weaken security controls through their configuration choices or interactions.

- **Malicious Agent Operators.** Because MCP hosts translate model output directly into tool invocations, operators who wire unvetted servers into agents or grant overly broad permissions effectively expand the agent's execution authority beyond intended policy Li & Gao (2025). Analyses of MCP ecosystems show that hosts can rely on tool metadata and model output without strong secondary validation, so a malicious or careless operator may cause routine-looking workflows to perform high-privilege actions or tamper with data across the agent ecosystem Guo et al. (2025b).

- **Insider Threats and Shadow MCP Servers.** Community guidance and ecosystem studies describe how ungoverned or unauthorized MCP servers can be introduced inside organizations without central oversight, creating "Shadow MCP" instances that access internal resources while bypassing standard monitoring and governance OWASP Foundation (2025b). When employees deploy such servers on organizational infrastructure or register them with agents outside formal review, they can inadvertently expose proprietary data and sensitive operations to abuse, and these paths are difficult for security and compliance teams to track or remediate Li & Gao (2025).

- **Agent Interface Social Engineering.** Prior work on MCP security treats the agent interface and host as an attack surface where adversaries can issue crafted prompts or interaction sequences that steer tool selection and execution Guo et al. (2025b). These findings imply that attackers who interact with agents can present seemingly legitimate requests or workflows that drive high-impact tool invocations and data exfiltration under the guise of normal usage, and similar interaction patterns can nudge human operators into approving unsafe actions.

**Multi-Agent Coordination and Collusion.** As MCP deployments shift from single assistants to graphs of cooperating agents, coordination itself becomes a security-critical surface. Multi-agent topologies can develop hidden dependencies and emergent behaviors, so a single compromised or misconfigured agent can influence others, bypass local controls, or silently expand system-wide capabilities. In this setting we focus

on three coordination risks in MCP-based multi-agent systems: agent collusion, unverified agent handoffs, and orchestration drift.

- **Agent Collusion.** Two or more compromised agents may collaborate to bypass access controls or exfiltrate data stealthily, including via covert channels or steganography in inter-agent messages Motwani et al. (2024). MCP's support for nested, server-initiated agentic behaviors (*Sampling*) increases the surface for such interactions if oversight is weak Model Context Protocol (2025n).

- **Unverified Agent Handoffs.** When one agent (or MCP server) delegates to another without verifying identity and scopes, it can enable privilege escalation or "confused-deputy" failures; the MCP *Authorization* spec explicitly requires token audience binding and forbids token passthrough to prevent these issues Model Context Protocol (2025l). Anthropic's multi-agent engineering note likewise emphasizes careful handoffs and coordination boundaries Anthropic (2025b).

- **Orchestration Drift.** Over time, loosely coupled MCP workflows can diverge, accumulate "shadow logic," or re-enable deprecated behaviors (e.g., tools coming and going) unless guarded by policy and observability. The MCP Tools spec includes change notifications and security considerations, while Anthropic reports emergent coordination issues and drift risks in production multi-agent systems Model Context Protocol (2025q); Anthropic (2025b).

The risks described in Section 3 show that MCP deployments face a coupled threat surface spanning identity, transport, tools, data flows, and multi-agent behavior, with many threats reinforcing and amplifying one another. Weak authentication or naming collisions can enable poisoned tools, while poor container or dependency hygiene can turn routine parsing into remote code execution. Misaligned lifecycle management or configuration drift further expands the blast radius of a compromise. As a result, MCP operators cannot rely on static policies or one-time audits. Instead, they require continuous, telemetry-driven defenses that track who calls which tool, on what data, and with what downstream effects. The next section moves from taxonomy to countermeasure design, mapping these risks to a defense-in-depth control plane built on secure gateways, zero-trust principles, hardened execution, and continuous monitoring.

## 4  Mitigation Strategies and Security Controls for MCP

In this section, we translate the threat landscape for MCP deployments into a defense-in-depth control plane. Figure 13 illustrates the overall architecture, where six control families are layered outward from the protected core of MCP tool invocation to enforce security boundaries at multiple levels. Concretely, we group countermeasures into six complementary control families: **Secure Gateway / Firewall, Zero-Trust Architecture, Containerization & Sandboxing, Cryptographic Signing & Registry, Fine-Grained Authorization & Policy, and Monitoring & Logging**, with the aim of reducing blast radius, ensuring verified execution, and making violations observable and attributable. This control plane aligns with established security frameworks: Zero Trust's identity and context-sensitive access enforcement (NIST SP 800-207); SLSA supply chain integrity objectives with verifiable provenance; operational guardrails that reflect the OWASP LLM Top 10 classes; and governance/assurance practices consistent with the NIST AI Risk Management Framework Rose et al. (2020); Tabassi (2023b); SLSA Community (2025); OWASP Foundation (2025d).

The ordering of these families follows the typical flow of an MCP request: (i) network-level gateways constrain external traffic; (ii) identity is continually attested via zero-trust principles; (iii) execution units are isolated and their dependencies pinned; (iv) all artifacts and tool interfaces are cryptographically verifiable; (v) runtime privileges are enforced at the narrowest feasible scope; and (vi) telemetry feeds a feedback loop for anomaly detection and policy refinement. Together, these layers form a coherent scaffold in which no single control is sufficient, but in aggregate they raise the bar for exploitation, limit blast radius, and provide the observability needed for rapid incident response.

For each control family that follows, we will specify: (i) security objectives and threat coverage; (ii) reference architecture patterns suitable for MCP (e.g., identity-aware proxies and default-deny egress for gateways;

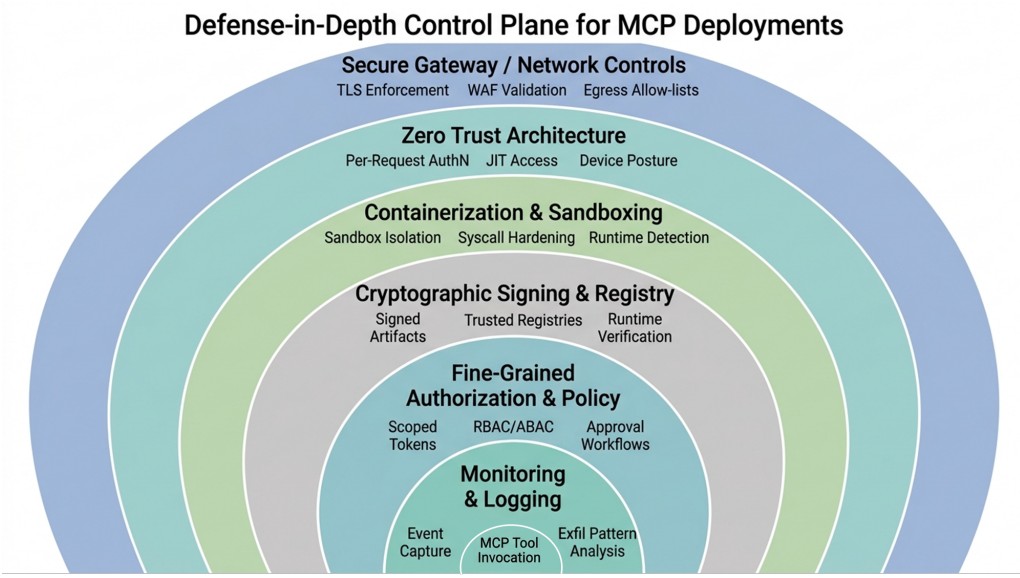

Figure 13: Six control families form a defense-in-depth architecture for MCP deployments. The outer layers enforce network controls, followed by Zero Trust, containerization and sandboxing, cryptographic signing and registry protections, fine-grained authorization and policy, and monitoring and logging. Together, these controls help contain, detect, and limit security failures.

per-request authentication/authorization for Zero Trust; per-tool isolation for sandboxing); (iii) verifiable controls and how to check them (e.g., signed artifacts with SLSA-conformant provenance and Sigstore/Cosign verification before deployment); and (iv) operational signals and metrics for continuous monitoring (structured logs, policy decision telemetry, anomaly thresholds) to support detection, response, and audit. This structure ensures each mitigation is both principled and testable, bridging high-level policy with concrete enforcement and measurable assurance.

The remainder of this section is as follows: Subsections 4.1–4.6 each develop a control family, stating objectives and MCP-specific patterns. Table 7 links these families to the threat categories defined in Sections 3.1–3.9 of the taxonomy and indicates whether the coverage is primary or indirect, with pointers to concrete checks. Finally, Tables 8 and 9 list the verifiable controls and operational signals for each control family.

## 4.1 Secure Gateways and Network Controls

In MCP deployments, agents invoke server-hosted tools over structured RPC and those calls often produce side effects on the filesystem, the network, or third-party APIs. The MCP specifications place the burden for safety at the boundary, which means that servers are expected to validate inputs, enforce access controls, rate limit requests, and sanitize outputs Model Context Protocol (2025k). Authorization guidance further cautions against token passthrough and emphasizes careful separation of token audiences to prevent confused deputy failures Model Context Protocol (2025o). When these requirements are considered alongside well documented prompt injection and insecure output handling risks in LLM applications, transport security, identity assurance, and strict request validation emerge as central security concerns for MCP traffic. Within this setting, secure gateways and network controls function as primary policy enforcement points for MCP. **Identity aware proxies and gateways** form the outer security boundary by terminating TLS, enforcing encrypted connections to upstream services, applying strong client authentication, and generating audit quality logs. These transport practices align with MCP guidance, which recommends using TLS for all network transports, enforcing authentication at the transport layer, and validating HTTP origins for remote connections Model Context Protocol (2025r). Recent MCP security work instead recommends gateway-based governance that combines per-user authentication, scoped authorization, inline policy enforcement, and end-to-end auditability before requests reach MCP servers Errico et al. (2025). By ensuring that inbound

Table 7: Mapping MCP risk categories to mitigation mechanisms across six defense layers. A checkmark indicates a direct control, "indirect" denotes partial or supporting coverage, and "no direct" indicates minimal or no native mitigation.

| Risk Category | Secure Gateway /Network Controls | Zero Trust Principles | Containeriz-ation & Isolation, and Runtime Defenses | Cryptographic Signing & Registry | Fine-Grained Auth & Policy | Monitoring & Logging |
|---|---|---|---|---|---|---|
| Identity / Trust | ✓namespace allow-lists | ✓continuous identity checks | indirect | ✓verify authenticity | ✓per-server trust zones | indirect: detect impersonation |
| Supply Chain & Code Integrity | ✓inspect downloads | indirect: verify components | ✓isolate install env | ✓prevent tampered packages | indirect: policy gates | indirect: anomaly detection |
| Prompt / Input Injection | ✓block malicious payloads | ✓validate inputs/outputs | indirect | indirect: trusts prompt sources | ✓block high-risk actions | ✓alert prompt attacks |
| Malicious Tools | ✓monitor tool API calls | ✓no implicit trust | ✓restrict tool actions | ✓verify tool code | ✓allow/deny lists | ✓detect tool misuse |
| Credential Leakage | ✓prevent outbound key leaks | ✓least-privilege, short tokens | no direct | no direct | ✓scoped API keys | ✓alert on key use |
| Authorization (authz) Gaps | ✓policy on traffic | ✓JIT access | no direct | no direct | ✓RBAC, approvals | ✓audit trails |
| Exec / Sandbox Risks | indirect: segmentation limits blast radius | ✓micro-segmentation | ✓prevent OS escape | no direct | no direct | ✓detect sandbox escapes |
| Workflow / Config Risks | ✓uniform network policies | ✓config posture checks | indirect: immutable images | ✓block unvetted versions | ✓central policy | ✓flag configuration drift |
| LLM Privacy Attacks | ✓DLP scan on responses | ✓least-privilege data paths | no direct | no direct | indirect: limit data exposure | ✓alert on PII/inversion |
| Insider Threats | indirect: logs constrain misuse | ✓restrict insider privileges | indirect: limit insider reach | no direct | ✓curb insider abuse | ✓correlate insider actions |
| Multi-Agent | indirect: agent allow-lists | ✓agent auth, context | indirect: isolate agents | no direct | ✓role isolation | ✓flag rogue agents |

requests are authenticated, policy-checked, and recorded at the gateway boundary, these controls reduce the risk of interception, impersonation, and unauthorized access.

Building on that perimeter, **Zero Trust segmentation** constrains which services and networks are permitted to communicate with MCP servers. Rather than allowing broad internal connectivity, segmentation limits lateral movement by isolating MCP components into tightly scoped network zones. Each server and its supporting services should be reachable only from explicitly approved peers, and egress paths should be restricted to required destinations. NIST recommends dividing internal networks into small isolated segments so that traffic entering and leaving each segment can be monitored and controlled Chandramouli (2022). In cloud and Kubernetes environments, operators can further enforce microsegmentation through service mesh controls that apply workload identity and encryption within clusters istio.io (2025); Chandramouli & Butcher (2022). This layered containment makes it significantly more difficult for an attacker to pivot from one compromised MCP component to another. Even within segmented environments, internal traffic must not be implicitly trusted. **Service mesh mutual TLS** extends identity bound encryption to service to service communication inside clusters istio.io (2025); Chandramouli & Butcher (2022). By authenticating workloads and encrypting their exchanges, service mesh controls ensure that internal MCP calls remain confidential and tied to specific identities rather than to network location alone.

Security hardening must also extend to the hosts themselves. **Host based firewalls and operating system hardening** reinforce network level protections by adopting a strict deny by default posture and minimizing externally exposed services Red Hat, Inc. (2025). Additional operating system level guidance recommends using minimal and hardened installations, enabling kernel level access control mechanisms such

as SELinux or AppArmor, disabling unnecessary background services and open ports, and prohibiting direct root logins in enterprise production environments Narajala & Habler (2025). These measures reduce the likelihood that general system vulnerabilities become indirect entry points into MCP workflows.

Finally, security controls must operate at the protocol boundary where individual JSON RPC calls are interpreted. **Application layer gateways and validation** provide this last line of defense by inspecting every request before it reaches tool handlers. Requests should be validated strictly against the MCP schema and rejected if they are malformed, oversized, or out of specification CSF Tools (2025). Additional rules aligned with OWASP input validation guidance can detect suspicious parameter values or invocation patterns that resemble tool poisoning or injection attempts OWASP Foundation (2025c). Granular rate limiting and anti automation controls should also be enforced on all MCP endpoints to reduce abuse and denial of service risks OWASP Foundation (2019; 2023a). Together, these layered controls ensure that MCP requests are encrypted, identity bound, segmented, hardened at the host level, and rigorously validated before reaching sensitive tool execution logic.

## 4.2 Zero Trust Principles

Many of the MCP-specific failures discussed above share a common root. Systems often assume that a component becomes safe once it has been authenticated or placed on a trusted network. Zero Trust Architecture rejects this assumption by refusing to grant implicit trust on the basis of location or prior approval and instead requiring continuous verification of each identity and each action Gambo & Almulhem (2025). In an MCP deployment, this means treating every tool invocation and every data access as a potentially risky operation and reevaluating on each call who is making the request, under what delegation, and with what minimum permissions. When applied consistently, this model directly counters several risks introduced by autonomous agents. Per request verification reduces reliance on long lived bearer tokens, narrowly scoped credentials limit the damage caused by stolen secrets, and ongoing policy enforcement at gateways or authorization services can block harmful tool calls even when the model has already been influenced by prompt hijacking or insider abuse. ~~**Per-request identity verification** provides the starting point for operationalizing Zero Trust in MCP because every agent call must be treated as a fresh authorization decision rather than an action that inherits trust from an earlier session Ramnarayanan (2025).~~ **Per-request identity verification** provides the starting point for operationalizing Zero Trust in MCP because every agent call should be treated as a fresh authorization decision rather than an action that inherits trust from an earlier session. This aligns with Zero Trust architecture principles that require continuous evaluation of subject identity, device state, policy, and request context before granting access Rose et al. (2020).

Each request to an MCP server should therefore be authenticated and authorized independently so that trust is established at the moment of action rather than assumed from prior interaction. In practice, this means that the client presents a user-scoped or service-scoped access token for every request and the MCP proxy or server validates security-critical token properties such as signature, issuer, audience, scope, expiration, and resource binding before granting access. This requirement is reinforced by recent empirical work on real-world remote MCP servers, which finds widespread authentication weaknesses in OAuth-enabled MCP deployments and motivates hardened token validation and authorization checks for MCP clients and servers Zhou et al. (2026a). Requests that lack credentials, carry expired or altered tokens, or violate audience or scope constraints should be denied immediately. This approach prevents stale sessions and long lived bearer tokens from quietly accumulating privilege over time and ensures that access decisions remain tied to the current security context rather than to an outdated trust state. Once identity has been verified on every call, **just-in-time least-privilege access** becomes essential for ensuring that authenticated agents do not retain more authority than they need. This need is reinforced by large-scale empirical work on MCP privilege management, which finds that MCP applications frequently exercise security-relevant resource accesses and argues for dynamic permission models and automated trust assessment to limit overbroad authority Li et al. (2025b). Zero Trust in MCP requires authority to remain minimal and short lived rather than broad and persistent. Tools and data should be accessible only when they are needed and only with the narrowest rights necessary for the current step. An agent may, for example, receive a temporary credential that permits a single database query or a bounded read operation, and that authority should disappear as soon as the task is complete. This design reduces the dangers associated with persistent credentials by limiting access to

individual actions rather than entire sessions. As a result, even when credentials leak, an attacker has very little time and very little scope in which to misuse them. Short lived and narrowly scoped access therefore turns credential exposure from a system wide failure into a much more contained event.

These protections become more effective when they are reinforced by **device and posture checks**. Zero Trust for MCP must evaluate not only who is making a request and what authority accompanies it, but also whether the machine running the agent remains trustworthy enough to participate in the workflow. A remote host should not be accepted simply because it belongs to an internal environment or because it was previously allowed to connect. Before access is granted, the host should demonstrate compliance with a defined security baseline. An MCP gateway can enforce checks on operating system patch status, the presence of endpoint protection, and whether approved agent software is installed. Clients that fail these checks can be denied access or placed in a restricted state so that outdated or compromised machines do not become hidden entry points into MCP tools. This posture aware approach follows Zero Trust guidance that emphasizes continuous monitoring of asset health and removal of access when a device falls out of compliance OWASP Foundation (2021).

Taken together, these controls show why Zero Trust provides a natural foundation for securing MCP systems. Recent MCP-specific research reflects this direction directly, arguing that MCP deployments benefit from continuous verification, least-privilege authorization, and policy enforcement across tools, agents, and servers Ando (2026). This direction also aligns with broader work on Zero Trust for agentic AI systems, which frames autonomous agents as security-relevant workloads requiring explicit identity, context-aware access control, and continuous monitoring Huang et al. (2025); Liu et al. (2025a). Industry practice is moving in the same direction: major providers such as Microsoft and Google increasingly describe AI agents as first-class enterprise identities or workloads that should be governed through authentication, authorization, and policy enforcement rather than implicit trust Microsoft (2025a); Santiago Díaz (2025). Within MCP, this perspective is especially valuable because security decisions are revalidated as workflows evolve, components change, and agents act with greater autonomy.

### 4.3 Containerization, Isolation, and Runtime Defenses

MCP and multi agent deployments should assume that components can be subverted in practice, including compromised tools, unvetted images with supply chain implants, and crafted inputs that steer execution into unsafe behavior. In this setting, containerization and isolation mechanisms aim to minimize blast radius, enable rapid containment and revocation, and preserve high fidelity telemetry for detection and attribution. These objectives are achieved by confining tool and agent workloads to narrowly scoped execution domains with strictly limited ambient privileges across network, filesystem, system calls, identities, and secrets, while simultaneously instrumenting runtime monitoring so that deviations are detected quickly and attributed to concrete principals. To make these mechanisms operational in MCP deployments, the controls described below are structured as a defense-in-depth progression from **prevent** (hard isolation, least privilege) to **contain** (sandbox boundaries) to **detect and respond** (runtime anomaly signals). Implemented together, controls along this progression reduce lateral movement and limit secret exfiltration even under partial compromise, while preserving the operational context needed for swift remediation.

Effective protection begins with **sandbox isolation**, which confines faults to a single execution domain by ensuring that each tool or model runs inside its own container with a deliberately constrained runtime surface. Containers should execute as non root users, unnecessary Linux capabilities should be removed, and the root filesystem should be mounted read only so that privilege seeking behaviors are curtailed and routine writes are prevented Kubernetes (2025b). Because outbound reachability often enables escalation and exfiltration, egress should be restricted to an explicit allowlist so that containers communicate only with endpoints required for their task rather than reaching public networks by default Kubernetes (2024a). Persistence should also be minimized by keeping containers stateless and short lived so that footholds and failed exfiltration attempts disappear with the container lifecycle. To further address supply chain risk, images should be admitted only when they are signed and hash verified prior to deployment Souppaya et al. (2017). These controls ensure that even when a tool is compromised, its operational domain remains tightly bounded. Building on this foundation, **agent confinement** narrows the scope within which the agent runtime itself can operate so that model or tool driven actions remain contained. The agent should run in its own pod with

Table 8: MCP controls: verifiable checks and runtime signals (part 1).

| Control Family | Verifiable controls & how to check | Operational signals & metrics |
| --- | --- | --- |
| **Secure Gateway / Firewall** | • *TLS enforcement:* Only strong proto/ciphers; verify via handshake tests; reject downgrades.
• *Identity-aware proxy:* Missing/expired/wrong-audience tokens → `401/403`.
• *Upstream (m)TLS:* Remove client cert → handshake fails.
• *Protocol-aware validation:* Malformed/oversized JSON-RPC → blocked (fuzz to test).
• *Origin validation (HTTP):* Spoofed `Origin` → denied.
• *Default-deny egress:* Non-allowlisted host/port → blocked. | • Allow/deny counts by reason (unauth, schema, policy).
• WAF rule hits (injection patterns, oversize payloads).
• TLS handshake failures/downgrades; cert errors.
• Blocked egress attempts; new external domains/day.
• % requests with complete edge audit (trace/span IDs). |
| **Zero-Trust Architecture** | • *Per-request AuthN/Z:* Expired/replayed token → denied.
• *Least-privilege, short-lived creds:* Inspect scopes/TTL; out-of-scope action → denied.
• *Continuous verification:* Flip device to non-compliant → blocked/quarantined.
• *Revocation/re-auth:* Revoke token; subsequent calls fail until re-auth. | • AuthZ decision logs (who/what/why; policy path).
• Token TTL distribution; expired/invalid reject rate.
• Posture check failures; deny-rate by resource.
• Elevation prompts/approvals per period (spikes flagged). |
| **Containerization & Sandboxing** | • *Isolated execution:* Non-root, `readOnlyRootFilesystem`, no privilege escalation.
• *Syscall filtering:* Seccomp/AppArmor/SELinux; disallowed syscall → blocked/logged.
• *Network isolation:* Default-deny; off-allowlist egress → blocked.
• *Trusted images:* Cosign/Sigstore verify on pull/start; unsigned → fail closed.
• *Ephemeral secrets:* Scan for hard-coded secrets; reuse JIT secret → fails. | • Seccomp/AppArmor violation counts; HIDS/Falco hits.
• Restart/crash loops from policy kills; throttling events.
• Blocked egress attempts; inter-namespace anomalies.
• % running images with verified signatures/provenance. |

a dedicated service account and the smallest RBAC footprint required for orchestration, which reduces the likelihood that credentials are reused across components Kubernetes (2024b). Host namespace features such as HOSTPID=FALSE and HOSTNETWORK=FALSE should remain disabled, and HOSTPATH mounts should be avoided to limit host exposure Kubernetes (2022; 2025a). Only the minimal volumes and sockets required for MCP input and output should be mounted, and writable paths should be marked NOEXEC to discourage code staging. On Kubernetes, this configuration aligns with Pod Security settings that require RUNASNONROOT, enforce ALLOWPRIVILEGEESCALATION=FALSE, apply a minimal SECCOMPPROFILE such as RUNTIMEDE-FAULT or a tailored alternative, and mandate AppArmor or SELinux policies Kubernetes (2022). Network policies can further restrict the agent to approved MCP servers and internal control endpoints while blocking access to metadata services and public egress by default. In combination, these measures ensure that even if the model is steered toward unsafe actions, the operational impact remains constrained.

While sandbox isolation and agent confinement restrict process scope, **system call hardening** further narrows the kernel interface available to compromised workloads. A deny-by-default system-call policy can be enforced with SECCOMP or strengthened through sandboxed container runtimes such as gVisor, admitting only the minimal set of calls required by the workload. Prior work on Linux syscall filtering shows that automatically generated seccomp policies can block large portions of the syscall surface and prevent exploit

paths that depend on unused system calls, while container-security guidance recommends limiting container privileges and kernel interfaces as part of defense in depth Canella et al. (2020); Wan et al. (2017); Souppaya et al. (2017). In practice this often begins with the RUNTIMEDEFAULT profile and proceeds through iterative refinement, first observing violations in audit mode and then enforcing restrictions once stable. High risk operations such as `ptrace`, `mount`, `keyctl`, `bpf`, and `perf_event_open` should be blocked for typical MCP tools Docker (2025); SUSE (2025). To reduce residual exposure, SECCOMP can be combined with AppArmor or SELinux type enforcement to constrain file access and process capabilities, and `no_new_privs` can be set so processes cannot gain additional privilege through execution or setuid transitions Linux Kernel Project (2012). These controls significantly reduce the kernel level attack surface even when arbitrary code executes.

Even with strong preventive controls in place, runtime behavior must still be monitored, which makes **runtime detection** an essential component of this defense model. Host based sensors or eBPF backed tools such as Falco can observe system calls, file activity, and network flows and surface policy violating behaviors including unexpected process launches, unauthorized writes in system directories, or connections to non allowlisted endpoints Sysdig (2023b). NIST recommends maintaining a runtime threat profile for each container that tracks processes, network activity, and filesystem changes so deviations are surfaced quickly Souppaya et al. (2017). In Kubernetes environments, these detections can trigger containment actions such as applying restrictive network policies, blocking access to cloud metadata endpoints, scaling affected workloads down, rotating or revoking exposed credentials, and redeploying clean instances, consistent with container-security guidance on runtime monitoring, least privilege, network isolation, and incident containment Souppaya et al. (2017); Sheriff et al. (2025). Through this continuous feedback loop, runtime monitoring reinforces isolation and capability reduction, ensuring that prevention, containment, and response operate as a coherent defense in depth system.

**Secret hygiene** represents a parallel control-plane concern because every host, client, and server in an MCP deployment carries a non-human identity, so a leaked token on any one component can rapidly expand into system-wide access. Practitioner scanning by GitGuardian reports that approximately five percent of public MCP server repositories exposed secrets, including API keys and bearer tokens in configuration files, examples, and container images Ferry (2025). To counter this sprawl, operators should adopt a secret broker pattern in which servers and tools obtain short lived, least privilege credentials from a vault at call time rather than relying on hard coded keys. Credentials should rotate automatically and be segmented across distinct wallets or accounts that enforce spending limits for high risk tools MCP Security Working Group (2025). The orchestration layer should treat secrets as out of band data so that prompts, tool descriptions, logs, and traces carry only opaque handles that MCP servers resolve through authenticated APIs, ensuring that credentials function as scoped and ephemeral capabilities rather than standing privileges MCP Security Working Group (2025). Beyond credential handling, security must also be enforced at the deployment layer, where platform level controls determine the effective permissions and isolation boundaries of agents and tools. At the platform level, **policy scoped deployment** ensures that agents and tools operate only within environments aligned with their declared permissions. This alignment begins with isolating each MCP component in its own pod and enforcing a strict security context so that execution privileges are narrowly bounded from the outset. In practice, this includes running workloads as non root users, dropping unnecessary Linux capabilities, applying an enforced seccomp profile rather than leaving it unconfined, and mounting the root filesystem as read only. These execution constraints must then be reinforced at the network boundary. NetworkPolicies should restrict connectivity so that pods can reach only the MCP servers, data stores, and control endpoints required for their roles Kubernetes (2024a). By narrowing ingress and egress paths in this way, the platform reduces the opportunity for lateral movement or unintended data exposure. Beyond execution and network isolation, access to orchestration and cloud resources must also reflect the same minimal scope. Fine grained RBAC should ensure that each component receives only the cloud and API permissions strictly necessary for its function rather than broad cluster wide roles Kubernetes (2024b). When these layers are applied together, the declared policy scope becomes enforceable in practice rather than merely descriptive. Finally, any activity that falls outside this declared scope, such as unexpected filesystem access or egress to non allowlisted destinations, should be surfaced through audit logs and treated as a policy violation that triggers automated containment. Together, these controls constrain runtime behavior and ensure that deviations from declared scope are observable and enforceable.

Table 9: MCP controls: verifiable checks and runtime signals (part 2).

| Control Family | Verifiable controls & how to check | Operational signals & metrics |
|---|---|---|
| **Cryptographic Signing & Registry** | • *Signed artifacts:* Enforce `cosign verify` in CI/load; unsigned → rejected.
• *Provenance checks:* SLSA attestations verified pre-deploy.
• *Server identity:* Valid/pinned TLS certs; expired/mismatch → refused.
• *Allow-listed tools:* Unknown tool ID/hash → rejected.
• *Change control:* Tool definition drift → block until approved. | • Sig/provenance failures; registry drift (hash/version).
• Cert expiry horizon; cert validation error rate.
• Registry access anomalies (bulk pulls, odd hours). |
| **Fine-Grained Authorization & Policy** | • *Minimal scopes:* Inspect tokens/consents; out-of-scope call → denied.
• *RBAC/ABAC tests:* Simulate unauthorized principal → deny.
• *Allow/deny lists:* Non-approved endpoint/domain → blocked.
• *User consent:* High-privilege tools require approval; no approval → cannot run.
• *No "god" tokens:* Audience-bound, service-specific; passthrough → denied. | • Deny/allow counts by rule/tool; surge flags.
• Scope-expansion attempts vs baseline.
• Policy drift detections; unauthorized edits.
• Consent prompts/grants per period (anomalies flagged). |
| **Monitoring & Logging** | • *Structured audit logs:* Immutable/WORM or hash-chained; field presence checks.
• *Distributed tracing:* Correlate edge→server→tool for a sample request.
• *Alerting pipeline tests:* Inject synthetic anomalies; ensure alerts fire.
• *Retention & time sync:* Enforce retention/NTP; simulate log tamper/clock skew → alert. | • Behavioral baselines/anomaly scores (per-tool freq, prompt/output size, fan-out).
• Security event volumes: auth failures, policy denials, WAF/HIDS alerts.
• MTTD/MTTR; alert acknowledgement/closure SLAs.
• Log/trace coverage: % requests with full trace; drop rate.
• Exfil indicators: unusual egress volume/destinations/entropy shifts. |

Finally, metadata sanitization is necessary because tool metadata becomes part of the model context and can therefore function as a channel for adversarial instructions. Recent MCP research shows that tool poisoning embeds malicious instructions in tool metadata such as names, descriptions, or schemas, allowing those instructions to enter the agent context during discovery or registration and steer later behavior even when the poisoned tool itself is not invoked Wang et al. (2025a); Li et al. (2026a); Huang et al. (2026b). Microsoft practitioner guidance similarly recommends reducing cross-domain trust, sanitizing tool metadata, and adding policy or human checks for indirect prompt injection in MCP deployments Young & Delimarsky (2025). This risk is not limited to MCP specific tooling. Research on instruction backdoors shows that natural language instructions can encode persistent triggers that reliably alter model behavior once introduced into context Xu et al. (2023). Taken together, these findings highlight that metadata itself must be treated as untrusted input. To reduce this attack surface, MCP hosts should enforce strict schemas and sanitization on tool metadata and prompts before they are exposed to the model. In practice, this includes constraining lengths, whitelisting allowed characters, stripping markup, and validating structured fields against defined schemas, with validation failures treated as hard errors OWASP (2023). While sanitization alone cannot eliminate prompt based and metadata based attacks, it establishes a defensive baseline in which all content entering the agent context is assumed untrusted and filtered as early as possible.

### 4.4 Cryptographic Signing & Registry

A core pillar of MCP security is ensuring the authenticity and integrity of communicating components, including tools, servers, and clients, through cryptographic verification. Without strong signing and validation mechanisms, MCP ecosystems remain vulnerable to tool poisoning, server spoofing, and broader supply chain compromise. This is especially important in MCP because agents routinely rely on externally provided tool definitions, manifests, and server metadata during ordinary workflows. As a result, trust cannot rest on names, endpoints, or transport alone, but must instead be grounded in verifiable provenance that allows clients and operators to confirm who produced an artifact and whether it has been altered since publication Microsoft (2025b).

This foundation begins with **cryptographic identities and digital signing of artifacts**. In a hardened MCP deployment, tools and servers do not appear as anonymous binaries but as components with verifiable cryptographic identities that clients can authenticate before accepting code or configuration Microsoft (2025b). By signing tool definitions, manifests, and server binaries with trusted keys, for example through X.509 certificates or Sigstore based attestations, providers make authenticity and integrity directly verifiable at the point of installation and update. Clients should validate these signatures as a mandatory admission step and treat any missing, invalid, or untrusted signature as a hard failure. Framed this way, signing is not merely a packaging convenience but a direct control against tool poisoning, server spoofing, and unauthorized deployment. Once artifact identity is established, **trusted tool registries** provide the operational mechanism for carrying that trust into routine discovery and deployment workflows. Tool discovery in MCP should therefore begin from a registry that the organization explicitly trusts rather than from arbitrary public catalogs. This registry should serve as the supply chain anchor for MCP servers and tools by recording expected versions, hashes, and cryptographic signatures, and by storing these records in tamper evident forms such as immutable ledgers or signed Merkle tree based transparency logs so that attempts to rewrite release history become detectable Sigstore (2022). Under this model, only registry listed artifacts should be treated as eligible for discovery or execution, sharply limiting the ability of unvetted tools to enter day to day workflows. Building on registry based control planes proposed in prior work, administrators should govern onboarding, enforce fine grained admission policies, and apply automated trust scoring so that squatted or malicious tools are identified and blocked before they are surfaced to agents Narajala et al. (2025).

These registry controls must be complemented by **server authentication** so that trust in artifacts is matched by trust in the endpoints that actually serve them. MCP hosts should treat every server connection as untrusted until it is authenticated and should rely on TLS certificates so that clients connect only to endpoints whose identities they can verify. On top of these transport level checks, server binaries, scripts, and container images should also be signed and verified before deployment or update in line with secure software development guidance Souppaya et al. (2022b). In service to service settings, hosts should further enforce mutual TLS with strict certificate validation or pinning and reject self signed or mismatched identities so that only authenticated MCP servers participate in the ecosystem. This matters because transport authentication confirms the endpoint being reached, whereas artifact signing confirms the integrity of the code and configuration that endpoint is actually running.

Trust must also persist after deployment, which makes **runtime signature verification** an essential continuation of these earlier checks. MCP clients should not assume that installation time verification is sufficient because a registered tool can later be modified while retaining the same name, endpoint, or apparent version. Whenever a tool definition changes, even if the version number does not, the client should treat the update as untrusted and require explicit revalidation or a fresh approval workflow before execution proceeds. This kind of continuous verification is particularly important against rug pull style attacks in which a provider silently alters a tool after users have already granted consent Bhatt et al. (2025). By reattaching trust to a specific signed state rather than to a static tool identity, runtime verification helps ensure that post approval drift does not quietly reenter trusted workflows.

### 4.5 Fine-Grained Auth & Policy

MCP implementations should adopt least privilege so that each tool operates only with the scopes or permissions it actually needs. Because MCP relies on OAuth like tokens to carry these permissions, those tokens

must remain tightly scoped and closely aligned with the intended function of each tool Lodderstedt et al. (2025). A mail reading tool that only needs read access to inbox messages, for example, should never receive credentials that also permit mailbox deletion. In practice, this means that authorization must be attached to the specific action being performed rather than granted broadly at the level of the overall agent or session. Role based and attribute based access control can help enforce these boundaries, while policy frameworks such as Open Policy Agent and AWS Verified Permissions make it possible to attach contextual rules to each tool or token OPA (2025); AWS (2025). The MCP authorization specification builds on OAuth 2.x and therefore allows implementers to encode permissions directly in scopes and JWT claims. Prior work further recommends including an explicit list of required OAuth scopes in each tool's signed definition so that the client requests exactly those scopes and the user approves them at consent time Bhatt et al. (2025). Together, these mechanisms turn least privilege from a general principle into an enforceable authorization boundary for day to day MCP operation.

**Scoped API tokens** provide the most direct way to enforce this boundary during execution. If a single bearer token can drive many MCP tools and servers, then one compromise is enough to unlock large parts of the deployment and enable lateral movement across workflows. To reduce this risk, the host should treat each tool invocation as a request for a narrowly defined capability rather than as an extension of a broad standing session. Before the call is issued, the authorization layer can mint an OAuth access token or API key whose audience is limited to one MCP server and whose scopes match only the specific operation that tool is permitted to perform, such as read only access to a constrained file namespace or a single portfolio quote endpoint. The server must then enforce this contract by validating audience and scope on every request and rejecting calls that exceed the policy attached to the tool. Applied consistently, this pattern replaces long lived multi purpose credentials with short lived fine grained tokens, which means that even if one token is exfiltrated it cannot be replayed to invoke other tools, reach unrelated servers, or silently expand agent privileges in downstream workflows Lodderstedt et al. (2025). This execution level scoping becomes stronger when it is backed by **RBAC and policy engines** that govern authorization as a dedicated control layer across the deployment. In MCP environments, principals such as users, agents, and tool servers should be associated with roles or attributes, and every request should be evaluated against explicit policy before any tool or resource is invoked Ferraiolo et al. (2001). Building on this model, a policy engine such as Open Policy Agent or Cedar can receive the request context and determine whether the invoking principal is permitted to perform the requested operation on the target resource Open Policy Agent Contributors (2025); Cutler et al. (2024). This makes access control declarative, auditable, and easier to evolve over time. It also allows the authorization layer to incorporate contextual signals such as time, source location, or recent activity, which is important because the same action may be acceptable in one context and unsafe in another. In this way, policy engines extend least privilege beyond static role assignment and support context aware authorization that remains aligned with runtime conditions.

The same principle also applies earlier in the workflow through **allow and deny lists**, which constrain what tools and servers can enter the trusted operating set in the first place. In stronger MCP deployments, tool and server selection should be treated as a curated admission decision rather than as an open-ended discovery process driven by whatever appears in public registries. This need is supported by ecosystem-level MCP security analysis showing that public registries can lack sufficient vetting, enabling server hijacking and sensitive-data exfiltration risks across hosts, registries, and servers Li & Gao (2025). A practical approach is to maintain a reviewed set of MCP servers, tools, and domains for each environment, pinned by origin, registry, and stable identifiers such as image digests, package hashes, or UUIDs, so that agents invoke only components that have already passed admission checks. When a new server or tool is proposed, it can pass through a lightweight intake process in which security and legal stakeholders review provenance, licensing, and data handling before it is added to the allowlist, while anything unvetted remains denied by default. At runtime, the host should reinforce this stance by preferring approved registries and rejecting discovery or fetch requests that target unapproved sources or suspicious look alike names. This reduces the chance that agents quietly bind to opportunistic or high risk endpoints during ordinary operation. Even after a tool or server has passed these admission checks, trust should not be treated as permanent or unconditional.

**User approval workflows** provide an additional safeguard by making privilege expansion visible at the point where new capabilities are about to be exercised. When an MCP host encounters a tool for the first

time or detects a new version, it should pause automatic use and present a clear approval request to the user or administrator. That request should explain what data the tool can read or write, which external systems it can contact, and what irreversible actions it can trigger, so that the approver can compare the requested privileges with the intended use. If the requested scope is broader than necessary, the workflow should highlight that mismatch and either constrain the permissions or escalate the decision to a higher trust tier. In high-assurance deployments, these approval flows align with agentic-AI research that treats human oversight as a structured control point with explicit intervention conditions, role resolution, interaction semantics, and communication channels, and with least-privilege frameworks for tool-calling agents that reconstruct permission hierarchies to reduce overbroad tool access Cheng & Cheng (2026); Zhu et al. (2025). By introducing a review checkpoint when capabilities change or expand, approval workflows help ensure that least privilege remains enforceable throughout operation rather than only at initial admission.

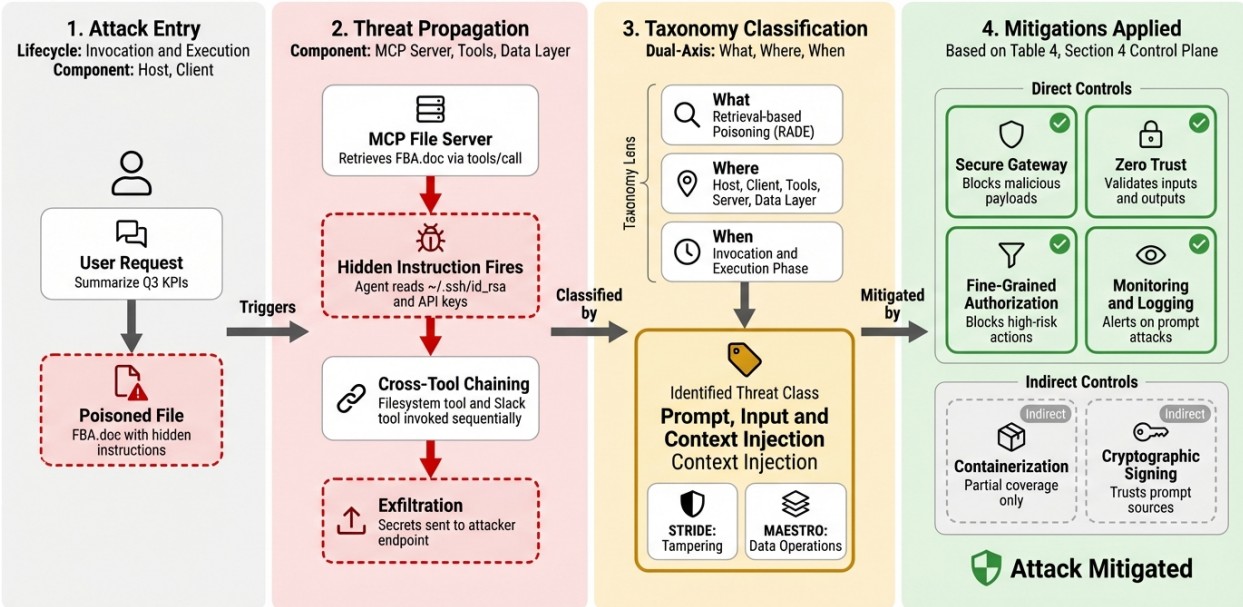

Figure 14: End-to-end threat lifecycle for the RADE attack, illustrating the dual-axis taxonomy in action. Stage 1 shows the attack entry via a poisoned file upload. Stage 2 traces propagation through the MCP file server, hidden instruction execution, cross-tool chaining, and credential exfiltration. Stage 3 applies the dual-axis threat taxonomy introduced in Section 3 to classify the attack by what (Retrieval-based Poisoning), where (Host, Client, Tools, Server, Data Layer), and when (Invocation and Execution phases), yielding the resulting threat class Prompt, Input, and Context Injection. Stage 4 maps this classification to the Section 4 control plane, distinguishing four direct controls (Secure Gateway, Zero Trust, Fine-Grained Authorization, Monitoring and Logging) from two indirect controls (Containerization, Cryptographic Signing) per Table 7.

## 4.6 Monitoring and Logging

Robust monitoring and logging provide the autonomous detection and forensic layer for MCP deployments by turning each authentication event, tool invocation, and data transfer into security evidence that can be queried and correlated. When telemetry is treated as a first class signal rather than as a by product of execution, hosts and operators gain the ability to detect misconfigurations and attacks in near real time while also preserving the traces needed to reconstruct how threats emerge and propagate across tools, servers, and agents. This visibility begins with **comprehensive event capture**. Instead of recording only final outcomes, deployments should treat every major control point as a source of evidence. Authentication attempts, tool invocations, prompt characteristics such as size and selected tools, configuration edits, policy updates,

and security control decisions should each produce structured traces. These records should then flow into immutable append only or WORM storage so that incident responders retain evidence they can trust when reconstructing what occurred Kent & Souppaya (2006). Once this telemetry stream is established, operators can route it through a centralized log-management and correlation pipeline, such as a SIEM, that aggregates events from heterogeneous sources, normalizes and correlates them, and supports analysis of attacker activity across components Kent & Souppaya (2006). And, as monitoring moves closer to execution, **runtime syscall and container telemetry** would extend this evidence layer into the underlying infrastructure. In Kubernetes based MCP deployments, hosts and servers should not be treated as opaque runtime units because the underlying nodes can observe each workload through a stream of low level signals. Kernel level and container aware sensors such as eBPF based probes and Falco daemons can monitor system calls in real time and enrich each event with container, pod, and namespace context Sysdig (2023a;b). Building on this visibility, Falco applies rule driven policies to characterize how an MCP component behaves over time and raises alerts when observed patterns depart from established baselines, such as an unexpected shell spawn, a write into a protected directory, an attempted privilege escalation, or an outbound socket to an unusual destination Sysdig (2023a;b). In Kubernetes environments, this style of monitoring aligns naturally with existing isolation boundaries because pod scoping and namespaces provide clearer behavioral units, making early signs of compromise easier to distinguish from routine system noise GCP (2023).

This infrastructure level telemetry becomes even more valuable when connected to **identity aware request tracing** at the protocol boundary. In MCP deployments, the first policy decision about whether an agent may reach a given server often occurs at an identity aware proxy or gateway rather than inside the tool itself. At that boundary, each JSON-RPC request and connection attempt can be recorded as an audit event that links the requesting principal, authorization context, targeted server or tool, invoked method, policy decision, and relevant runtime metadata. Recent MCP security work motivates this design. Gateway-based governance provides provenance tracking and end-to-end auditability Errico et al. (2025), while protocol-level extensions add comprehensive, tamper-evident audit logs for admission and policy decisions Hou et al. (2026a); Metere (2026).

As these records accumulate, they form an identity centered trace of MCP activity that makes tool use, context access, and configuration changes directly observable. They also align naturally with compliance requirements around access monitoring and change traceability, which strengthens the evidentiary basis for frameworks such as SOC 2 and HIPAA in MCP based systems. Once request level and runtime telemetry are in place, **behavioral baselining and anomaly analytics** help distinguish ordinary activity from suspicious deviations. ~~In MCP deployments, behavioral models can characterize routine operation by learning statistical or machine learning baselines over signals such as per tool call frequency, typical prompt length and structure, response size, inter request timing, and the chains of tools and servers that recur in normal workflows ScaleSec (2025).~~ In MCP deployments, behavioral models can characterize routine operation by tracking signals such as tool-call frequency, prompt and response structure, inter-request timing, invoked methods, and recurring tool or server chains. Recent MCP security work supports this direction through behavioral-deviation detection, where tool intent and execution trajectories are analyzed to identify malicious or abnormal MCP server behavior Huang et al. (2026c). Practitioner monitoring guidance further illustrates how such telemetry can be operationalized as detection rules for suspicious MCP activity Mooney (2025).

With such a baseline established, departures from it become meaningful indicators of possible misuse. A sudden spike in invocations of a previously quiet tool, a sequence of abnormally long prompts directed at a sensitive server, or an unexpected fan out across many downstream resources can all indicate prompt injection, unauthorized tool chaining, or covert enumeration. These patterns can then trigger targeted investigation or automated containment before the behavior develops into a broader incident.

A closely related extension of this analysis is **data exfiltration pattern analysis**, which focuses on how anomalous outbound behavior may reveal covert leakage and therefore connects naturally to the broader problem of telemetry integrity. MCP operators should correlate outbound request volume, destination cardinality, payload size variance, and time of day deviations to identify possible exfiltration channels. Outlier egress flows become especially concerning when they target newly observed domains or exhibit abrupt entropy shifts in transferred data, since such patterns may indicate staging, covert encoding, or unauthorized movement of sensitive information. Detecting these anomalies early allows the monitoring

layer to trigger containment workflows before large scale leakage is completed. For these signals to remain operationally useful, however, the underlying telemetry must also be collected and preserved in a trustworthy form. Deployments therefore need **sanitization and integrity controls** that protect both evidentiary value and operational safety by scrubbing or tokenizing credentials, secrets, and personally identifiable information before logs are aggregated, while still retaining structural features such as field names, lengths, and categorical tags that are necessary for investigation. At the same time, each log record should be cryptographically hashed, or linked through chained hashing, so that tampering becomes detectable if any record is altered after collection. Trust can be strengthened further by periodically anchoring hash roots to an external trusted timestamping service or ledger. Supported by accurate time synchronization and retention policies aligned with incident response requirements and regulatory minima, these controls ensure that telemetry remains both safe to store and reliable enough to support forensic reconstruction and accountability.

Taken together, these control families show that MCP security depends on coordinated defenses across the full request lifecycle, not on any single safeguard. Secure gateways reduce exposure at the boundary. Zero trust, cryptographic signing, and fine grained authorization restrict which identities, tools, and artifacts can be trusted and under what conditions. Containerization and sandboxing help contain compromise and reduce blast radius when prevention fails. Monitoring and logging preserve the visibility needed to detect misuse, investigate incidents, and support recovery. Figure 14 makes this concrete by tracing the RADE attack path from poisoned input to cross tool propagation, and by showing how layered controls can disrupt that path at multiple stages.

## 4.7   Mitigation Responsibilities Across the Matrix

The control families in Sections 4.1–4.6 describe defenses in aggregate. Because MCP security is a shared responsibility, we now make explicit, for each region of the dual-axis matrix (Section 3), how the corresponding threats can actually be mitigated at two complementary levels. *Systemic* mitigations are owned by the operator or platform and act across servers, such as gateways, trusted registries, sandboxes, and policy engines. *Local-developer* mitigations are implemented inside an individual MCP server or tool by its author, such as input validation, scoped tokens, secret handling, and structured logging. Table 10 pairs each taxonomy category, anchored to its dominant component and lifecycle phase, with a concrete mitigation at each level and a pointer to the control family (Sections 4.1–4.6) or machine-checkable artifact (Section 4.8) that realizes it. The two levels are complementary: the systemic control bounds blast radius and enforces a uniform baseline, while the local control removes the threat's root cause at the point where the developer has the most context.

Table 10: Implementable mitigations for each part of the dual-axis matrix, separated into systemic (operator/platform) and local-developer (per-server/tool) responsibilities. Bracketed pointers name the realizing control family (§4.1–§4.6) or artifact (§4.8).

| Category (component × lifecycle) | Systemic mitigation (operator / platform) | Local-developer mitigation (per server / tool) |
|---|---|---|
| Identity & Trust (Transport/Server × Connection) | Terminate traffic at an identity-aware gateway enforcing mTLS and verify server identity against a trusted registry/PKI; keep namespace allowlists so look-alike names cannot resolve Model Context Protocol (2025r); Errico et al. (2025). [§4.1] | Pin the server certificate or public key in client config; refuse unauthenticated `stdio`/HTTP transports; validate issuer/audience before binding SlowMist Team (2025). [§4.8, L3] |
| Supply Chain & Integrity (Server/Tools × Creation, Update) | Admit only signed, hash-pinned artifacts with SLSA provenance from a trusted registry; fail CI/CD on signature/provenance failure SLSA Community (2025); Sigstore (2022); Souppaya et al. (2022b). [§4.8, L1/L5] | Sign your server/tool and publish a manifest with `sha256` + declared scopes; pin and SCA-scan dependencies; verify installer provenance before running Bhatt et al. (2025). [§4.8, L1] |

| Category (component × lifecycle) | Systemic mitigation | Local-developer mitigation |
|---|---|---|
| Malicious / Poisoned Tools (Tools/Client × Discovery, Invocation) | Bind tools by signed, versioned identity (not bare name); sanitize tool metadata at the host before it reaches the model; re-verify the tool hash on every update Bhatt et al. (2025); Young & Delimarsky (2025). [§4.8, L1] | Validate tool schemas; strip/escape names, descriptions, and docstrings; never interpolate parameters into shell; reference tools by pinned version OWASP Foundation (2025c). [§4.8, L2] |
| Prompt / Context Injection (Host/Client/Data × Invocation) | Treat retrieved/forwarded content as untrusted; constrain which tools the model may call after ingesting external data; tag provenance and isolate untrusted context Errico et al. (2025); Greshake et al. (2023). [§4.1, §4.8 L2] | Mark tool outputs and retrieved data as data-not-instructions; validate and length-limit content; gate high-risk tools behind explicit allow after retrieval OWASP (2023). [§4.6] |
| Credential Leakage (Data layer/Auth × Invocation, Shutdown) | Run a secret broker issuing short-lived scoped tokens at call time; DLP/egress scanning at the gateway; purge residual data at shutdown MCP Security Working Group (2025); Kissel et al. (2014). [§4.1] | Never hard-code secrets—fetch from a vault by opaque handle; scope tokens minimally; redact secrets from logs and errors; clear caches/temp on teardown Ferry (2025). [§4.8, L1] |
| Authorization Gaps (Auth Infra/Server × Creation, Invocation) | Evaluate every request in a central policy engine (OPA/Cedar) with JIT least-privilege, audience binding, context revalidation, and revoke-on-shutdown MCP Working Group (2025b). [§4.8, L3] | Enforce per-request scope/audience checks in the server; declare required scopes in the signed manifest; gate destructive actions behind explicit consent Lodderstedt et al. (2025); Bhatt et al. (2025). [§4.8, L2/L3] |
| Execution / Sandbox (Host/Server × Creation, Invocation) | Run each server in a non-root container with seccomp/AppArmor, read-only FS, default-deny egress, and quotas; admit only signed images; runtime detection (Falco/eBPF) Kubernetes (2022); Souppaya et al. (2017); Sysdig (2023a). [§4.1] | Drop capabilities and set `no_new_privs`; parse untrusted input safely (no unsafe deserialization); set per-call timeouts and rate limits; pin parser libraries Souppaya et al. (2017). [§4.8, L5] |
| Workflow / Config (Host/Client/Server × Update, Shutdown) | GitOps with signed manifests and drift detection; uniform config baselines; centralized high-granularity audit logging; workflow-layer rollback; complete de-registration Errico et al. (2025); Chang & Geng (2025). [§4.6] | Emit structured per-call audit records; return sanitized errors (no internals); make operations idempotent/compensatable; remove tool entries on teardown Sakib (2025). [§4.8, L4] |
| Emergent / Multi-Agent (Host/LLM × Invocation, Training) | Isolate per-tenant model contexts; verify agent handoffs with audience-bound tokens (no passthrough); monitor for collusion/drift; govern shadow MCP via discovery + allow-lists Model Context Protocol (2025l); Motwani et al. (2024). [§4.6] | Scope each agent's tools minimally; verify peer identity before delegating; log inter-agent messages; avoid sharing one model/context across trust boundaries Anthropic (2025b). [§4.8, L3] |

Read together with the threat tables (Tables 4–6) and the coverage map (Table 7), Table 10 closes the loop from *where/when/what* a threat arises to *who* implements its mitigation and *how*. It also clarifies that no single actor is sufficient: operators cannot patch insecure tool code, and developers cannot enforce a fleet-wide trust boundary, so durable MCP security requires both levels acting on the same matrix cell.

## 4.8 Operationalizing Controls as Machine-Checkable Artifacts

The control families above specify *what* to enforce; here we make a representative subset *machine-checkable* by giving concrete artifacts that an operator can drop into a registry admission step, an authorization service, a logging pipeline, or a CI/CD gate. Each artifact is designed to *fail closed*, to emit a reviewable evidence record, and to be testable in isolation. Table 11 maps the five artifacts to the control family they implement and the threat category they target; the full listings appear in Appendix D. These artifacts are illustrative reference implementations rather than a complete policy suite, and they are intended to be adapted to a deployment's own registry, identity provider, and telemetry stack.

Table 11: Machine-checkable artifacts, the control family each implements, and the threat category each enforces. Listings appear in Appendix D.

| Artifact | Control family | Threat category enforced |
|---|---|---|
| Signed-manifest admission check (Listing 1) | Cryptographic Signing & Registry | Supply Chain; Malicious Tools |
| Scope-linting rules (Listing 2) | Fine-Grained Auth & Policy | Authorization Gaps; Credential Leakage |
| Token-audience validation (Listing 3) | Zero-Trust Architecture | Authorization Gaps (confused deputy) |
| Runtime log signals (Listing 4) | Monitoring & Logging | Workflow / Logging Gaps; Exfiltration |
| CI/CD failure conditions (Listing 5) | Cryptographic Signing & Registry; Containerization | Supply Chain; Patch Gaps; Config Drift |

Two design rules connect these artifacts to the taxonomy. First, enforcement is placed at the *earliest* lifecycle phase where the failure can be prevented rather than where its symptom appears: scope-linting and signed-manifest admission act at creation/registration, audience validation at invocation, and CI/CD gates at update/maintenance. Second, every artifact produces a structured evidence record (Listing 4) so that an enforced denial is also an auditable event, directly supporting the observability controls discussed in Section 4.6.

## 5 Structured Evaluation of Verifiable Checks

~~After introducing the control families, we now test whether they can be implemented as concrete checks in a real MCP deployment. The point of this evaluation is not to build a large benchmark or to claim broad empirical coverage. Instead, it examines a narrower question that follows directly from the framework developed in the earlier sections. If the taxonomy is useful operationally, then checks derived from it should be able to detect representative MCP failures, place those failures at the correct component and lifecycle phase, and leave behind evidence that an operator could inspect later. In that sense, this experiment connects the descriptive taxonomy in Section 3 to the control-oriented view developed in Section 4.~~ After introducing the control families, we present a compact proof-of-concept evaluation of whether selected taxonomy-derived controls can be implemented as concrete checks in a real MCP deployment. The goal is not to build a large benchmark, claim broad empirical coverage, validate the full taxonomy, or assess all mitigation categories. Instead, the evaluation asks a narrower feasibility question: whether selected checks can produce expected enforcement outcomes, localize curated representative failures to their intended component and lifecycle phase, and leave behind inspectable evidence for operators. In that sense, this experiment connects the descriptive taxonomy in Section 3 to the control-oriented view developed in Section 4, while leaving broader empirical validation to future work.

The testbed is intentionally small, but it exercises the main interactions that matter for the taxonomy. It contains one host, one MCP client, and three servers that expose different kinds of risk. The first server provides a directory-scoped file tool over `stdio`. The second provides a network-facing fetch tool over HTTP with an egress allow-list. The third provides a privileged secret retrieval tool behind token and scope checks. Checks derived from the framework are then applied at the points where the system can

Table 12: Benchmark scenarios used to evaluate executable MCP controls

| ID | Scenario | Viol. | Component | Lifecycle | Expected evidence |
|----|----------|-------|-----------|-----------|-------------------|
| S1 | Over-broad file scope exposed through `read_file` | Yes | Tools | Creation / Registration | policy violation |
| S2 | Blocklisted outbound fetch denied by egress policy | Yes | Tools | Invocation / Execution | policy violation |
| S3 | Poisoned tool description rejected during discovery | Yes | Tools | Discovery | metadata violation |
| S4 | Unapproved tool hash rejected at admission | Yes | Tools | Creation / Registration | hash mismatch |
| S5 | Out-of-scope token denied for a privileged call | Yes | Auth. Infra. | Invocation / Execution | authorization denial |
| S6 | Unauthorized principal denied for a privileged call | Yes | Auth. Infra. | Invocation / Execution | authorization denial |
| S7 | Post-approval drift detected through hash divergence | Yes | Tools | Update / Maintenance | hash mismatch |
| S8 | A real egress denial occurs but the required denial trace is missing | Yes | Server | Update / Maintenance | observability gap |
| S9 | Approved tool with valid scope and metadata | No | Tools | Creation / Registration | clean execution |
| S10 | Authorized HTTP call with a valid token and complete evidence | No | Auth. Infra. | Invocation / Execution | clean execution |
| S11 | An in-session tool-definition mutation yields a hash mismatch although a naive phase mapping favors creation and registration | Yes | Tools | Update / Maintenance | hash mismatch |
| S12 | An over-broad scope failure surfaces as an authz denial although the root cause lies in tool configuration | Yes | Tools | Creation / Registration | authorization denial |

actually intervene. Scope checks reject requests when file access, destination rules, or token privileges exceed policy. Integrity checks compare approved and observed tool hashes, which makes it possible to catch both unapproved admission and later drift. Discovery checks block poisoned or policy-violating tool descriptions before invocation. Observability checks verify that selected denials leave behind the trace or audit artifact an operator would expect to review afterward. The benchmark therefore does more than record whether a request succeeds or fails. It also asks where the failure belongs, when the relevant control should have fired, and what evidence should remain once the event is over.

The benchmark contains 12 scenarios in total, including 10 violations and 2 benign cases as shown in Table 12. Table 13 complements Table 12 by mapping each scenario to the taxonomy category it exercises, the check expected to fire, the observed outcome, and the limitation of that curated case. The violating cases were chosen to cover the kinds of failures the framework is meant to distinguish, including over-broad scope, blocked outbound fetches, poisoned metadata, integrity failures at admission, authorization denials, post-approval drift, and an observability-gap case. Two additional cases were designed to separate single-axis from dual-axis reasoning more clearly. In one of them, an in-session tool-definition mutation produces a hash-mismatch signal that a naive mapping associates with creation and registration even though the correct phase is update and maintenance. In the other, an over-broad scope problem appears as an authorization denial even though the root cause lies in tool configuration rather than in the authorization infrastructure.

~~Each scenario is annotated with a ground-truth failure mode, component label, lifecycle label, expected enforcement outcome, and expected evidence artifact.~~ Each scenario is annotated with a component label, lifecycle label, expected enforcement outcome, and expected evidence artifact; localization is then derived from structured runtime signals emitted by the benchmark servers rather than from a failure-mode answer key. The two benign cases play an equally important role because they confirm that valid registration and valid authorized execution do not trigger false alarms.

~~The value of the dual-axis view becomes clearer when it is compared with two reduced baselines. One baseline assigns only a lifecycle phase using naive signal-based inference from runtime output. The other assigns only a component using the same kind of runtime signal. By contrast, the proposed method uses the taxonomy to trace the failure back to its source and assigns both a component and a lifecycle phase.~~ The value of the dual-axis view becomes clearer when it is compared with two reduced ablations. These ablations are not intended to represent modern security frameworks; they isolate the contribution of using both axes. One ablation assigns only a lifecycle phase using coarse response-level signals. The other assigns only a component using the same kind of signal. By contrast, the dual-axis method uses taxonomy-informed interpretation of structured runtime signals to assign both a component and a lifecycle phase. This difference matters because the visible symptom and the true enforcement point often do not coincide. A denial at runtime may reflect a scope problem that should have been constrained earlier. A hash mismatch may be observed only after the system is already running. An observability failure may appear only when a real denial occurs but the expected trace is missing. For that reason, performance is evaluated along four dimensions, whether a method flags real violations, whether it avoids false alarms on benign cases, whether it localizes failures correctly, and whether the expected audit artifact is actually produced. For the dual-axis method, correct localization requires both the right component and the right lifecycle phase. ~~For the reduced baselines, correctness is defined only on the single axis each one models.~~ For the reduced ablations, correctness is defined only on the single axis each one models. We compare the dual-axis view qualitatively against established threat-modeling frameworks such as STRIDE, MAESTRO, and MITRE ATLAS in Appendix B, positioning them as complementary reference frameworks rather than direct empirical baselines.

Table 14 reports the final results. All three methods detect all 10 violating scenarios and raise no false positives on the 2 benign cases, so the main difference does not lie in simple detection. It appears in localization. The dual-axis method localizes all 10 violating cases correctly, while the lifecycle-only ~~baseline~~ ablation localizes 5 of 10 and the component-only ~~baseline~~ ablation localizes 8 of 10. The asymmetry is important. Component-only reasoning still retains part of the failure location, but lifecycle-only reasoning degrades more sharply when the visible signal does not match the phase in which the relevant control should have fired. This is clearest in S1 and S12, where the observed denial signal is associated naively with invocation and execution even though the root cause belongs to creation and registration. S7 and S11 expose two integrity-related mismatches. In S7, the live tool hash diverges from the registered approval baseline, capturing post-approval drift. In S11, the tool definition is mutated during an active session and detected by re-verifying the live hash against the session-start snapshot. In both cases, the hash-mismatch signal is mapped naively to creation and registration, but the true phase is update and maintenance. S8 reveals a related but slightly different failure. There, the runtime denial is real, but the benchmark then checks for the required server-side denial trace and finds that it is missing, which makes the failure a workflow/configuration observability gap rooted in the server and its maintenance state rather than only a policy-enforcement event.

S12 also shows why component-only reasoning is insufficient. The system returns an authorization denial, but the root cause lies in tool configuration and registration rather than in the authorization infrastructure itself. ~~The dual-axis method resolves these ambiguous cases by using failure-mode annotations from the scenario taxonomy to trace each failure back to its source, whereas the single-axis baselines rely only on the raw response signal.~~ The dual-axis method resolves these ambiguous cases through taxonomy-informed interpretation of structured runtime signals, including denial-origin fields, integrity-report fields, overbroad-scope flags, and missing-trace audit gaps, whereas the single-axis ablations rely only on coarse response-level signals. These are precisely the situations in which a single-axis view can still detect that something went wrong while losing track of where responsibility lies or when the control should have fired. Evidence completeness is 100% in the final run because each violating case produces the expected artifact. ~~Taken together, these findings show that the benchmark primarily serves to validate whether the framework can~~

Table 13: Benchmark coverage: each scenario mapped to the taxonomy category it exercises, the expected check, the observed outcome, and the limitation of that curated case. Taxonomy category names follow Section 3.

| ID | Taxonomy category | Expected check | Observed outcome | Does not prove |
|---|---|---|---|---|
| S1 | Authorization & Access Control Gaps | File-scope allow-list check | Denied; policy-violation evidence emitted | Single curated path; scope fixed at configuration time; no adaptive bypass |
| S2 | Authorization & Access Control Gaps | Egress allow-list check | Denied; policy-violation evidence emitted | Static allow-list; no evasion, domain fronting, or dynamic destination abuse |
| S3 | Malicious or Poisoned Tool Servers | Poisoned-metadata description check | Rejected at discovery; metadata-violation evidence emitted | Single metadata pattern; no obfuscated or multi-step injection |
| S4 | Supply Chain & Code Integrity | Tool-hash admission check | Rejected at admission; hash-mismatch evidence emitted | Curated approved hash; no admission-time TOCTOU or package ecosystem attack |
| S5 | Authorization & Access Control Gaps | Token-scope check | Denied; authorization-denial evidence emitted | Single token and server; no replay, escalation, or federation failure |
| S6 | Authorization & Access Control Gaps | Principal / identity authorization check | Denied at transport with HTTP 403; authorization evidence emitted | Static principal policy; no dynamic, federated, or compromised-identity case |
| S7 | Supply Chain & Code Integrity | Hash-drift check against registered approval baseline | Post-approval drift detected; hash-mismatch evidence emitted | Synthetic baseline; single integrity checkpoint; no broad supply-chain campaign |
| S8 | Workflow & Configuration Risks | Audit / trace-completeness check | Real denial occurs, but required trace is absent; observability gap flagged | Single observability toggle; one audit-gap mode only |
| S9 | N/A (benign control) | Scope and metadata checks expected to pass | Allowed; clean execution; no false alarm | Happy-path control only; does not test adversarial behavior |
| S10 | N/A (benign control) | Token and scope checks expected to pass | Allowed; clean execution; no false alarm | Single valid-token case; does not test authorization edge cases |
| S11 | Supply Chain & Code Integrity | Runtime integrity re-verification against session-start snapshot | In-session mutation detected; hash-mismatch evidence emitted; localized to Update / Maintenance | Synthetic mid-session mutation; isolates phase localization rather than broad runtime compromise |
| S12 | Authorization & Access Control Gaps | Scope check with overbroad-scope policy flag | Denied; authorization evidence emitted; root cause localized to tool configuration | Synthetic misconfiguration; isolates component localization rather than arbitrary authorization failures |

Table 14: Final benchmark results

| Metric | Dual-axis | Lifecycle-only | Component-only |
|---|---|---|---|
| Violation detection rate | 100% | 100% | 100% |
| False positive rate | 0% | 0% | 0% |
| Scenario-level localization agreement | 100% | 50% | 80% |
| Evidence completeness | 100% | 100% | 100% |
| Localized / detected | 10/10 | 5/10 | 8/10 |

~~localize failures correctly and preserve actionable evidence.~~ Taken together, these findings show that the benchmark primarily serves as a compact proof of concept for whether selected checks can localize curated failures as expected and preserve actionable evidence.

~~The benchmark remains intentionally compact and should therefore be interpreted as a structured validation exercise rather than as a comprehensive suite. Even with that limitation, it shows that the framework has operational value beyond description alone. Checks derived from the taxonomy can be implemented in a real MCP stack, they can detect representative failures reliably, and they can place those failures more accurately than reduced single-axis views while preserving concrete audit evidence for debugging and review. Larger multi-host and multi-agent settings may reveal further interactions, but the central result already holds in this smaller deployment. The additional structure in the taxonomy does not merely reorganize known risks. It improves how failures are localized and how the resulting evidence can be used.~~ The benchmark remains intentionally compact and should therefore be interpreted as a proof-of-concept feasibility study rather than as a comprehensive validation suite. Within this controlled setting, the results show that selected checks derived from the taxonomy can be implemented in a real MCP stack, detect curated representative failures, localize them to expected component and lifecycle labels, and preserve concrete audit evidence for debugging and review. These results do not establish taxonomy completeness, general real-world detection performance, or the effectiveness of all mitigation categories. Larger multi-host, multi-agent, and independently constructed benchmarks are needed before making broader empirical claims.

## 5.1 Limitations of the Benchmark

The benchmark has several limitations. First, it is intentionally compact and uses one host, one MCP client, three servers, and a small set of curated scenarios. Second, the scenarios are designed to exercise selected taxonomy cells and verification hooks, so high agreement in this setting should be interpreted as an implementation sanity check rather than evidence of general classifier performance. Third, the benchmark covers a representative subset of controls rather than the full mitigation space in Section 4. It therefore does not validate the completeness of the taxonomy or the effectiveness of all mitigation categories. Fourth, the testbed does not model large multi-host deployments, heterogeneous registries, multi-agent collusion, adaptive attackers, noisy production telemetry, or independently generated failures. These limitations mean that the benchmark provides feasibility evidence that selected taxonomy-derived checks can be made executable and auditable in a controlled MCP deployment, while broader empirical validation remains future work.

## 5.2 Scaling Considerations for Larger MCP Deployments

The benchmark above is intentionally compact and should be read as a feasibility study for executable checks rather than as evidence of production-scale enforcement. Larger MCP deployments are different, since they involve many hosts, agents, tenants, registries, and server versions. In that setting the proposed controls can no longer be local checks on a single host–client–server path, and must instead run as shared platform services. At the entry point, gateways take on admission and policy enforcement, and registry checks extend this into supply-chain gates that vet servers and versions before they are admitted. Authorization then moves inward, binding each request to a specific tenant, principal, session, tool, and token audience so that access is scoped at every call. Underpinning all of this, telemetry is aggregated across hosts and servers, which is what allows detection and audit to work fleet-wide rather than per host.

Multi-agent and multi-tenant deployments raise the stakes for isolation and delegation control. Session state, memory, retrieved data, tool outputs, and audit traces should be partitioned by tenant and agent identity, and each agent handoff, delegated authorization, and cross-tenant data flow should be checked explicitly. The same component×lifecycle structure from our taxonomy still applies here. What changes is the enforcement substrate, which moves from local scripts to distributed policy engines, admission controllers, registry attestations, tenant-aware authorization services, and production monitoring pipelines.

These larger deployments also introduce scaling risks that our benchmark does not measure, such as policy-distribution lag, inconsistent tenant configuration, noisy telemetry, cross-tenant access attempts, correlated failures across shared servers, and multi-agent collusion. In short, the benchmark shows that selected controls can be made executable and evidence-producing in a real MCP stack, while evaluating their latency, throughput, robustness, and tenant-isolation guarantees in realistic multi-agent and multi-tenant deployments remains important future work.

# 6 Emerging Frameworks and Protocol Extensions from MCP Security Research

Having mapped where and when MCP-specific threats emerge through our dual-axis taxonomy in Section 3, attached concrete controls and verification hooks to each failure mode in Section 4, and ~~evaluated those checks in a compact benchmark~~ illustrated selected checks in a compact proof-of-concept benchmark in Section 5, we now turn to how these ideas are beginning to take shape in real systems. At this point, the question is no longer only which threats exist or which controls are desirable in principle, but how recent MCP security proposals are translating them into practical mechanisms. Seen in this way, emerging frameworks can be read as early attempts to operationalize parts of the control plane developed in the previous sections, with each emphasizing a different point in the MCP lifecycle and a different portion of the attack surface. Rather than exhaustively surveying every prototype, we focus on a small set of representative efforts that show how runtime guards, Zero Trust patterns, protocol-aware intermediaries, and security-conscious interfaces are beginning to crystallize within the MCP ecosystem.

One prominent direction is represented by **Model Contextual Integrity Protocol**, proposed by Jing et al. (2025), which augments MCP with structured tracking and a dedicated guard model along the tool invocation path. MCIP introduces a Model Contextual Integrity log format that records each MCP interaction as a trajectory of information flows, giving tool calls and their surrounding context a structured representation that can be audited and reused as labeled telemetry. On top of this telemetry, the MCIP Guardian model classifies trajectories and individual tool invocations as safe or as one of several interaction phase risk types, allowing the system to block or flag unsafe calls even when they are syntactically valid. Through the lens of our taxonomy, this framework most directly addresses Prompt Input & Context Injection threats such as Poisoned Prompt Templates, RADE, Format Embedded Poisoning, and related interaction phase misuse of tools on the client and server sides. At the same time, it provides a concrete realization of the Monitoring & Logging and Fine Grained Authorization & Policy control families discussed earlier, since the MCI logs instantiate the structured telemetry assumed by those controls and the Guardian model acts as a runtime decision point for individual tool calls.

A related but more infrastructure centric direction appears in the **MCP Gateway Architecture** described by Brett (2025). In this design, a gateway sits between model hosts and self hosted MCP servers as a reverse proxy, terminating TLS, centralizing OAuth 2.1 authentication, and funnelling all JSON RPC traffic through a single inspection and routing point with secure tunnels to backend servers. This shifts security enforcement toward the network and protocol boundary, which makes the gateway a natural embodiment of the Secure Gateway / Firewall family developed in Section 4. In particular, it aligns closely with the Application Layer Gateways and Validation and Host based Firewalls and OS Hardening strategies, since the gateway becomes the protocol aware choke point through which requests are authenticated, inspected, rate limited, and routed before they ever reach an MCP server. Because OAuth enforcement and scoped tokens are attached at this boundary, the design also supports the Zero Trust controls of Per Request Identity Verification and Just In Time Least Privilege Access. As a result, the gateway pattern directly mitigates several Identity & Trust Management threats, including Server Name Collisions and Impersonation, Mutual Authentication and Encryption Gaps, Lack of Verified Server Identity or PKI, and Implicit Trust in Unauthenticated Servers, while also reducing part of the blast radius for Prompt Input & Context Injection and Credential Leakage and Data Exposure by ensuring that cross boundary traffic is constrained and logged at a single ingress point. This emphasis on the gateway as a controlled choke point is pushed further by **MCP Guardian**, introduced by Kumar et al. (2025), which presents a security first middleware layer that proxies traffic between MCP clients and servers rather than simply relaying it. In this design, all MCP requests flow through the guardian, which enforces authenticated sessions with scoped credentials, applies strict per token and per tool rate limiting, and records end to end logs and traces for every call. At the same time, WAF style inspection examines JSON RPC payloads for malformed structure, oversized bodies, and common injection patterns before they reach a server. Compared with the broader gateway architecture discussed above, MCP Guardian places even greater weight on active security enforcement at runtime, and therefore serves as an especially direct realization of the Secure Gateway / Firewall, Zero Trust Architecture, and Monitoring & Logging control families. It concretely instantiates the Application Layer Gateways and Validation, Per Request Identity Verification, Just In Time Least Privilege Access, Comprehensive Event Capture, and Identity Aware Request Tracing strategies described earlier in the paper. Through this layered enforcement model,

MCP Guardian hardens several categories in our taxonomy at once, including Identity & Trust Management failures such as Server Name Collisions and Impersonation and Implicit Trust in Unauthenticated Servers, as well as Malicious or Poisoned Tool Servers and Prompt Input & Context Injection threats whose payloads can be filtered at the edge. It also improves visibility into Credential Leakage & Data Exposure by routing sensitive flows through a single observable middleware layer rather than through multiple ad hoc endpoints.

While these gateway oriented systems concentrate primarily on protocol boundaries and live request enforcement, **Enterprise Grade Security**, proposed by Narajala & Habler (2025), broadens the focus to a defense in depth blueprint for hardening MCP deployments as a whole. Their framework begins from a systematic threat model and adapts NIST and OWASP guidance to the MCP stack, arguing for MCP aware application gateways, continuous identity verification, host based EDR and HIDS, segmentation, and rich telemetry for intrusion detection. This broader framing is important because it connects several control families that are often treated in isolation and instead presents them as mutually reinforcing parts of one deployment model. Under our taxonomy, this blueprint directly addresses Authorization & Access Control Gaps such as Decentralized and Inconsistent Policy Enforcement, No Context Aware Revalidation, Uneven RBAC Support, Binary Permissions, and Confused Deputy or Role Leakage, while also covering Supply Chain & Code Integrity risks such as Trojanized or Compromised Installers and Unverified Package Sources and Automatic Invocation. From the perspective of Section 4, it operationalizes the Secure Gateway / Firewall, Zero Trust Architecture, Cryptographic Signing & Registry, Fine Grained Authorization & Policy, and Monitoring & Logging families through segmented deployments, hardened identity paths, MCP aware gateways, and SIEM backed intrusion detection over MCP traffic.

A different form of operationalization appears in **MCP Safety Scanner**, developed by Radosevich & Halloran (2025), which approaches MCP security through empirical auditing rather than through an inline enforcement layer. Their work shows that LLMs connected to standard MCP servers can be steered into malicious code execution, remote access control, and credential theft during routine workflows. Building on the `McpSafetyScanner` codebase, they instantiate an agent based auditor that discovers a server's tools, crafts adversarial test flows, and returns a structured report of vulnerabilities and suggested remediations that operators can run before deployment. This shifts attention from runtime mediation to pre-deployment and recurring assessment, which makes the framework especially relevant to the interaction between Prompt Input & Context Injection, Malicious or Poisoned Tool Servers, and Credential Leakage and Data Exposure during Invocation / Execution. In terms of our mitigation strategies, it points toward MCP specific scanning as a practical extension of the Cryptographic Signing & Registry and Monitoring & Logging families, where recurring audits and adversarial checks complement registry hygiene, version pinning, and runtime telemetry by catching high impact misconfigurations before they become cross workflow incidents.

~~This auditing oriented perspective is developed further by **MCP Scan**, introduced by McGinley-Stempel (2025), which treats tool manifests and installed MCP configurations as a supply chain that should be inspected rather than blindly trusted. The `mcp-scan` CLI inspects local MCP configuration files for clients such as Claude, Cursor, and Windsurf, enumerates all registered tools, and runs static checks over their descriptions and metadata to flag known prompt injection and tool poisoning patterns. It also detects cross origin tool shadowing, in which one server silently overrides another's tools, and identifies rug pull scenarios by comparing current tool hashes against previously known values. In practical terms, it plays a role similar to `npm audit` for MCP and can already be embedded into CI pipelines so that changes to tool manifests or newly added servers are scanned automatically before deployment. Through our taxonomy, this framework strengthens Prompt Input & Context Injection, Supply Chain & Code Integrity, and Malicious or Poisoned Tool Servers by turning poisoned metadata, cross origin escalation, and silent post approval redefinition into concrete findings that can block a release. From the perspective of Section 4, it gives operators a practical way to realize the Cryptographic Signing & Registry family in everyday workflows, particularly Trusted Tool Registries, allow listed tools, and change control, because its hash based checks and drift reports mirror the guidance to reject unknown tool identities and block execution on unapproved definition changes even though the current implementation remains primarily static and rule based.~~

A related practitioner tool example is `mcp-scan`, described by McGinley-Stempel (2025). This tool inspects local MCP client configuration files, enumerates registered tools, and applies static checks over tool descriptions and metadata for patterns associated with prompt injection, tool poisoning, cross-origin tool

shadowing, and post-approval definition changes. We use this source as practitioner evidence that MCP manifests and installed tool configurations are increasingly treated as auditable supply-chain artifacts, not as primary evidence of scanner effectiveness. In our taxonomy, such scanners provide partial operational checks for Prompt Input & Context Injection, Supply Chain & Code Integrity, and Malicious or Poisoned Tool Servers by flagging suspicious metadata or unexpected tool-definition drift before deployment. However, these checks should be viewed as complementary to, rather than a substitute for, stronger mechanisms such as cryptographic signing, trusted registries, allow-listed tool identities, change-control review, and runtime monitoring.

Taken together, these frameworks show that the dual axis taxonomy and control plane developed in this paper already map onto concrete mechanisms such as structured context logs, guard models, protocol aware gateways, middleware enforcement layers, and supply chain scanners. At the same time, the proposals surveyed here tend to concentrate their effort on a relatively narrow band of risks centered on prompt injection, malicious or misconfigured servers, and unsafe tool use during live interaction. Many also define their own policy schemas, telemetry formats, or trust assumptions rather than converging on shared standards, and only partially address lifecycle wide hardening, multi tenant deployments, and multi agent coordination. As a result, real world MCP deployments will still need to assemble these point solutions into more standardized patterns for authentication, authorization, evaluation, and continuous assurance so that defenses can be composed, compared, and audited across hosts and servers. The next section therefore moves beyond individual systems and turns to broader research directions, with particular attention to common security schemas, shared benchmarking infrastructure, and more robust self monitoring behavior in MCP based ecosystems.

## 7 Limitations and Future Directions

This work has important limitations that directly shape the research directions that follow. First, the benchmark is deliberately limited in scope and is intended to show that the framework can be instantiated, enforced, and evaluated in practice, rather than to offer broad empirical coverage of MCP deployments. Second, the taxonomy is built from a mixed evidence base that includes archival research, specifications, incident reports, and practitioner sources. That breadth is appropriate for a fast-moving ecosystem, but it also means that some categories are supported more by emerging operational evidence than by mature longitudinal study. Third, although the paper shows that dual-axis mapping improves localization, control placement, and evidence collection, it does not yet provide a fully operationalized framework for deployment-wide scoring or assurance.

Taken together, these limitations point to the broader question of what a more mature MCP security ecosystem should look like. Table 15 maps the seven open problems that emerge from these gaps: the reason each remains unsolved, the research required to close it, and the taxonomy region most affected. Each of the seven open problems expands into a dedicated subsection (§7.1–§7.7), where rows 6 and 7 cover the directions with the most ground still to cover. Addressing these problems requires moving beyond ad hoc controls and one-off prototypes toward shared mechanisms that can be embedded in specifications, implementations, evaluation practice, and governance, so that MCP security can develop as a coordinated discipline rather than remain a loose collection of isolated tools.

Table 15: Open problems for MCP security: the gap keeping each unsolved, the research required to close it, and the research type and taxonomy area involved. Rows 1–7 map to §7.1–§7.7

| Open problem | Why it remains open | Research required | Type / taxonomy area |
|---|---|---|---|
| **Protocol-level authentication & authorization** See §7.1 | AuthN/AuthZ are optional in the spec; each server uses bespoke identity and scopes, so access control cannot be audited deployment-wide. | An OAuth2/PKI/mTLS profile for MCP; signed manifests declaring scopes; machine-checkable per-user/per-role policy with security-context propagation. | Protocol & standards §3.1, §3.6 |

*Table 15 continued*

| Open problem | Why it remains open | Research required | Type / taxonomy area |
|---|---|---|---|
| **Standardized security benchmarks & adversarial evaluation** See §7.2 | No shared attack corpus or comparable risk scores; evaluation targets server-side flaws, not agent behavior under adversarial conditions. | A reusable MCP security benchmark; agent-behavior adversarial test modules; an MCP CTF series; community red-team and incident sharing feeding a hardening guide. | Benchmarking & evaluation §3.4, §3.3, §3.9 |
| **Human oversight & graded consent UX** See §7.3 | Consent is coarse (allow/deny) despite high-impact actions; no graded or standing-preference controls, kill-switches, or governor standard. | Graded consent interfaces; standing-preference policies; agent-governor and dual-approval mechanisms; kill-switch standards; evaluated with HCI methods. | HCI & human-in-the-loop §3.6, §3.9 |
| **Governance, certification & ecosystem trust** See §7.4 | Trust rests on self-reported metadata; no vetted registry, security certification, or threat-intel feed for malicious servers and poisoned tools. | Signed reviewed registries; certification schemes (SOC2/HIPAA-style); an MCP threat feed; coordinated baselines via standards bodies and regulatory mandates. | Governance & policy §3.2, §3.1 |
| **Security-aware models & alignment for tool use** See §7.5 | LLMs readily follow injected instructions and lack security context; no robust internal integrity check before an action is taken. | Security-aware training (Constitutional AI / RLHF for tool safety); Prompt Flow Integrity; differential testing; context-aware access control; action-level explainability. | ML robustness & alignment §3.4, §3.9 |
| **Deployment-wide assurance, scoring & enforcement** See §7.6 | Dual-axis mapping improves localization but there is no operationalized framework for deployment-wide risk scoring or continuous assurance; the taxonomy is still a static reference. | Telemetry-driven analytics mapped to taxonomy cells; integrity enforcement comparing expected vs. observed behavior; uniform policy engines; aggregate risk scoring. | Systems & runtime §3.8, §4.6 |
| **Multi-agent identity, provenance & emergent-behavior detection** See §7.7 | No standard for agent-to-agent attestation; collusion, orchestration drift, and shadow-MCP governance are immature and largely unmeasured (E3 evidence). | Attested agent handoffs; inter-agent provenance tracking; collusion and drift detection; shadow-MCP discovery and governance; longitudinal measurement to raise evidence strength. | Systems & ML §3.9 |

## 7.1 Standardized Authentication & Authorization Schemas

There is a clear need for official authentication and authorization frameworks in the MCP specification. At present, **authentication and authorization are explicitly optional for MCP implementations**, as noted in the official draft Model Context Protocol (2025l). In practice this means a client can talk to several MCP servers in the same workspace, each with its own bespoke model of identity, credentials, and permissions. One server might use API keys, another might rely on environment variables, and a third might skip authentication entirely, so access control behavior becomes hard to reason about or audit at system level. A more coherent future is one where MCP defines an OAuth 2.0 or PKI-style handshake so that clients can verify server identity (for example, servers sign their manifest and clients verify this against a certificate authority or trusted registry), and where there is a built-in notion of user or agent identity and roles so that access control can be expressed per user and per role. Standard schemas would then let an enterprise state, in one place, that "LLM agents of type X can only access tools A, B, C with read-only permissions," rather than encoding that policy in ad hoc configuration spread across many hosts.

Within this picture, research is needed into lightweight authentication methods that fit dynamic, tool-centric workflows. One plausible direction is a token exchange at session start, followed by JSON Web Tokens attached to each JSON-RPC call, carrying user, agent, and session context Microsoft (2025). This naturally connects to Zero Trust principles, where every call must be authenticated and authorized, not just the initial connection, and where servers treat each tool invocation as a fresh decision rather than assuming that once a channel is open it remains trustworthy.

In addition, **mutual authentication**, where both the client and server verify one another, would help prevent rogue clients or servers from silently joining the ecosystem. An MCP server offering access to sensitive internal systems might require **client-side certificates or API keys** issued only to approved AI applications, and refuse connections from anything else. The OAuth 2.0 ecosystem already supports this pattern with mutual-TLS (mTLS), which binds client credentials to certificates in a way that resists credential theft and replay Lodderstedt et al. (2020), and similar patterns could be adopted or profiled for MCP-aware stacks.

On the authorization side, standardized schemas could allow tools to declare their scopes in a way that closely mirrors OAuth scopes. A tool might declare that it can only read from a particular class of resources, or that it requires a specific organizational role such as `OrgAdmin` to perform write operations. Today, many of these constraints live out-of-band in documentation, environment configuration, or human convention. Encoding them directly in MCP manifests and protocol messages would make them machine-checkable and auditable, at the cost of additional complexity in server and client implementations. This direction is reflected in SMCP, which extends MCP with unified identity management, robust mutual authentication, ongoing security context propagation, fine-grained policy enforcement, and comprehensive audit logging Hou et al. (2026a). Together, these mechanisms show how authentication and authorization patterns could be embedded directly into the protocol itself.

It is plausible that researchers and industry groups, such as OWASP's Agentic Security Initiative OWASP Foundation (2025a), will push these patterns toward a more formal standard. An "MCP 2.0" security model could look much closer to OAuth-style connector security, with client identifiers, secrets, permission scopes, and first-class support for mechanisms such as mTLS or Decentralized Identifiers (DIDs) for trust establishment Sporny et al. (2022). In that world, MCP servers and hosts would speak a shared language of identities, credentials, and scopes, making it far easier to reason about which agents can call which tools, under what conditions, and with what blast radius if something goes wrong.

### 7.2 Security Benchmarking and Evaluation Frameworks

While tools like **MCPSafetyScanner** Halloran (2025) and early benchmarks such as **MCPBench** Modelscope Team (2025) are a useful first wave, the community still lacks *standardized benchmarking suites for MCP security*. A mature ecosystem would have a shared corpus of attack scenarios, including prompt injection chains, RADE bypass attempts, and over-privileged tool calls, that any new MCP implementation or LLM-based agent can be run against in a repeatable way. Much as **CVE benchmarks** help characterize software vulnerabilities across vendors MITRE Corporation (1999), an "MCP Security Benchmark" could drive convergence on comparable risk scores and structured evaluation reports, making it easier to understand how well a given stack withstands known classes of abuse.

In parallel, there is a need for **evaluation frameworks that focus on agent behavior under adversarial conditions**, not only on server-side vulnerabilities. These frameworks would spell out concrete behaviors to measure, such as whether agents seek user confirmation when instructions are ambiguous or high impact, how they respond when tool outputs conflict, and whether they leak sensitive data when juggling multiple contexts or tenants. Emerging standards like **NIST AI 600-1** National Institute of Standards and Technology (2024) and community-developed guidance such as **OWASP's Top 10 for LLMs** OWASP Foundation (2023b) could naturally evolve MCP-specific test modules, so that security evaluations cover both the protocol surface and the decision-making patterns of the agents that drive it.

One concrete direction is an **MCP CTF (Capture The Flag)** ecosystem, where researchers and practitioners interact with deliberately vulnerable MCP servers in a controlled environment. Practitioner projects such as Hack The Box's MCP Server and MCP-Kali-Server illustrate early experimentation with MCP-based

security training and hands-on evaluation environments Hack The Box (2025); Nahya (2025). We view a more formalized MCP CTF series as a promising future direction that could feed directly into benchmark suites and best practices.

Finally, **community-driven red-teaming and incident sharing** will be essential if MCP deployments are to harden quickly in the face of real-world attacks. Public efforts such as Microsoft's AI red-team work and DEF CON-style AI security contests illustrate how collaborative threat discovery can surface edge cases that formal analysis alone might miss Johnson & Mehrotra (2023); Newman (2023). Over time, such a feedback loop could converge into a widely adopted **MCP Hardening Guide**, analogous to secure configuration baselines for operating systems or cloud platforms, giving operators a concrete checklist for deploying and maintaining MCP systems with security in mind from day one.

### 7.3 Human-in-the-Loop UX and Control Design

Today, user consent UIs around AI tools are often rudimentary, even though the underlying actions can be complex and high impact. This creates a research and design opportunity to build more intuitive, graded, and informative consent interfaces for AI-driven tools Nakao et al. (2022); Mandel et al. (2019). Instead of a generic prompt such as "Allow tool to run? (Yes/No)", future interfaces could present richer, structured context, for example "This tool will delete five files. Estimated impact: High. Admin approval required." or require multi-factor authentication before performing sensitive actions. Giving users simple ways to express standing preferences (for example, "Never allow AI to send email externally" or "Ask me if more than 10 records will be affected") can bridge the gap between today's coarse consent and fully automatic operation. There is substantial Human–Computer Interaction (HCI) work to be done on how to surface an AI's intentions so that users can make informed decisions Cappuccio et al. (2025); Chromik & Butz (2021).

A second pillar is explicit oversight mechanisms such as kill switches and supervisory controls Orseau & Armstrong; Hadfield-Menell et al. (2017). Users and operators may need a visible "panic button" that immediately halts all tool usage when something appears to be going wrong, as well as an oversight mode where a human or a separate AI guardian monitors the agent's calls in real time and can intervene or veto. Recent work has proposed **Agent Governors**, dedicated processes that continuously watch an AI system's actions and decisions against a policy budget Xiang et al. (2024). In an MCP setting, this might correspond to an intermediary component that inspects each tool invocation and requires dual approval for certain categories of tools, similar to how safety-critical operations in other domains require two independent operators to confirm an action before it proceeds.

### 7.4 Governance, Certification, and Ecosystem Trust

For MCP to be viable at scale, especially in enterprises and regulated industries, there needs to be an ecosystem of trust and certification for third-party tools. This could take inspiration from mobile app stores or browser extensions, for example an MCP Server Marketplace where submissions undergo security review, automated scanning, and signing Apple Inc. (2021); Iqbal et al. (2023). We might also see the emergence of MCP security certifications, similar to SOC2 AICPA (2023) or FedRAMP FedRAMP Program Management Office (2024). For instance, an MCP connector handling healthcare data might need to certify HIPAA compliance, implying not just technical measures but documentation, audit logs, and third-party pen-tests.

Another governance angle is community-driven threat-intelligence sharing for MCP. As threats evolve, a central database of known malicious servers or attack signatures could help automatically guard systems, similar to anti-virus definitions or threat-intelligence feeds used in network security MITRE Corporation; Google (2024). An MCP-specific threat feed could distribute indicators for malicious servers, hijacked registry entries, poisoned tools, and vulnerable connector versions Phylum Research (2022).

From a standards perspective, groups like the **Frontier Model Forum** are already publishing shared safety recommendations and could define best practices for MCP usage. The Model Context Protocol Authorization draft maintained by Anthropic sketches concrete flows for securing MCP interactions with API tokens and OAuth-style credentials Model Context Protocol (2025l). The next step is to turn these proposals into a broadly adopted baseline, which will require coordinated governance and consensus building by AI safety

coalitions and open-spec bodies. In parallel, under the EU AI Act, high-risk AI systems must log key lifecycle events for audit and compliance purposes EUA (2024), creating a strong external push for MCP deployments to adopt robust, standardized security and audit practices.

## 7.5 Continual Model Robustness and Alignment

Finally, a more research-oriented direction is to improve the LLMs themselves to be more security-conscious when using tools. Currently, LLMs have a limited understanding of security context and will readily follow instructions Li et al. (2024); Zhao et al. (2025c); Ge et al. (2025). Future research might train models with "security awareness," so the AI can flag its own concerns: "This request would result in sending sensitive data externally, which seems unsafe. I should ask for confirmation." Early work in **Constitutional AI** Bai et al. (2022) instructs LLMs to critique their own actions according to a constitution, and **RLHF** has been shown to be effective in training models to obey complex human preferences Christiano et al. (2017). Recent efforts to extend Constitutional AI into tool-using agents, such as the Agentic Superego framework Watson et al. (2025), illustrate how internal ethics checks can be embedded into MCP workflows, allowing AI systems to proactively surface security-relevant explanations and seek user confirmation before executing sensitive actions.

One emerging concept is **Prompt Flow Integrity (PFI)**, a security framework aimed at ensuring that instructions received by an AI agent have not been tampered with and do not result in unintended privilege escalation Kim et al. (2025). PFI involves isolating prompt-processing stages and enforcing guardrails around tool-use permissions. To support this, **differential testing techniques**, where multiple agent instances cross-check each other's decisions, have been proposed to detect inconsistencies or malicious deviations Rao et al. (2025). Additionally, recent work shows that **instrumenting the model's reasoning process** via trace analysis or sabotage detection can proactively catch when an LLM is veering into unsafe behavior Kutasov et al. (2025). Related work is beginning to push this idea closer to MCP tool use itself, with MCPShield proposing an adaptive trust-calibration layer that evaluates risk before and after tool execution so that unsafe or suspicious actions can be constrained and reevaluated during runtime Zhou et al. (2026c). Together, these strategies can enhance the robustness of tool-using agents by enforcing internal consistency, highlighting anomalies, and providing early intervention signals within an MCP workflow.

Another future direction is **context-aware access control**, where the AI's own understanding of context labels (confidential, public, and so on) influences its tool choices, for instance refusing to use a public posting tool if the content is labeled confidential. This requires the AI to accurately classify data sensitivity, which is nontrivial but not impossible with metadata and training.

Lastly, integrating AI explainability into MCP workflows is vital because when a model can articulate why it is invoking a tool with specific data, users or oversight systems can better judge whether that action is appropriate. Bilal et al. Bilal et al. (2025) show that LLMs can produce rich, natural-language explanations of their own reasoning, and work on layered prompting in multi-agent settings demonstrates how to surface action-level justifications, creating a foundation for real-time human-in-the-loop validation before tool execution Faisal & Tunkel (2025).

## 7.6 Deployment-Wide Assurance, Scoring, and Enforcement

The dual-axis taxonomy improves threat localization and control placement, but it currently functions as a static reference: operators use it to structure design reviews and incident analysis rather than to drive live enforcement. A mature MCP security ecosystem requires moving the taxonomy from a descriptive artifact into an operational one.

Three concrete research directions would close this gap. First, **telemetry-driven anomaly analytics** that align observed deviations in MCP traffic, covering tool-call frequency, prompt size, egress patterns, and denial rates, with specific cells in the component-by-lifecycle matrix would give operators early warning of emerging attacks without requiring manual log review. Second, **integrity enforcement mechanisms** that continuously compare expected agent and tool behavior against observed execution and emit explicit violations when workflows deviate from policy would turn the taxonomy's control assignments into machine-

checkable invariants rather than design-time guidance. Third, **uniform policy engines** that express MCP-specific controls in a single rule language and drive configuration, deployment, and runtime enforcement from the same rule base would reduce the configuration fragmentation documented in §3.6 and §3.8. Together, these directions would produce an aggregate risk score per deployment, making it possible to compare security posture across MCP stacks and track improvement over time rather than treating each audit as a one-off exercise.

### 7.7 Multi-Agent Identity, Provenance, and Emergent-Behavior Detection

As MCP deployments shift from single assistants to graphs of cooperating agents, coordination itself becomes a security-critical surface that the current taxonomy can map but cannot yet measure or govern reliably. Three gaps stand out.

The first is the absence of a standard for **agent-to-agent identity and attestation**. The MCP Authorization specification requires audience-bound tokens and prohibits passthrough Model Context Protocol (2025l), but no ecosystem-wide mechanism exists for agents to cryptographically prove their identity and delegated scope to peer agents before a handoff. Research is needed into lightweight attestation formats that travel with inter-agent messages and can be verified without a round-trip to a central authority.

The second gap is **collusion and orchestration drift detection**. Agent collusion via steganographic inter-agent messages and orchestration drift through accumulated shadow logic are both classified as Emerging (E3) threats in the taxonomy because no MCP-specific empirical measurement exists yet. Longitudinal studies of real multi-agent MCP deployments, instrumented with inter-agent provenance tracking and behavioral baselining, would raise the evidence strength of these threats and motivate concrete detection rules.

The third gap is **shadow-MCP governance**. Ungoverned MCP servers introduced inside organizations without central oversight create unmonitored trust boundaries that security teams cannot enumerate or audit OWASP Foundation (2025a). Research into automated shadow-MCP discovery, for example through network traffic analysis or agent handoff graph reconstruction, combined with policy-driven allow-listing, would give enterprises a tractable path to governing agent ecosystems that grow faster than formal review processes.

## 8 Conclusion

Taken together, our findings show that MCP security is governed less by isolated vulnerabilities and more by structured cascades that traverse components and lifecycle phases. By constructing a failure mode oriented taxonomy and mapping each threat to specific components and phases, we turn scattered incidents into a single structured space. Once threats are viewed in this joint space of failure mode, component, and phase, patterns that previously appeared unrelated become instances of the same underlying weakness. Concrete MCP scenarios make this structure visible. A creation time misconfiguration that widens a tool scope, an interaction time prompt manipulation that steers an agent toward that tool, and a maintenance time drift that leaves stale credentials in place may appear as distinct issues, yet they are different manifestations of shared failures in identity, isolation, or integrity rather than separate classes of bugs. This perspective keeps the taxonomy compact while still reflecting the range of behaviors observed in real MCP deployments.

Building on this view, the taxonomy provides a scaffold for a mitigation oriented control plane that spans design, deployment, and operation. For each failure mode, it identifies the components and phases where relatively small changes in configuration, authentication, or isolation can have the greatest security effect. At those control points, the taxonomy is paired with concrete mechanisms and verification artifacts, so that abstract failure modes become specific checks and enforcement actions rather than only descriptive categories. ~~Section 5 shows that this control oriented view can also be realized in a compact benchmark, where representative MCP failures are detected through verifiable checks, localized to the relevant component and lifecycle phase, and tied to explicit audit artifacts.~~ Section 5 provides feasibility evidence that selected taxonomy-derived checks can be made executable and auditable in a controlled MCP deployment, while not validating the full taxonomy, all mitigation categories, or general real-world detection performance. Read together, these elements make scoped capabilities, hardened gateway patterns, sandboxed execution

environments, and structured telemetry appear not as isolated defenses, but as recurring control patterns that can be adapted across deployments while preserving their intended security effect. Because many threats span multiple component and phase cells, no single control is sufficient on its own. Effective enforcement must therefore be layered across configuration validation, pre-deployment checks, and runtime monitoring, which helps constrain the blast radius of individual failures, slows the propagation of cross-phase cascades, and leaves operators with structured evidence for diagnosis and remediation.

This map is intended to give both researchers and practitioners a shared structure for reasoning about MCP threats and controls. For researchers, it clarifies which regions of the MCP threat space remain weakly understood and which combinations of failure mode, component, and phase are not yet covered by existing frameworks and benchmarks. For system architects and operators, it supports systematic reasoning about where to place controls, which forms of evidence to collect, and how to prioritize remediation work when incidents occur. Taken together, these roles mean that the taxonomy and control plane currently operate as a static reference that guides design decisions, structured reviews, and incident analysis. Building on this, a natural next step is to embed the same structure into the infrastructure of MCP deployments so that the same concepts inform automated detection and enforcement. One line of work is the development of telemetry driven analytics that align anomalies in MCP traffic with specific cells in the taxonomy and provide early warning of emerging attacks. A second line of work is the design of integrity enforcement mechanisms that compare expected agent and tool behavior with observed executions and emit explicit violations when workflows deviate from policy. A third line of work is the construction of scalable policy engines that express MCP specific controls in a uniform language and drive configuration, deployment, and runtime enforcement from the same rule base, particularly as recent work explores shared-context collaboration, large-scale tool routing, and governance-aware enterprise data access Jayanti & Han (2026); Yao et al. (2026); Tonnarelli et al. (2026). Pursued together, these directions can move MCP from a loosely structured collection of components toward a security aware environment where threats are mapped, monitored, and constrained using the structure laid out in this survey.

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

# A    Appendix

This appendix provides standalone definitions for each threat in the MCP threat taxonomy. Each subsection aligns with a category defined in Section 3 of the main paper. To avoid redundancy, the dual-axis mappings to components, lifecycle phases, STRIDE, and MAESTRO are not reproduced here, as they are already presented in Tables 4 through 6. This appendix is intended to serve as a reference for readers seeking precise definitions of individual threats beyond the synthesized treatment in Section 3.

## A.1    Identity and Trust Management

This category addresses how MCP deployments name, authenticate, and bind clients, servers, and tools. The absence of a global naming authority and the lack of protocol-level cryptographic identity enforcement mean that clients and agents routinely rely on self-reported metadata to make trust decisions. The four threats below represent the primary failure modes through which this structural weakness is exploited.

### A.1.1 Server Name Collisions and Impersonation

Adversaries register MCP server names that are typographically or visually similar to trusted ones, for example `dropboxuploader` versus `dropbox-uploader`, to redirect users or agents toward attacker-controlled endpoints. During configuration and discovery in ecosystems that rely primarily on human-readable names and descriptions, and that do not enforce verified namespaces, signed metadata, or equivalent provenance checks, clients may bind to a lookalike server instead of the intended one. As a result, names alone are not a reliable basis for distinguishing legitimate servers from impersonating ones without additional trust verification.

**Evidence.** Empirical analyses of MCP-style registries and adjacent plugin ecosystems show that weak submission vetting allows adversaries to publish lookalike packages and endpoints, creating realistic typosquatting and impersonation opportunities that are difficult for both human operators and automated

agents to detect during routine registration and discovery Posta (2025) MCP-specific work identifies tool squatting and name-collision attacks as realistic risks when clients rely on human-readable names, descriptions, or weakly vetted metadata during discovery. ETDI motivates cryptographic identity verification and immutable versioned tool definitions to reduce such risks, while MCP Security Bench evaluates name-collision attacks in an end-to-end benchmark using real benign and malicious MCP tools Bhatt et al. (2025); Zhang et al. (2025).

**Impact.** Once a client or agent is bound to an impersonating server, the adversary can intercept sensitive context, inject malicious tool outputs, and alter execution paths without detection. Because the same client configuration is often reused across multiple agents and hosts, a single successful name collision can propagate as a persistent supply chain foothold across an entire MCP deployment.

### A.1.2 Mutual Authentication and Encryption Gaps

MCP's transport specification defines client–server transports including `stdio` and HTTP-based transports, while its authorization specification standardizes OAuth-based authorization for HTTP transports rather than a universal mutual-authentication model across all client–server modes Model Context Protocol (2025s;l). As a result, authentication, channel security, certificate validation, and client verification remain deployment-sensitive hardening decisions rather than uniformly enforced protocol guarantees. In deployments lacking mutual authentication, strict certificate validation, or equivalent endpoint verification, an attacker who controls a network path or misconfigured gateway may be able to interpose on traffic without protocol-level detection.

**Evidence.** A formal security analysis of the MCP specification finds that it provides no cryptographic authentication between clients and servers at the protocol level. Across 847 controlled attack scenarios on five server implementations, these architectural choices raised attack success rates by 23% to 41% relative to comparable integrations that do not use MCP Maloyan & Namiot (2026).

**Impact.** A client may silently bind to attacker-controlled endpoints, transmit sensitive context, and act on manipulated tool outputs, enabling man-in-the-middle attacks, credential theft, and long-lived workflow hijacking throughout the affected deployment.

### A.1.3 Lack of Verified Server Identity or PKI

MCP does not define a native mechanism for binding servers to cryptographic identities beyond generic HTTPS/TLS and OAuth infrastructure, so clients and registries may rely heavily on human-readable metadata such as names, descriptions, or repository provenance when selecting servers Posta (2025) Model Context Protocol (2025l); Metere (2026). Unlike protocols that enforce a standardized ecosystem-wide identity-binding mechanism for endpoint authenticity, MCP leaves identity verification largely to deployment-specific choices. As a result, environments that rely primarily on metadata-based discovery without stronger provenance checks may remain vulnerable to server impersonation and misleading server listings.

**Evidence.** Security analyses highlight that MCP ecosystems can expose identity-confusion risks when discovery depends on names, descriptions, or other human-readable metadata without stronger provenance verification, so effective identity assurance depends on deployment-specific controls rather than a universal protocol-enforced identity layer Anthropic (2025). The MCP authorization specification standardizes OAuth-based authorization for protected resources, but it does not provide a protocol-level admission check for server trust Model Context Protocol (2025l). Attested-server work addresses this gap by proposing offline-signed server clearance assertions, pinned trust roots, per-server allowlists, and auditable enforcement before tool dispatch Metere (2026).

**Impact.** The absence of a standardized identity-binding layer makes it more difficult to reliably distinguish legitimate from adversarial servers during registration and discovery, increasing the risk of impersonation, workflow hijacking, and long-lived malicious footholds, particularly in decentralized MCP ecosystems where trust verification is inconsistent.

### A.1.4 Implicit Trust in Unauthenticated Servers

Implicit trust in unauthenticated servers becomes risky when MCP tools are exposed beyond local or tightly controlled environments. Recent measurement work finds that 40.55% of 7,973 live remote MCP servers expose tools without authentication, showing that weak authentication is not merely a hypothetical deployment concern Zhou et al. (2026a). When such tools and agents are exposed over HTTP, cloud infrastructure, or shared internal networks without equivalent security hardening, network-reachable services may accept tool invocations without robust client authentication or identity verification.

**Evidence.** CVE-2025-49596 documents a concrete instance of this pattern. In affected versions of MCP Inspector below 0.14.1, authentication was missing between the Inspector client and proxy. As a result, unauthenticated requests could launch MCP commands over `stdio`, leading to remote code execution in the context of the vulnerable process National Vulnerability Database (2025).

**Impact.** Agents that bind to and invoke unverified services can effectively extend their own privileges to those services, allowing malicious or misconfigured servers to trigger privileged tool executions, expand access, and exfiltrate sensitive data while the surrounding workflow continues to appear legitimate.

## A.2 Supply Chain and Code Integrity Risks

MCP's reliance on decentralized, third-party tool ecosystems and ad hoc deployment practices means that many deployments operate without centralized vetting or consistent integrity policies for server binaries, manifests, and libraries. The five threats below represent the primary failure modes through which supply chain and code integrity breakdowns arise.

### A.2.1 Trojanized or Compromised Installers

Tools such as `Smithery-CLI`, `mcp-get`, and `mcp-installer` provide streamlined workflows for discovering, installing, and configuring MCP servers for use with clients, agents, and hosts Smithery (2025); mcp (2025a;b). If an installer binary, its package source, or its execution path is trojanized, the same convenience workflow can become a delivery mechanism for a backdoored MCP server or malicious configuration. A trojanized installer can add malicious server entries or package-based servers under plausible names and persist those changes in local configuration files or installation paths that are reused by clients on subsequent runs. A single privileged run can add rogue server entries, alter local configuration, or install malicious packages that blend into later discovery and execution workflows.

**Evidence.** Public documentation for these installers describes discovery, installation, local client integration, and configuration-management capabilities Smithery (2025); mcp (2025a;b). These documented capabilities confirm that installer-mediated workflows can modify the same persistent configuration surface on which MCP-aware clients later rely.

**Impact.** Any MCP-aware client that relies on the modified configuration risks telemetry exfiltration, credential harvesting via newly installed tools, and staged payloads that execute in later updates or orchestrated workflows. Such compromise can persist across multiple agents, hosts, and projects that share the same installer-managed environment.

### A.2.2 Malicious Dependencies and Backdoored Code

MCP tools distributed through ecosystems such as PyPI and npm inherit the broader software supply-chain risks of those platforms, including dependency hijacking, typosquatting, and backdoor injection into both direct and transitive dependencies. Attackers can exploit dependency chains to deliver malicious code through packages that are installed either directly or indirectly, allowing MCP servers or tools built on those packages to execute attacker-controlled code with the tool's privileges Sejfia & Schäfer (2022). Within an MCP deployment, packaging a server or tool around such backdoored dependencies means that routine tool invocation may silently activate malicious functionality.

**Evidence.** ~~Recent incidents such as the compromise of `rand-user-agent` demonstrate how seemingly innocuous packages can be repurposed into delivery vectors for remote access trojans Toulas 2025~~. Recent

incident analysis of the compromise of `rand-user-agent` illustrates how an apparently ordinary package can be repurposed into a delivery vector for remote access trojans Eriksen, Charlie (2025). Furthermore, Typosquatting and name-confusion cases involving `colorizr` and `colorama` similarly show how small, deceptive naming changes can steer operators toward malicious code ~~Meyer 2025; The Hacker News 2025; Imperva 2024~~ Meyer (2025).

**Impact.** Routine MCP tool calls can become reliable opportunities for remote access, credential theft, or data exfiltration wherever the compromised package or image is reused, even when the tool appears to behave normally from the operator's perspective.

### A.2.3 Lack of Integrity Verification

~~MCP's core protocol does not define a universal, mandatory mechanism for message signing or end-to-end integrity verification across transports and package types, so integrity assurance for tool calls, server artifacts, and server responses depends on transport-specific, registry-specific, and deployment-specific controls rather than a protocol-wide guarantee Strobes Security Labs (2025).~~ MCP defines JSON-RPC message semantics and supports multiple transports, but it does not impose a universal protocol-wide signing mechanism for tool calls, server artifacts, or server responses; consequently, integrity assurance depends on transport security, artifact-signing and provenance mechanisms, registry policy, and deployment-specific controls Model Context Protocol (2025r); Kalu & Davis (2025). Thus, an adversary who can modify a repository, package feed, or other software distribution path may be able to swap or patch MCP server code or alter its behavior without any MCP-specific integrity signal.

**Evidence.** ~~Security guidance recommends signing MCP components, verifying dependency integrity, and using hardened build pipelines with SAST and software composition analysis, treating MCP servers as ordinary executable code whose integrity protections must be implemented around MCP deployments rather than being enforced by the protocol itself Red Hat Product Security (2025).~~ Software-supply-chain security research identifies transparency, validity, and separation as core design properties for resisting compromise across build and distribution pipelines, while software-signing work argues that signing provides provenance, integrity, and accountability across distribution boundaries where registry controls alone may not suffice Okafor et al. (2024); Kalu & Davis (2025). In MCP deployments, these controls must be applied around server artifacts, tool definitions, dependencies, and registries because MCP itself does not provide a universal protocol-wide signing mechanism.

**Impact.** Routine installs, updates, and tool invocations can become software supply-chain or message-integrity attack vectors that propagate across clients and workflows that depend on a compromised server, especially where stronger registry, transport, or build-pipeline controls are absent.

### A.2.4 Vulnerable Redeployments and Stale Manifests

Lost, delayed, or corrupted updates can cause MCP deployments to drift into inconsistent state, leaving different agents or hosts operating on conflicting context or stale tool assumptions. In environments where MCP configurations and tool definitions are reused across developer IDEs, CI systems, and shared agent hosts, incomplete propagation of updates can cause some nodes to continue operating on outdated state while others advance to newer behavior.

**Evidence.** ~~Production-oriented MCP security guidance describes drift arising when updates are lost, delayed, or corrupted, causing agents to operate on conflicting information. The same guidance recommends strict synchronization, integrity checks, and automated conflict resolution to reduce this risk Macvittie 2025.~~ **Evidence.** MCP-specific security work identifies rug-pull attacks in which tool descriptors are altered after approval, motivating descriptor-integrity mechanisms such as manifest signing and version-aware validation Jamshidi et al. (2025). More broadly, empirical studies of software and cloud-system misconfiguration show that configuration inconsistencies, dependency errors, and evolving configuration designs can produce real-world failures and security risks Liu et al. (2024); Zhang et al. (2021).

**Impact.** Hosts and agents may continue invoking tools under stale assumptions or inconsistent state long after a corrective update is issued. An attacker or failure condition that exploits this inconsistency can

therefore affect multiple workflows and nodes before operators recognize that different parts of the deployment are no longer operating on the same effective state.

### A.2.5 Unverified Package Sources and Automatic Invocation

In MCP deployments, hosts can translate LLM-generated outputs into external tool invocations once servers are configured, making server-provided tool metadata part of normal agent workflow selection. Recent ecosystem analyses of several public MCP registries report weak or inconsistent vetting and show that a substantial number of listed servers can be hijacked Li & Gao (2025). When servers from unverified sources are installed, a compromised server can place malicious tools directly in the host's execution path, especially in deployments that auto-execute tools or rely on permissive approval settings.

**Evidence.** We use Li & Gao (2025) as the primary evidence for this threat because it directly connects the relevant chain. Users discover servers through public registries, configure them in MCP hosts, hosts translate LLM-generated outputs into server-provided tool invocations, and weak registry vetting allows listed servers to be hijacked. As supporting practitioner evidence, Birur (2025) reports a scan of 1,000 public MCP servers in which roughly one third exposed at least one critical vulnerability. We treat this as an operational signal rather than systematic prevalence evidence because it measures vulnerability exposure rather than registry trust or automatic invocation.

**Impact.** A single compromised server can exfiltrate chat or backend data or trigger unintended actions during otherwise routine workflows, with the risk amplified when hosts do not require strong confirmation, input review, or result-validation safeguards before tool execution Model Context Protocol (2025q).

### A.3  Malicious or Poisoned Tool Servers

Because MCP clients rely on server-advertised names, schemas, and tool metadata to bind calls, malicious or manipulated definitions can misbind requests to untrusted implementations, inject tainted context, and exploit name or command collisions to bypass policy. The seven threats below represent the primary failure modes through which this binding surface is exploited.

### A.3.1 Poisoned Tool Metadata

Adversaries can embed hidden instructions in MCP tool descriptions or description-bearing docstrings that are surfaced to the LLM during tool discovery and selection, causing these fields to be interpreted as instructions rather than neutral metadata. Poisoned metadata can influence agent behavior by steering the model toward high-privilege actions or covert exfiltration paths. A seemingly harmless `add()` tool whose description instructs the model to read and exfiltrate `~/.ssh/id_rsa` illustrates how a host that auto-registers or insufficiently reviews such a tool can be coerced into leaking credentials during routine completions Invariant Labs (2025).

**Evidence.** Large-scale empirical evidence shows that description-level cues materially shape agent tool selection and that MCP does not enforce consistency between tool descriptions and underlying implementations, allowing benign-looking descriptions to mask materially different behavior Li et al. (2026b).

**Impact.** Hosts or agents that auto-register tools or insufficiently validate tool descriptions can be coerced into leaking SSH keys, configuration files, or API credentials from their local environment during otherwise routine workflows, turning generic tool metadata into a reusable exfiltration channel.

### A.3.2 Tool Shadowing and Name Collisions

When multiple MCP servers expose tools with identical or confusingly similar names, adversaries can exploit string-based discovery and selection logic to place attacker-controlled tools alongside legitimate ones. Deliberate tool shadowing and name collisions can divert invocations intended for trusted tools toward malicious implementations that reuse similar names and descriptions, allowing attackers to hijack critical workflows while mimicking the expected interface.

**Evidence.** ~~Security analysis of MCP tool selection shows that tool name collisions arise when different servers offer identical or similar tool names, and that AI models rely on tool names and descriptions when choosing which tool to invoke. This makes similarly named malicious tools a realistic way to bias selection and redirect execution, especially when hosts aggregate tools from multiple servers without stronger provenance or approval controls Beretta et al. (2025).~~ MCP Security Bench identifies name-collision attacks as part of a systematic taxonomy of MCP-specific attacks and evaluates them in an end-to-end harness that runs real benign and malicious tools through MCP. Its results show that tool metadata, names, and descriptions can be manipulated during tool selection, making similarly named malicious tools a realistic way to bias selection or redirect execution when hosts aggregate tools from multiple servers without stronger provenance or approval controls Zhang et al. (2025).

**Impact.** Data and execution can be silently redirected across workflows even though invocations still appear to be routine tool calls. Attackers may intercept arguments, model outputs, and configuration updates intended for trusted tools, then inject modified artifacts back into the same pipelines.

### A.3.3 Pre-Invocation Code Injection

Malicious MCP tools can embed harmful behavior either in description-level metadata or in unsafe execution paths. Line-jumping attacks place immediate-action directives in tool description fields, allowing connected servers to influence agent behavior or trigger hidden actions before any explicit tool invocation occurs. Separately, tools that interpolate unvalidated parameters directly into shell commands, such as `os.system("notify-send " + info["msg"])`, can turn crafted inputs into command injection and remote code execution during routine invocation.

**Evidence.** ~~Published MCP security analyses describe both patterns. Security guidance on line-jumping shows that malicious instructions embedded in tool description fields can influence agent behavior before any explicit tool invocation occurs Fenstermacher (2025). Separate analysis of insecure MCP server implementations shows that unvalidated shell interpolation can turn ordinary tool inputs into command injection and remote code execution during routine invocation Cross (2025).~~ MCP-ITP shows that malicious instructions embedded in tool metadata can influence agent behavior during registration and tool selection, including cases where the poisoned tool itself is never invoked Li et al. (2026a). Separately, large-scale MCP vulnerability analysis shows that MCP servers expose security-sensitive sinks such as shell execution, network access, and file-system manipulation to agent-driven invocation, and that taint-style flaws can create end-to-end paths from natural-language inputs to confirmed exploit traces Sun et al. (2026).

**Impact.** Compromised tools can leak secrets, modify configuration, or run arbitrary commands during routine agent workflows, sometimes before the user recognizes that a sensitive action has been triggered and sometimes through seemingly normal tool execution.

### A.3.4 Rug Pull via Dynamic Tool Redefinition

A tool that is initially trusted can later be redefined if its server code or tool description is updated in place, while the host continues to resolve the same registered tool. Song et al. describe this as a rug-pull pattern, in which an MCP server first appears benign and is later modified to embed malicious instructions; they also show that malicious servers uploaded to aggregation platforms can trigger harmful actions in users' local environments after installation Song et al. (2025).

**Evidence.** ETDI proposes immutable, versioned, cryptographically signed tool definitions as a direct defense against rug-pull attacks. It further argues that changes to a tool's effective behavior should require a new definition version and renewed approval, indicating that baseline MCP does not provide these protections by default Bhatt et al. (2025).

**Impact.** Previously benign servers can become durable backdoors, allowing attackers to hijack file access, API calls, or transaction flows across hosts and workflows that continue to trust the original registration, even though the tool's effective behavior has changed.

### A.3.5 Preference Manipulation

Attackers can steer tool selection by shaping what the agent sees during discovery through persuasive descriptions that claim greater safety or capability, suggest that a tool is the preferred default, or otherwise increase its relative salience compared with competing tools. Unlike poisoned tool metadata that embeds explicit directives, preference manipulation relies on subtle non-imperative phrasing and selection bias so that the attacker's tool is chosen under ordinary tool-selection behavior. The manipulation operates entirely within the metadata layer exposed to the model during discovery.

**Evidence.** Recent work on MCP Preference Manipulation Attack shows that malicious servers can bias LLM tool selection by modifying tool names and descriptions, including through manipulative phrases and stealthier advertising-style descriptions, causing agents to prioritize attacker-controlled tools over competing MCP servers Wang et al. (2025b). MCP security benchmarks also include preference manipulation as a distinct MCP attack class, reinforcing that biased tool metadata is an evaluation-relevant threat rather than only a practitioner-observed pattern Zhang et al. (2025). Community TTP catalogs and practitioner analyses describe similar metadata-level manipulation patterns, but we treat them as supporting taxonomy and operational guidance rather than primary evidence Model Context Protocol Security (2025); Elastic Security Labs (2025).

**Impact.** The result is a persistent bias toward unsafe tools that increases the likelihood of bypassed safeguards in routine workflows, even without embedding an explicit instruction that simpler prompt-injection filters might detect.

### A.3.6 Command and Function Collisions

When hosts aggregate tools from multiple MCP servers, identical or confusingly similar tool names can create ambiguity during discovery and invocation. In clients that also expose slash-style commands, overlapping commands such as `/query` or `/upload` can similarly route execution to the wrong tool if duplicate registration is allowed. Without stronger disambiguation, provenance cues, or unique registration rules, a malicious tool can shadow a legitimate one by reusing a similar name or interface and thereby divert routine invocations.

~~**Evidence.** Security analyses describe both cross-server tool shadowing and slash-command overlap as realistic MCP attack patterns. Benchmark results show that these attacks can successfully redirect execution to unintended tools, while enterprise security guidance notes that similar functionality or duplicate command registration can confuse tool selection and allow malicious tools to intercept actions intended for trusted ones dts (2025).~~

**Evidence.** MCPSecBench describes tool-centric MCP attacks including cross-server tool shadowing and slash-command overlap, and its benchmark results show that such attacks can redirect execution to unintended tools across tested MCP platforms Yang et al. (2025).

**Impact.** Attackers can intercept intended actions, capture sensitive data, or trigger unauthorized operations by redirecting invocations to malicious handlers. Even without malicious intent, collisions and overlaps can produce information disclosure, privilege misuse, or effective denial of service against the legitimate tool.

### A.3.7 ANSI Escape Code Injection

An attacker can embed ANSI escape sequences in strings rendered to humans, such as a tool's `name` or `description` returned by `tools/list` or text returned by `tools/call`. In terminal-based MCP clients that render these strings without sanitization, the terminal may interpret them as control instructions that hide text, clear the screen, move the cursor, or display deceptive hyperlinks rather than showing the raw content as-is. Unlike poisoned tool metadata, which targets model interpretation, this attack targets the human display layer and exploits the disconnect between what the reviewer sees and what the underlying content actually contains.

**Evidence.** MCP-specific testing showed that ANSI escape sequences in tool descriptions and outputs were not sanitized in Claude Code 0.2.76, allowing malicious payloads to be hidden from users while still influencing the surrounding workflow Hoodlet (2025). Independent CVEs outside the MCP ecosystem also

confirm the general feasibility of ANSI-based screen spoofing, including vulnerabilities that allowed attackers to spoof console output and even create fake file listings or hide genuine content through escape-sequence injection nvd (2024a;b).

**Impact.** Staged warnings, masked prior lines, and misleading screens raise the chance of unsafe approvals or missed indicators during tool or log review, allowing attackers to obtain human authorization for actions that the operator would not approve if the display were accurate.

### A.4  Prompt, Input, and Context Injection Attacks

This class of attacks targets how MCP agents assemble and reuse context at runtime. An agent's working context is constructed from many channels including user prompts, server-supplied templates, retrieved content, local files, API responses, and intermediate tool outputs, and any of these channels can carry hidden instructions. The four threats below represent the primary failure modes through which insecure context assembly is exploited.

### A.4.1 Poisoned Prompt Templates

MCP servers can expose reusable prompt templates that clients discover through `prompts/list` and retrieve through `prompts/get`. Because these prompts are provided by the server as structured templates intended to shape model interactions, a compromised server can embed hidden instructions that are presented as trusted guidance rather than ordinary untrusted user input Model Context Protocol (2025m).

**Evidence.** ~~Public security analysis notes that a malicious server can supply a subtly modified prompt designed to trick the application into revealing sensitive information or misusing tools. Because such prompts are delivered as reusable templates from the server, a single compromised or hijacked server can propagate the same malicious guidance across every host that imports and invokes them Palo Alto Networks (2025).~~ Recent work on chat-template injection shows that adversarially structured templates can manipulate LLM agents more effectively than plain-text prompt injection and can remain effective across models and agent benchmarks Chang et al. (2025). A compromised MCP server that supplies a malicious reusable prompt can therefore propagate the same adversarial guidance across hosts that import and invoke that template.

**Impact.** MCP prompt templates can become high-leverage inputs whose compromise steers assistants toward information leakage, credential exposure, or unsafe tool use wherever the affected server's templates are imported or reused.

### A.4.2 Retrieval-Based Poisoning (RADE)

Adversaries can embed malicious MCP instructions inside files or online content that retrieval tools ingest, so the LLM later treats them as benign context and follows them as part of an otherwise ordinary query. Retrieval-Agent DEception (RADE) shows that when a crafted file is added to a vector database, a model can retrieve attacker-written instructions and then use connected MCP tools to add SSH keys, search for secrets, or exfiltrate credentials. Follow-on work further broadens this threat model by showing that attackers may only need to post malicious content online if MCP-enabled scraping and retrieval pipelines ingest it automatically Radosevich & Halloran (2025).

**Evidence.** In the original RADE demonstrations, Claude used the Chroma MCP server to retrieve malicious content from a vector database and then used filesystem and Slack-related tools to add attacker SSH keys and exfiltrate API keys. Subsequent TRADE experiments showed that similar retrieval-based attacks can succeed even when the payload is hosted online rather than downloaded directly, and that existing refusal guardrails often remain weak against these falsely benign retrieval exploits Halloran et al. (2025).

**Impact.** A single compromised document, indexed file, or ingested web page can escalate into system-level data exfiltration or unauthorized access across workflows that connect retrieval servers to high-privilege tools, with the payload executing as a natural consequence of normal information retrieval.

### A.4.3 Transitive-Trust Poisoning (Fourth-Party Injection)

Transitive trust poisoning arises when a trusted MCP server retrieves content from external systems and forwards it into the model context as trusted contextual input. If that upstream content contains hidden instructions, the server can become an unwitting relay that launders attacker-controlled text across a trust boundary the model treats as higher-confidence than ordinary user input. ~~This risk remains even when deployments add partial mitigations, because retrieved content may still influence downstream model behavior if sanitization or review is incomplete Supabase (2025).~~ This risk remains even when deployments add partial mitigations, because hidden adversarial prompts in external resources can be relayed through MCP-enabled tools and influence downstream model behavior, including malicious content relay and sensitive-data leakage scenarios Xiong et al. (2025).

**Evidence.** In a publicly discussed Supabase MCP scenario integrated with Cursor, the assistant operated with `service_role` privileges while reading customer-submitted ticket content. A single crafted support ticket could induce the agent to read a sensitive table such as `integration_tokens` and write the results back into the same ticket, turning retrieved upstream content into an exfiltration path General Analysis (2025).

**Impact.** A single poisoned upstream message can become a channel for sensitive-data exfiltration and unsafe tool use in workflows that rely on the same trusted retrieval path, even when the MCP infrastructure itself has not been directly compromised.

### A.4.4 Format-Embedded Poisoning (Weaponized Content Channels)

Format-embedded poisoning arises when MCP clients surface retrieved documents, web content, chat messages, or tool outputs directly into the model context, allowing attackers to hide model-interpretable instructions inside content that appears to be ordinary data. The attack relies on the fact that once such content is passed through a trusted retrieval or messaging path, the model may interpret embedded instructions as part of the task context rather than as inert payload.

**Evidence.** Public MCP guidance warns that servers fetching untrusted content expose clients to prompt-injection risk Anthropic (2025). In MCP-specific demonstrations, a crafted file indexed through the Chroma MCP server caused Claude to retrieve attacker-controlled instructions and then use other connected tools to add SSH keys and exfiltrate API keys Radosevich & Halloran (2025). ~~Separate experiments showed that a single injected WhatsApp message returned through a trusted MCP integration could compromise the agent and exfiltrate sensitive contact information, even without installing a separate malicious server Beurer Kellner & Fischer (2025).~~ More broadly, MCP-specific work on parasitic toolchain attacks shows that malicious instructions embedded in external data sources can propagate through legitimate MCP workflows, trigger sensitive tool invocations, collect private data, and disclose it to attacker-controlled endpoints Zhao et al. (2025a).

**Impact.** A poisoned document, message, or retrieved page can become a portable attack payload across workflows that ingest the same content channel, turning ordinary retrieval into a path for data exfiltration and unsafe follow-on tool use.

### A.5 Credential Leakage and Data Exposure

Where the preceding categories focus on trust establishment, code integrity, and unsafe context assembly, this section turns to what MCP workflows can unintentionally reveal once those mechanisms begin operating across real services. MCP servers often function as high-privilege intermediaries that hold tokens, relay content, and combine outputs from multiple tools, so exposure can arise not only from direct secret theft but also from logging, forwarding, retention, and cross-system aggregation. The threats collected here therefore capture the main ways credentials and sensitive data escape their intended scope in MCP deployments.

### A.5.1 Centralized Credential Storage

MCP servers often consolidate API keys, OAuth tokens, and other service credentials in a single configuration file or secret store, so compromising that host or backing credential store can give an attacker valid bearer tokens for every connected integration. The centralized nature of the failure means that a single breach can cascade across multiple integrations rather than requiring separate compromise of each downstream service.

**Evidence.** ~~Public security analysis of MCP deployments describes servers that retain long-lived OAuth tokens for email and enterprise services, and warns that a breached server can reuse those tokens to perform arbitrary tool actions and API calls that appear indistinguishable from legitimate agent activity. The same analysis also notes that MCP servers are high-value targets because they often store authentication tokens for multiple services at once Pillar Security (2025).~~ OAuth bearer-token guidance treats possession of a bearer token as sufficient to access protected resources, so tokens stored by an MCP server become reusable capabilities if the host or backing credential store is compromised Jones & Hardt (2012). MCP authorization guidance further warns that servers mediating access to downstream APIs must avoid token passthrough and validate token audiences, because failures in these checks can let one compromised integration reach services beyond its intended scope MCP Working Group (2025b).

**Impact.** Once attackers control these credentials, downstream APIs may treat their traffic as ordinary MCP usage, allowing them to exfiltrate data, send or delete messages, and invoke administrative operations across multiple connected services until credentials are rotated or revoked, with blast radius proportional to the number of services consolidated behind that store.

### A.5.2 Insecure Secret Management

In MCP deployments, secrets such as API keys and OAuth tokens can leak through operational choices such as hard-coded configuration files or other insecure credential-handling practices. Because MCP servers often mediate access to multiple external services at once, a single credential exposure can grant attackers broad access across connected integrations.

**Evidence.** ~~Public security analysis of MCP describes how servers often store API keys and may retain access to multiple external services simultaneously, which magnifies the damage of a single leak. The same guidance recommends secret scanning for configuration files and using environment variables or dedicated secret-management solutions instead of hard-coded credentials Palo Alto Networks (2025)~~ Cleartext credential storage and insertion of sensitive information into logs are recognized software weaknesses that can expose API keys, OAuth tokens, and other bearer credentials MITRE (2025a;b). In MCP specifically, a measurement study of real-world remote MCP servers finds widespread OAuth deployment flaws and shows that these weaknesses can lead to sensitive information leakage and account takeover Zhou et al. (2026a).

**Impact.** Once exposed, these bearer tokens can allow attackers to issue legitimate-looking calls to downstream services on behalf of the MCP host or agent until revocation and rotation fully propagate.

### A.5.3 Overprivileged and Persistent Tokens

Tools and servers may request broad permission scopes for external services and rely on OAuth tokens that can remain valid beyond a single interactive session. Even when secrets are not directly exposed, the breadth of the granted scope means that a single stolen token may authorize actions across multiple connected services simultaneously.

**Evidence.** OAuth security guidance recommends that bearer tokens carry least privilege scopes and short lifetimes, with refresh token rotation, precisely to limit the damage of overly broad or persistent grants Lodderstedt et al. (2025). Consistent with this, OAuth incident reporting shows that malicious or unauthorized applications can retain cloud access even after password resets or multifactor authentication changes, because the authorized application grant remains valid until it is explicitly revoked Proofpoint Threat Research (2025).

**Impact.** A compromised overprivileged token becomes a long-lived capability that lets an adversary operate across multiple tools and workflows on behalf of the MCP host or agent until rotation and revocation fully

complete, with the scope of the token determining the full extent of actions the attacker can perform without triggering an additional authorization step.

### A.5.4 Cross-Tool Data Flows

In complex MCP agent pipelines, a single hidden directive can steer data from internal tools, retrieved resources, or conversation history toward external endpoints within one multi-step interaction. The base MCP specification standardizes how hosts, clients, and servers exchange tools, resources, prompts, and sampling requests, but leaves security controls to implementors rather than defining protocol-level provenance tracking or data-flow restrictions across chained tool use.

**Evidence.** ~~In a documented indirect prompt injection scenario, an assistant retrieved a compromised webpage and was then induced to URL-encode the user's conversation history and send it as a query parameter to an attacker-controlled domain Palo Alto Networks Unit 42 (2025).~~ Empirical work on prompt-injection data leakage shows that tool-calling agents can be induced to disclose personal data observed during task execution, especially in data-extraction and authorization-style workflows Alizadeh et al. (2025). MCP-specific practitioner threat analysis further describes a composability-chaining pattern in which a seemingly benign MCP server forwards a tool request to a second hidden server, merges the returned malicious instructions into the response, and causes the model to exfiltrate sensitive environment variables within the same session CyberArk Threat Research (2025).

**Impact.** Cross-tool data flows can silently move private conversations, local files, and environment-scoped secrets from internal agents and hosts to attacker endpoints. Because each individual tool call may still look legitimate in isolation, per-call monitoring alone may miss the larger exfiltration sequence.

### A.5.5 Trojanized MCP Servers

A compromised or malicious MCP server can observe and modify tool traffic, including user queries, tool parameters, file contents, and credential-bearing responses, while continuing to present itself as a legitimate endpoint. Because agent workflows may rely on server responses as contextual input for subsequent reasoning and tool use, a trojanized server can occupy a central position in the agent's decision loop and influence downstream behavior across the workflow.

**Evidence.** Recent work on malicious MCP servers shows that servers themselves can act as attackers rather than passive infrastructure. In real host-LLM settings, proof-of-concept malicious servers can manipulate MCP components while still appearing to the client as ordinary integrations Zhao et al. (2025b). Complementary work on trivial trojans shows that even a benign-looking MCP server can exploit cross-server trust to exfiltrate sensitive data through otherwise legitimate tool workflows Croce & South (2025).

**Impact.** A single trojanized server can become both a credential-theft pivot and a workflow-poisoning point, enabling cascading data leakage and unauthorized actions across the workflows that depend on that server.

### A.5.6 Emergent Aggregation Risks

In MCP deployments that connect multiple servers at once, agents can accumulate low-sensitivity fragments such as calendar entries, file names, and chat metadata into a shared model context even when each individual tool call appears benign. Because MCP enables a single assistant to access multiple external tools and data sources through one workflow, outputs from email, file, database, and calendar servers can be correlated in the same reasoning loop. The protocol leaves privacy and access-control safeguards to implementors rather than defining protocol-level limits on how such cross-source outputs may be combined.

**Evidence.** MCP governance work identifies data-driven exfiltration and cross-system privilege risks as consequences of connecting agents to multiple tools and data sources, and argues for provenance tracking, input/output checks, DLP, and audit logging to detect unsafe data movement across workflows Errico et al. (2025). More generally, work on privacy aggregation and the mosaic effect shows that records, metadata, or external queries that appear low-risk in isolation can reveal identities, routines, sensitive attributes, or private task intent once aggregated Gurung et al. (2026); Solove (2004). Figure 10 illustrates this directly,

where aggregating low-risk outputs from email, calendar, and file servers exposes sensitive project details that no single call would reveal.

**Impact.** Because each invocation may reveal only a small fragment, emergent leaks are difficult to detect through per-call review even when the accumulated context becomes sensitive. An attacker or over-privileged agent that can observe partial results from several tools may still reconstruct detailed organizational maps, schedules, or user profiles from the combined context.

### A.5.7 Residual Data Exposure

Within MCP deployments, tool inputs, outputs, or credentials may persist in host-side storage artifacts such as logs, caches, temporary files, or crash dumps after a session ends. If these artifacts are not explicitly sanitized, sensitive values intended to be ephemeral can remain recoverable on disk long after the original interaction has completed.

**Evidence.** Guidance on media sanitization explains that ordinary deletion or reformatting may leave residual data that can still be reconstructed unless explicit clearing or purging is applied Kissel et al. (2014). Operating-system guidance for encryption tools similarly warns that crash dump files can contain cached passwords, encryption keys, and the contents of sensitive files from RAM VeraCrypt Project (2015).

**Impact.** An attacker who later gains access to the host or its retained storage artifacts may be able to recover prior tool inputs, outputs, or credentials that were expected to be transient, even without access to the original MCP session.

### A.6 Authorization and Access Control Gaps

Authorization failures in MCP rarely arise from the absence of credentials alone. More often, they stem from permissions that are too coarse, too static, or too weakly revalidated as workflows evolve. Even when tokens, approvals, and annotations are present, enforcement can still break down when policy remains binary, contextual risk is ignored, consent becomes routine, or different servers apply incompatible access rules. Taken together, the threats in this section show how least privilege erodes in practice as MCP deployments become longer-lived, more automated, and more distributed.

### A.6.1 Binary Permissions

In many MCP deployments, authorization remains coarse-grained, and a bearer token or equivalent approval can grant broad access to all tools exposed by a server without finer-grained controls over which specific actions may be invoked. This all-or-nothing model leaves little room for intermediate permission levels such as allowing read access while blocking writes or administrative operations.

**Evidence.** Practical hardening guidance describes traditional MCP implementations as relying on static API tokens that provide binary, unrestricted access to all available tools, violating least privilege Nikhil (2025). Enterprise security work correspondingly argues for fine-grained, scoped access tokens, per-request authorization, and role- or context-aware controls, indicating that these protections are not provided by baseline MCP deployments by default Narajala & Habler (2025).

**Impact.** An attacker who compromises an overprivileged tool token can gain broad control over data access, state changes, and destructive operations reachable through the approved tool surface. A single leaked credential may therefore be sufficient to exercise the full capability set exposed to that token until revocation, rotation, or additional authorization checks take effect.

### A.6.2 Lack of Risk-Aware Differentiation

MCP does not define a mandatory mechanism for separating high-sensitivity tool calls from low-risk ones. The specification treats `ToolAnnotations` such as `readOnlyHint` and `destructiveHint` as advisory meta-data rather than enforceable security controls, and the documentation warns that clients should never make security-critical decisions based solely on these annotations Anthropic (2025c).

**Evidence.** Recent MCP security frameworks argue that descriptive metadata is insufficient on its own and recommend Zero Trust style controls such as fine-grained, scoped access tokens, just-in-time access, context-aware access decisions, and per-request authorization to gate each interaction Narajala & Habler (2025).

**Impact.** In deployments that do not add such extra-protocol safeguards, destructive actions may be exposed through the same tool-selection and invocation flow as benign read-only operations, increasing the chance that a misrouted call, compromised agent flow, or weak approval path leads to high-impact changes Anthropic (2025c); Narajala & Habler (2025).

### A.6.3 Consent Fatigue and Automation Drift

Repeated permission prompts for MCP tools can desensitize users over time, causing them to click through approvals or skim long, complex requests instead of carefully reviewing what a tool is about to do. In deployments where previously approved tools are allowed to run without renewed scrutiny, the intended human-in-the-loop safeguard can gradually weaken and become easier for attackers to bypass.

**Evidence.** ~~One published MCP threat analysis describes a sampling-based attack in which a malicious server hides exfiltration instructions inside a long, innocent-looking story prompt and relies on a user who no longer reads the request or response carefully. The same analysis assumes that a shell-capable tool had already been approved for constant use under an "Always allow" setting, allowing the attacker to bypass meaningful review and extract environment variables through the generated output CyberArk Threat Research (2025).~~ MCP practitioner threat analysis describes a sampling-based attack in which a malicious server hides exfiltration instructions inside a long, innocent-looking story prompt and relies on weak review of generated content. The scenario assumes that a shell-capable tool had already been approved for constant use under an "Always allow" setting, allowing the attacker to extract environment variables through the generated output CyberArk Threat Research (2025). Complementary empirical work on MCP tool poisoning shows that agents can be induced to perform unauthorized operations through legitimate tools, with MCPTox evaluating 1,312 malicious test cases across 45 real-world MCP servers and 353 tools Wang et al. (2025a).

**Impact.** Sensitive actions such as exfiltrating environment variables or invoking shell-capable tools can occur without fresh user scrutiny. In workflows that reuse stale approvals or permissive trust settings, the human oversight layer that MCP depends on for safety can be substantially weakened.

### A.6.4 Confused Deputy and Role Leakage

In multi-server or delegated MCP deployments, an agent or intermediary server with higher privileges can be induced to execute actions on behalf of a less-privileged requester when user identity, authorization scope, or token audience are not tightly bound to each request. Where tool handoffs occur frequently and automatically, each improperly scoped delegation can become a path for privilege escalation.

**Evidence.** The MCP authorization specification explicitly warns that MCP servers acting as intermediaries to third-party APIs can be abused through token passthrough and failures of token audience validation, creating a confused-deputy risk when downstream actions are performed under the wrong authority MCP Working Group (2025b). Separate empirical audits of MCP-enabled agents show that large language models can be induced to chain tools and servers to perform remote access, credential theft, and other unauthorized actions within a single workflow Radosevich & Halloran (2025).

**Impact.** Coarse session-wide grants, shared tool contexts, and missing per-request scoping can allow nominally low-sensitivity requests to transitively exercise higher-privilege capabilities under the agent's or server's effective identity, enabling silent privilege escalation, unauthorized data access, and side effects that are difficult to attribute from individual tool calls alone.

### A.6.5 Decentralized and Inconsistent Policy Enforcement

Each MCP server in a deployment can independently define and enforce its own OAuth settings, tool allow-lists, logging rules, and update policies. Although an MCP host may enforce local security policies for

the clients it manages, the protocol does not standardize a fleet-wide mechanism for keeping these controls aligned across independently deployed servers. As a result, a vulnerability discovered and patched on one server may remain exploitable on others if policy and update state drift across the deployment.

**Evidence.** Official MCP architecture documentation describes hosts as managing separate client connections to individual servers Model Context Protocol (2025f). MCP governance work identifies this decentralization as a source of inconsistent authorization, provenance, auditability, and policy enforcement across tools and workflows, motivating centralized governance through private registries, gateway layers, scoped authorization, inline policy enforcement, and end-to-end auditability Errico et al. (2025). Related secure-MCP work similarly argues for unified identity management, security-context propagation, fine-grained policy enforcement, and comprehensive audit logging across MCP interactions Hou et al. (2026a).

**Impact.** Two servers that expose the same nominal tool may still enforce materially different protections, so compromising the weakest instance can give an adversary capabilities that users incorrectly assume are governed by a uniform security baseline across the broader deployment.

### A.6.6 No Context-Aware Revalidation

Clients obtain OAuth-based access tokens for MCP servers and then reuse those tokens for subsequent tool and resource calls. The MCP authorization specification standardizes token discovery, issuance, validation, and runtime scope step-up for HTTP-based transports, but it does not define an intent-aware or continuous authorization mechanism that rebinds each call to changing workflow context after a valid bearer token has been issued MCP Working Group (2025a).

**Evidence.** Recent work on secure delegation for autonomous agents argues that bearer-style OAuth and JWT tokens satisfy identity checks at issuance time but remain possession-based thereafter, so any party that holds the token can continue issuing API calls until expiry. That analysis further notes that resource servers cannot distinguish authorized intent from unauthorized execution unless tokens are augmented with intent, delegation, or per-agent binding information Goswami (2025).

**Impact.** A token granted in a low-risk or tightly supervised context can continue to authorize sensitive MCP tool invocations under materially different runtime conditions, yielding durable access that is difficult to constrain once the workflow, agent state, or surrounding trust context has changed.

### A.6.7 Uneven RBAC Support

MCP leaves detailed authorization design to individual servers. The authorization specification standardizes transport-level OAuth for HTTP-based transports, but it does not define shared scopes or an interoperable RBAC model across servers, so a role or permission enforced on one server need not have any corresponding meaning on another Model Context Protocol (2025l).

**Evidence.** Recent MCP security work argues that static OAuth scopes are insufficient for fine-grained tool governance and proposes explicit permission management and policy-based access control in which tool capabilities are evaluated against dedicated policies and runtime context Bhatt et al. (2025). This supports treating RBAC and related policy mechanisms as server- or deployment-level controls rather than interoperable semantics guaranteed by the base MCP authorization specification. As a result, server-by-server authorization design can produce fragmented permission models and make least-privilege reasoning harder across multi-server deployments.

**Impact.** Fragmented and inconsistent authorization makes it difficult to understand and audit an agent's effective capabilities across tools and hosts, and an agent operating across multiple servers may have materially different permissions on each one without any operator-visible summary of the combined capability surface.

### A.6.8 Orphaned Sessions and Credentials

Hosts and servers may shut down, restart, or be decommissioned while leaving access tokens, API keys, or other session identifiers valid if those credentials are not explicitly rotated, expired, or invalidated during

teardown. In environments where MCP services are frequently restarted or redeployed, retained credentials can outlive the process or session that originally obtained them and remain usable until their normal expiry or explicit revocation.

**Evidence.** The MCP authorization specification warns that attackers who obtain tokens stored by the client, or cached or logged on the server, can access protected resources with requests that appear legitimate to the resource server, and it explicitly recommends short-lived access tokens to reduce the impact of leaked credentials Model Context Protocol (2025l).

**Impact.** Any attacker or previously authorized client that retains those credentials may continue calling MCP servers and downstream services after restart or teardown until expiry, rotation, or revocation takes effect, turning a temporary deployment lifecycle event into a longer-lived access path.

### A.7 Execution Environment and Sandbox Risks

Once an MCP workflow crosses from reasoning into execution, the security question is no longer only what the model selects, but what the runtime is actually allowed to do. Tools that execute with broad filesystem access, unrestricted network reach, shared infrastructure, or weak isolation can turn prompt-level mistakes and server-side compromise into host takeover, lateral movement, or resource exhaustion. The threats in this section therefore center on the containment boundary itself, including how execution is isolated, what privileges it inherits, and how far a failure can propagate beyond the initial tool invocation.

#### A.7.1 Lack of Execution Isolation

In the `stdio` transport, the client launches the MCP server as a subprocess on the same machine and communicates with it over standard input and output. MCP does not require sandboxing or resource confinement for such local servers, so absent additional controls they execute with the privileges of the client process rather than inside a mandatory containment boundary.

**Evidence.** ~~Public security guidance notes that local MCP servers can execute commands with client privileges and typically run without strong isolation unless operators add containerization, sandboxing, or similar host-level controls Red Hat Product Security (2025).~~ **Evidence.** MCP governance work argues that unvetted MCP server code should not run outside a sandbox and recommends containerized sandboxing, input/output checks, inline policy enforcement, and centralized governance to constrain tool execution and data access Errico et al. (2025). The official MCP transport specification also distinguishes local `stdio` servers, which are launched by the client as local processes, from remote HTTP transports Model Context Protocol (2025r).

**Impact.** Compromise of an unsandboxed local server can yield host-level consequences, including arbitrary command execution, sensitive file access, data exfiltration, and persistent effects on workflows or agents that depend on that server.

#### A.7.2 Sandbox Escape Vulnerabilities

Even when MCP servers run inside containers or virtual machines, they still depend on the underlying container runtime or hypervisor for isolation. A vulnerability or escape in that lower layer can therefore let a compromised server interact with host resources outside its intended sandbox, and the severity of the failure grows when multiple workflows share the same host or runtime.

**Evidence.** Analyses of container-escape vulnerabilities such as CVE-2019-5736 in `runc` show that crafted workloads can overwrite host binaries and obtain code execution on the host National Vulnerability Database (2019). Separately, MCP safety audits show that once tools can reach the host environment, LLM-driven agents can be induced to inject malicious commands into system files and exfiltrate API keys or other secrets from environment variables and configuration files Radosevich & Halloran (2025).

**Impact.** A single server compromise can escalate into a host-level breach that enables persistence and affects multiple MCP workflows, and every workflow sharing the same container runtime inherits the blast radius of the escape.

### A.7.3 Root or Elevated Execution

Running tools and helper processes with root or Administrator privileges removes the operating system privilege separation that would otherwise limit the impact of a compromise. As a result, any successful exploit or prompt-driven misuse of a tool can escalate directly into host-level control without an intermediate containment boundary.

**Evidence.** Official MCP security guidance warns that local MCP servers run with the same privileges as the client and recommends sandboxing them with minimal default privileges Model Context Protocol (2025o). Public advisories for CVE-2025-49596 further show that MCP Inspector versions below 0.14.1 allowed unauthenticated requests to launch MCP commands over `stdio`, resulting in remote code execution in the context of the Inspector process National Vulnerability Database (2025).

**Impact.** If the affected process runs with elevated privileges, a single exploit can yield broad host-level control, including arbitrary command execution, access to sensitive files and configuration, and the ability to alter local artifacts in ways that hinder detection and recovery.

### A.7.4 Prompt-Based Remote Code Execution

An agent that can invoke shell or scripting tools can be driven by prompt injection to issue arbitrary operating system commands, because the model may treat injected tool requests as actionable instructions rather than as untrusted input. When shell-capable MCP tools are available, a single successful prompt injection can therefore translate directly into command execution on the host.

**Evidence.** An MCP safety study demonstrates a proof-of-concept attack in which an injected prompt causes the agent to append a netcat command to the user's shell run configuration file, creating a reverse shell that connects back to an attacker-controlled host whenever a new terminal is opened Radosevich & Halloran (2025).

**Impact.** A single successful prompt injection into an agent with shell access can enable persistent backdoor installation, credential theft from local files and environment variables, and further compromise of workflows that rely on the affected MCP tooling.

### A.7.5 DoS and Resource Exhaustion

Although the MCP specification requires servers to rate limit tool invocations, it does not define a shared quota model, default thresholds, or a fleet-wide enforcement mechanism across deployments. A misconfigured or malicious agent can therefore still repeatedly write large files, flood external APIs, or spawn CPU-heavy tasks until host resources or downstream services degrade or fail if server-side limits are weak or inconsistently implemented. In shared MCP deployments, resource exhaustion triggered by one agent can degrade availability for other agents and workflows that depend on the same server or infrastructure.

**Evidence.** The MCP tools specification requires servers to rate limit tool invocations, while operational hardening guidance recommends implementing quotas and rate limits per user, agent, or session together with timeouts and throttling for costly operations Model Context Protocol (2025q).

**Impact.** A single faulty workflow or compromised agent can still exhaust CPU, memory, or third-party quotas and cause denial of service for other workloads when limits are weak or inconsistently enforced. Because quota design and operational thresholds are left to implementers, these protections must still be configured and maintained by each MCP server operator.

### A.7.6 Vulnerable Third-Party Dependencies

MCP security guidance accordingly recommends software composition analysis, dependency verification, and hardened build pipelines because vulnerable dependencies become part of the effective attack surface of the server itself Red Hat Product Security (2025). A large-scale empirical study of open-source MCP servers finds that conventional software vulnerabilities remain present in MCP server implementations, showing that dependency and code-level weaknesses are part of the effective attack surface of MCP deployments Hasan

et al. (2025). Accordingly, conventional software-supply-chain controls such as software composition analysis, dependency-integrity verification, and hardened build pipelines remain relevant for MCP server deployments.

**Evidence.** Widely deployed parsing libraries have repeatedly shown that attacker-controlled content can trigger serious compromise when unsafe parsing paths are enabled. For example, SnakeYAML's `Constructor()` allowed attacker-supplied YAML deserialization to lead to remote code execution, and PyYAML versions before 5.3.1 and 5.4 were likewise vulnerable to arbitrary code execution when processing untrusted YAML through unsafe loading paths National Vulnerability Database (2022b; 2020b;a).

**Impact.** A single vulnerable dependency can turn an otherwise routine fetch-and-parse operation into remote code execution on the MCP server host, and because these flaws may reside in transitive dependencies several layers below the application code, the resulting compromise can affect every workflow that relies on the affected tool.

### A.7.7 Lateral Movement via Shared Infrastructure

In shared MCP deployments, multiple servers may run on the same hosts or virtual networks and may centralize OAuth tokens or API credentials for connected services. If one server or its credential store is compromised, an attacker can reuse those tokens and existing network reach to access other connected services or pivot through the same trust boundary.

**Evidence.** Public security analysis describes MCP servers as high-value targets because they often store authentication tokens for multiple services at once. Official MCP security guidance likewise warns that if a token is accepted by multiple services without proper validation, an attacker compromising one service can use that token to access other connected services Model Context Protocol (2025o).

**Impact.** A single server compromise in a shared deployment can expand into lateral access across connected services, enabling unauthorized tool use, token reuse, and data exfiltration well beyond the originally breached integration.

## A.8 Workflow and Configuration Risks

Even when tools are trusted and execution is contained, MCP workflows remain vulnerable if operational state stops being consistent across steps, sessions, and servers. Small mismatches in context, configuration, or teardown can accumulate until the workflow that actually runs is no longer the one users and operators intended. This section focuses on the failures of continuity and control that make that divergence possible.

### A.8.1 Context Drift and State Inconsistency

Multi-step MCP workflows rely on consistent propagation and update of shared state across tools and servers, so when an early step is delayed, lost, corrupted, or applied inconsistently, later steps may continue from an outdated or conflicting view of the world. Small local inconsistencies can therefore accumulate into broader workflow failures that may only become visible after downstream tools have already acted on stale context.

**Evidence.** Work on context drift in LLMs shows that model behavior can diverge over time as surrounding context departs from a stable, goal-consistent reference Wu et al. (2025); Dongre et al. (2025). Related work on multi-agent planning frameworks argues that transaction guarantees, validation, and context management are necessary because current systems struggle with inconsistent shared state, context loss, and failures that require later repair Chang & Geng (2025). ~~MCP-specific operational guidance likewise warns that lost, delayed, or corrupted updates can cause agents to operate on conflicting information Macvittie (2025).~~

**Impact.** In MCP deployments that compose models into long-running workflows, inconsistencies and contextual drift can accumulate across steps and agents, causing downstream tools to act on outdated or contradictory information and turning operational failures into broader integrity and reliability problems.

### A.8.2 Configuration Drift Across Servers

Enterprise MCP deployments can accumulate many server instances with small divergences in authentication settings, logging, allowed scopes, and dependency versions, so uneven maintenance and decentralized administration may leave some nodes substantially weaker than others. As a result, a vulnerability mitigated or remediated on one server may remain exploitable on another if security posture drifts across the fleet.

**Evidence.** Practitioner reports suggest that the public MCP ecosystem expanded rapidly by May 2025; one industry assessment identified more than 13,000 publicly visible MCP-related GitHub repositories or servers and characterized many deployments as lightly governed or insufficiently authenticated Prompt Security Team (2025). We treat this source as an operational signal rather than systematic empirical evidence. Stronger empirical support comes from a large-scale analysis of 1,899 open-source MCP servers, which finds that 7.2% contain general vulnerabilities and 5.5% exhibit MCP-specific tool poisoning, indicating that materially different security postures persist across publicly available MCP servers Hasan et al. (2025).

**Impact.** Once even a single weak or poisoned server remains inside an agent's trust boundary, an attacker can use that outlier as an entry point to harvest secrets, misroute tool calls, or compromise workflows that implicitly treat servers across the deployment as equivalently trustworthy.

### A.8.3 Transactional Integrity Gaps

HTTP-based MCP interactions send each client JSON-RPC message as a separate HTTP POST request, and although the transport can support sessions and streaming, the protocol does not define atomic multi-step transaction semantics or built-in rollback across related calls ~~Knostic Team (2025)~~ Model Context Protocol (2025r); Morley (2010). As a result, recovery from partial failure is left to higher-level workflow logic rather than guaranteed by the protocol itself Li et al. (2025a).

**Evidence.** ~~Practitioner guidance places rollback for suspicious tool-call sequences at the workflow layer rather than the protocol layer Network Intelligence (2025).~~ Recent work on trustworthy and reliable agent systems argues that multi-step tool-using agents need transactional execution, write-ahead logging, rollback, compensation, and conflict resolution above the base tool-call interface to preserve consistency under failures or concurrent execution Li et al. (2025a); Mohammadi et al. (2026). Related work on multi-agent LLM planning likewise argues that current systems lack transactional safeguards and can enter inconsistent states or partial-failure conditions unless additional compensation, validation, and recovery logic is added above the base interaction layer Chang & Geng (2025).

**Impact.** In MCP-based enterprise automations that span multiple services, a misaligned or aborted run can leave one subsystem updated while others remain unchanged, creating integrity and policy violations that may only be detected after the workflow has completed.

### A.8.4 Patch Gaps and Vulnerable Instances

MCP servers often reuse reference implementations, connectors, and community-maintained packages, so when vulnerabilities appear in widely copied code, unmaintained forks can quietly retain exploitable behavior across registries and internal deployments. The same forking model that makes MCP tools easy to customize and redistribute also increases the chance that insecure code will persist in downstream copies long after the original project is archived or no longer maintained.

**Evidence.** ~~Recent reporting describes a SQL injection flaw in Anthropic's SQLite MCP server that had been forked or copied more than 5,000 times before the repository was archived, with no official patch planned and downstream maintainers left to audit and fix their own copies Park (2025); Park (2025).~~

**Evidence.** Recent security reporting describes a SQL injection flaw in Anthropic's SQLite MCP server that had been forked or copied more than 5,000 times before the repository was archived, with no official patch planned and downstream maintainers left to audit and remediate their own copies Park (2025); Lyons (2025). We use this example to illustrate patch-propagation risk in widely reused MCP servers rather than as systematic evidence of ecosystem-wide prevalence.

**Impact.** ~~A single vulnerable MCP server can leave many downstream agents and workflows exposed to stored prompt injection, data exfiltration, and workflow hijacking even after the original project has been deprecated. Because remediation depends on each fork maintainer acting independently, vulnerable copies can persist across the ecosystem long after the original disclosure.~~

**Impact.** A widely forked vulnerable MCP server can leave downstream agents and workflows exposed to stored prompt injection, data exfiltration, or workflow hijacking even after the original project has been deprecated. Because remediation depends on downstream maintainers acting independently, vulnerable copies can persist after disclosure when patch responsibility is decentralized.

### A.8.5 Session Isolation in Multi-Agent Environments

Shared MCP servers that host multiple agents or tenants can implement sessions with weak boundaries, so an attacker who can guess, replay, or intercept a session identifier may be able to impersonate another client or cause state intended for one user or agent to be delivered into a different active session. In multi-agent systems where context is handed off automatically across components, a single hijacked session can therefore contaminate downstream agent behavior if later steps trust the injected or misrouted state.

**Evidence.** MCP security guidance explicitly describes both session-hijack impersonation and session-hijack prompt-injection scenarios in stateful HTTP deployments. It warns that servers must not use sessions for authentication, must verify inbound requests, should bind session identifiers to user-specific information, and must not associate state with session identifiers alone, precisely because weak session handling can enable unauthorized actions and cross-user state confusion Model Context Protocol (2025j).

**Impact.** A compromised session can expose private tools, prompts, and retrieved state across workflows that share the affected server, leading to unauthorized actions and disclosure of user-specific context well beyond the original interaction.

### A.8.6 Unsafe Fallbacks and Error Handling

When MCP tool calls fail, servers can return structured application-level error payloads inside otherwise successful `tools/call` results, and these payloads may be visible to the model and influence its next step Anthropic (2025c).

**Evidence.** ~~Public guidance on MCP error handling recommends returning context-rich but sanitized error messages because models may use them to recover, retry, or request human intervention Barthelet (2025).~~

**Evidence.** Practitioner implementation guidance on MCP error handling recommends returning context-rich but sanitized `tools/call` error messages because such messages may be injected back into the model context and used by the model to recover, retry, or request human intervention Barthelet (2025). We treat this source as implementation guidance rather than systematic evidence of error-handling failures in deployed MCP systems.

**Impact.** ~~If client-visible error messages include internal URLs, stack traces, credentials, or other diagnostic details, an ordinary tool failure can become an information-disclosure or workflow-manipulation path that exposes sensitive system information to the model or end user and steers the agent toward unsafe recovery behavior Hora (2025).~~

**Impact.** If client-visible error messages include internal URLs, stack traces, credentials, or other diagnostic details, an ordinary tool failure can expose sensitive system information to the model or end user. Practitioner guidance therefore recommends logging full error context server-side while returning sanitized, user-safe messages that avoid leaking system internals, tokens, stack traces, or authentication details Hora (2025).

### A.8.7 Operational Visibility and Logging Gaps

Effective forensics in MCP deployments depends on consistent, high-granularity logging of tool invocations, context changes, and access events, yet many real-world implementations either log too little or do so

inconsistently. These telemetry gaps make it difficult to reconstruct agent behavior or determine which tools ran and what data was accessed.

**Evidence.** ~~Monitoring guidance for MCP deployments reports that the vast majority of implementations lack sufficient telemetry to audit what users and agents are doing Sakib (2025).~~ Practitioner observability guidance for MCP deployments argues that operators need method-level logging, response-size monitoring, user/session attribution, and alerting over JSON-RPC tool calls to audit user and agent behavior Sakib (2025). Enterprise security work likewise identifies *Insufficient Auditability* as a distinct MCP risk, defined by inadequate logging that constrains detection and investigation of security events Narajala & Habler (2025).

**Impact.** Logging gaps leave operators unable to reliably reconstruct agent behavior or verify what tools ran and what data was accessed across workflows. In the absence of adequate telemetry, incident response can degrade into post hoc guesswork about what the agent actually did during the affected period.

### A.8.8 Incomplete Tool De-Registration

Outdated MCP tools or endpoints that are not properly removed during synchronization can continue to appear in indexes, configuration stores, or orchestration layers after operators believe they have been retired. In large deployments where tools are frequently added, updated, and deleted, the gap between the server's current tool list and downstream cached or indexed state can grow over time if removal is not handled correctly.

**Evidence.** Research on MCP-aware tool synchronization treats the server tool list as the single source of truth and includes explicit create, update, and delete operations to keep downstream tool stores aligned Lumer et al. (2025). Production systems likewise expose dedicated removal operations for MCP tools, including OpenSearch's Remove MCP Tools API and TrustGraph's `tg-delete-mcp-tool` command, indicating that stale tool records must be actively deleted rather than assumed to disappear automatically OpenSearch Project (2025); TrustGraph (2025).

**Impact.** If removal or synchronization steps are skipped or fail, stale tool records can linger in indexes and configuration groups, causing agents or orchestration layers to continue discovering or invoking tools that operators intended to retire. This creates a persistent mismatch between operational intent and effective capability exposure, increasing the risk of misrouting, outdated behavior, and avoidable attack surface.

### A.9 Emergent Threats from Model Behavior and Human Factors

### A.9.1 Model Inversion Attacks.

When multiple MCP agents rely on a shared model backend, an attacker may use one agent interface to issue carefully crafted queries that infer sensitive properties of data the model has learned during training or adaptation. In enterprise MCP deployments, this creates a shared privacy boundary in which the confidentiality of information exposed through one workflow depends not only on that workflow's own controls, but also on whether other agent interfaces can be used to probe the same underlying model.

**Evidence.** Prior work on model inversion shows that black-box access to model outputs can, in some settings, reveal sensitive attributes or approximate information about the data used to train the model Fredrikson et al. (2015).

**Impact.** In MCP environments built around shared model backends, a malicious workflow could use ordinary inference access to extract information correlated with proprietary or tenant-sensitive data, even without compromising MCP servers, tool implementations, or the surrounding infrastructure.

### A.9.2 Membership Inference.

In deployments where MCP agents expose a shared enterprise model trained or adapted on internal data, an attacker can probe one agent interface to test whether particular records, users, or documents were included in the model's training corpus. Because the attack requires only query access to normal model outputs,

it may be available to any user or external actor who can interact with the deployment through ordinary channels.

**Evidence.** Prior work on membership inference shows that black-box access to model outputs can, in some settings, distinguish training examples from non-members with significant reliability Shokri et al. (2017).

**Impact.** In MCP deployments built around shared enterprise models, attackers may be able to infer whether sensitive internal records contributed to training without compromising MCP servers or tools, enabling targeted follow-on attacks against individuals, datasets, or business processes linked to the confirmed training data.

### A.9.3 Pre-MCP Data Poisoning and Backdoored Models.

MCP implementations typically treat the underlying model as a trusted component, so covert behaviors introduced during pre-training, fine-tuning, or instruction tuning may be inherited by every agent that relies on it. Because such behaviors are embedded in the model itself rather than in MCP infrastructure, they can persist across server restarts, configuration changes, and software updates until the affected model is replaced, retrained, or otherwise remediated.

**Evidence.** Prior work shows that poisoned training or instruction-tuning pipelines can implant behaviors that remain dormant under normal use but activate in response to specific prompts, triggers, or contextual patterns Xu et al. (2023). Broader safety analyses also note that data poisoning and hidden model-level behaviors can be difficult to detect or rule out once incorporated into deployed systems Anthropic (2024b).

**Impact.** In MCP deployments built around a shared compromised model, ordinary conversations or tool-mediated contexts may activate hidden behaviors that leak information, bypass intended safeguards, or systematically bias downstream tool use across multiple agents and workflows. The attack surface therefore extends beyond any single MCP component to the full model training, adaptation, and deployment pipeline.

### A.9.4 Malicious Agent Operators.

Because MCP hosts translate model outputs into tool invocations and enforce the surrounding configuration and permission boundaries, operators who connect unvetted servers or grant overly broad privileges can expand an agent's effective execution authority beyond intended policy. In such cases, abuse may not require exploitation of any protocol flaw at all. Instead, it can arise from trusted administrative control over host configuration, server selection, and authorization settings.

**Evidence.** Recent analyses of MCP ecosystems describe hosts as components that translate LLM-generated outputs into external tool invocations and note weak or missing output-verification and trust-validation mechanisms in current deployments Li & Gao (2025); Guo et al. (2025b). These trust assumptions mean that high-impact behavior can result not only from malicious servers or prompt manipulation, but also from unsafe operator choices about what servers to connect and what permissions to grant.

**Impact.** A malicious or careless operator can cause routine-looking workflows to execute high-privilege actions, expose sensitive data, or alter downstream behavior across multiple agents. Because the operator acts through legitimate configuration channels, the resulting abuse may be difficult to distinguish from authorized administrative activity, and downstream servers or connected services may observe only apparently valid requests.

### A.9.5 Insider Threats and Shadow MCP Servers.

Ungoverned or unauthorized MCP servers may be introduced inside organizations without central oversight, creating shadow MCP instances that access internal resources while bypassing standard monitoring, authentication, and governance controls. Because MCP servers can often be deployed and connected quickly, the organizational attack surface may expand faster than security and compliance teams can inventory, review, or contain it.

**Evidence.** OWASP guidance defines shadow MCP servers as unapproved or unsupervised MCP deployments operating outside formal organizational governance and warns that they can bypass centralized authentication, monitoring, and data-governance controls while creating new unmonitored endpoints OWASP Foundation (2025b). Broader ecosystem analyses further show that MCP deployments rely heavily on host-server trust relationships and that weak vetting of server onboarding can expose users and organizations to malicious or unsafe server behavior Li & Gao (2025).

**Impact.** Each shadow server introduces an unreviewed trust boundary that may expose sensitive data, execute internal capabilities, or provide attackers with an unmonitored foothold into enterprise systems. Because these deployments sit outside normal governance and asset-tracking processes, their misuse or compromise may remain undiscovered until after substantial operational, security, or compliance harm has occurred.

### A.9.6 Agent Interface Social Engineering.

The agent interface and host create an attack surface where adversaries can issue crafted prompts or interaction sequences that steer tool selection and execution. Because these attacks operate through seemingly legitimate interactions rather than low-level protocol exploits, they can remain effective even when core MCP mechanisms are implemented as intended.

**Evidence.** Prior work on MCP security treats the agent interface and host as a primary attack surface where adversaries can present seemingly legitimate requests or workflows that drive high-impact tool invocations and data exfiltration under the guise of normal usage Guo et al. (2025b).

**Impact.** Attackers can manipulate agent behavior through ordinary interaction channels, causing unsafe tool use or disclosure without exploiting a protocol flaw in MCP itself. These risks therefore cannot be addressed through protocol hardening alone and also require stronger validation, approval design, and interaction-level safeguards.

### A.9.7 Agent Collusion.

Two or more compromised agents may coordinate to bypass access controls or exfiltrate data covertly, including through encoded or steganographic content in inter-agent messages. In multi-agent MCP deployments, such coordination can combine capabilities that no single agent would be authorized to exercise alone, while remaining difficult to detect when oversight is fragmented across individual agents or workflows.

**Evidence.** Prior work on secret collusion among generative AI agents formalizes the risk of covert coordination through steganographic communication and presents empirical results suggesting that such behaviors warrant continued monitoring as model capabilities improve Motwani et al. (2024). MCP further expands the surface for nested and server-initiated agentic interactions through Sampling, which allows servers to request LLM generations inside other MCP features when implementations expose that capability Model Context Protocol (2025n).

**Impact.** Collusion-based attacks can be difficult to identify because each agent's local behavior may appear policy-compliant when inspected in isolation, with the actual violation emerging only when cross-agent information flow, coordination patterns, or hidden communication channels are reconstructed end to end.

### A.9.8 Unverified Agent Handoffs.

When one agent or MCP server delegates work to another without verifying identity, audience, or effective scope, the handoff can create privilege-escalation or confused-deputy failures. In complex multi-agent topologies where delegation occurs frequently and automatically, each unverified handoff becomes a potential escalation path that may be difficult to enumerate, monitor, or constrain exhaustively.

**Evidence.** The MCP *Authorization* specification explicitly requires audience validation and forbids token passthrough to prevent confused-deputy failures in delegated interactions Model Context Protocol (2025l). Broader multi-agent engineering experience likewise shows that systems with multiple cooperating agents

introduce additional coordination and handoff complexity, increasing the risk that trust boundaries will be enforced inconsistently in practice Anthropic (2025b).

**Impact.** An unverified handoff can allow a lower-privileged agent to trigger actions through a more privileged agent or server that fails to validate the caller's authority. The resulting escalation mirrors the confused-deputy pattern described in A.6.4, but here the failure arises across agent or delegation boundaries rather than within a single server interaction.

### A.9.9 Orchestration Drift.

Over time, loosely coupled MCP workflows may diverge from their intended behavior, accumulate shadow logic, or reintroduce deprecated behaviors as tools, prompts, and coordination patterns evolve. Such orchestration drift can be difficult to detect because the affected workflow may still produce plausible outputs under normal conditions, with the divergence becoming visible only when an edge case, deployment change, or adversarial input exposes the gap between the intended and actual execution logic.

**Evidence.** MCP provides observability and change-management primitives such as tool list change notifications and structured logging, which help implementations track evolving capabilities and runtime behavior Model Context Protocol (2025q). Field reports and engineering experience further show that MCP and multi-agent deployments can accumulate hard-to-trace workflow logic, exhibit emergent behavior under small prompt or orchestration changes, and require careful coordination during deployment to avoid breaking active agents or silently altering behavior Anthropic (2025b).

**Impact.** Drifted workflows may silently grant capabilities beyond what operators intended, re-enable previously closed attack surfaces, or degrade policy enforcement in ways that remain invisible during routine use. Because the divergence often accumulates gradually across prompts, tools, and orchestration logic, point-in-time reviews of configuration alone may fail to capture the actual behavior of the live system.

## B   Qualitative Comparison with Established Threat-Modeling Frameworks

We do not treat STRIDE, MAESTRO, or MITRE ATLAS as direct empirical baselines for the compact benchmark in Section 5. In their standard form, these frameworks organize threats along complementary dimensions rather than directly producing the MCP-specific tuple evaluated in our benchmark, which consists of component, lifecycle phase, expected check, and audit artifact. We therefore compare them qualitatively as reference frameworks in Table 16. STRIDE captures classical security properties, MAESTRO captures agentic-AI system layers and AI-specific threat context, and MITRE ATLAS captures adversarial tactics and techniques for AI-enabled systems. The proposed dual-axis view has a narrower operational focus, centering on MCP-specific component/lifecycle localization, executable checks, and operator-facing evidence.

## C   Evidence-Strength Justification

Table 17 defines the three evidence-strength labels (E1–E3) used to annotate each threat in Tables 4–6. Table 18 records the rationale for every evidence-strength label (E1, E2, and E3) assigned in those tables.

Table 18: Rationale for every evidence-strength label (E1 established, E2 supported, E3 emerging), grouped by tier and annotated with the taxonomy category.

| Category | Threat | Tier | Basis |
|---|---|---|---|
| *— E1: Established —* | | | |
| Identity & Trust | Implicit Trust in Unauthenticated Servers | E1 | MCP-specific CVE (CVE-2025-49596, MCP Inspector) plus large-scale MCP handler analysis National Vulnerability Database (2025); Sun et al. (2026). |

| Category | Threat | Tier | Basis |
|---|---|---|---|
| Authorization & Access Control | Confused Deputy / Role Leakage | E1 | MCP authorization specification explicitly warns about token passthrough / audience-validation abuse, plus empirical MCP audit MCP Working Group (2025b); Radosevich & Halloran (2025). |
| Execution Env. & Sandbox | Lack of Execution Isolation | E1 | MCP transport specification plus large-scale empirical study of 2,562 MCP applications Model Context Protocol (2025s); Li et al. (2025b). |
| Execution Env. & Sandbox | Sandbox Escape Vulnerabilities | E1 | Confirmed runtime CVE (CVE-2019-5736, runc) plus MCP safety audit demonstrating host reach AWS Compute (2019); Radosevich & Halloran (2025). |
| Execution Env. & Sandbox | Root or Elevated Execution | E1 | MCP-specific CVE (CVE-2025-49596) plus large-scale privilege/over-privilege studies National Vulnerability Database (2025); Li et al. (2025b); Huang et al. (2026a). |
| Workflow & Config. | Configuration Drift Across Servers | E1 | Quantified MCP measurements: 13,000+ public servers and a 1,899-server study (7.2% vulnerable, 5.5% tool-poisoning) Prompt Security Team (2025); Hasan et al. (2025). |

*— E2: Supported —*

| Category | Threat | Tier | Basis |
|---|---|---|---|
| Identity & Trust | Server Name Collisions and Impersonation | E2 | Empirical MCP and adjacent plugin-ecosystem analyses of typosquatting / look-alike endpoints; mechanism shown, exploitation prevalence not quantified Hou et al. (2025); Bhatt et al. (2025). |
| Identity & Trust | Mutual Authentication and Encryption Gaps | E2 | MCP transport and authorization specifications plus protocol-level security analyses; spec-acknowledged architectural gap treated as operational guidance, not prevalence Model Context Protocol (2025s;l); Hou et al. (2026a); Maloyan & Namiot (2026). |
| Identity & Trust | Lack of Verified Server Identity or PKI | E2 | Specification-level absence of native server-identity binding beyond TLS/OAuth; design gap, not an exploited-incident measurement Anthropic (2025); Metere (2026). |
| Supply Chain & Code Integrity | Malicious Dependencies or Backdoored Code | E2 | Peer-reviewed npm-malware study and confirmed real-world package incidents in adjacent ecosystems; strong evidence transferred to MCP by extrapolation Sejfia & Schäfer (2022); Eriksen, Charlie (2025); Meyer (2025). |
| Supply Chain & Code Integrity | Lack of Integrity Verification | E2 | MCP defines no protocol-wide signing of tool calls/artifacts; spec-grounded gap with software-signing support Model Context Protocol (2025r); Kalu & Davis (2025). |
| Supply Chain & Code Integrity | Unverified Package Sources and Automatic Invocation | E2 | Large-scale empirical MCP study finds a substantial number of listed servers can be hijacked (strongest E1 promotion candidate) Li & Gao (2025); Birur (2025). |

| Category | Threat | Tier | Basis |
|---|---|---|---|
| Malicious / Poisoned Tool Servers | Tool Shadowing and Name Collisions | E2 | MCP benchmark (MCPSecBench) demonstrates redirection to attacker-controlled tools across tested platforms Yu et al. (2025); Yang et al. (2025). |
| Malicious / Poisoned Tool Servers | Rug Pull via Dynamic Tool Redefinition | E2 | Reproducible MCP demonstration of post-approval tool redefinition on public aggregators Song et al. (2025); Bhatt et al. (2025). |
| Malicious / Poisoned Tool Servers | Poisoned Tool Metadata | E2 | Large-scale MCP analysis of description/implementation inconsistency plus implicit-tool-poisoning proof-of-concept Li et al. (2026b); Invariant Labs (2025); Li et al. (2026a). |
| Malicious / Poisoned Tool Servers | Pre-Invocation Code Injection | E2 | MCP taint-style analysis links natural-language inputs to confirmed exploit traces; implicit poisoning acts before invocation Li et al. (2026a); Sun et al. (2026). |
| Malicious / Poisoned Tool Servers | Slash Command Collisions | E2 | MCP benchmark evidence of command shadowing within shared namespaces; single benchmark source Yang et al. (2025). |
| Malicious / Poisoned Tool Servers | ANSI Escape Code Injection | E2 | Confirmed display-spoofing CVEs (CVE-2024-33899/36052) in adjacent tooling; MCP rendering application by analogy nvd (2024a;b). |
| Prompt & Context Injection | Poisoned Prompt Templates | E2 | Empirical prompt-injection on high-privilege agentic editors (84% ASR) plus MCP template analysis Microsoft (2025b); Liu et al. (2025b). |
| Prompt & Context Injection | RADE (Retrieval-based poisoning) | E2 | Concrete MCP demonstration (Chroma + filesystem/Slack tools) plus follow-on guardrail-bypass study Radosevich & Halloran (2025); Halloran et al. (2025). |
| Prompt & Context Injection | Transitive-Trust Poisoning | E2 | Single documented MCP incident (Supabase + Cursor, `service_role` SQL written back into a support ticket) General Analysis (2025). |
| Prompt & Context Injection | Format-Embedded Poisoning | E2 | MCP parasitic-toolchain work plus established peer-reviewed indirect-prompt-injection foundation Zhao et al. (2025a); Greshake et al. (2023). |
| Credential Leakage & Data Exposure | Centralized Credential Storage | E2 | OAuth bearer-token standards establish the concentration risk; MCP amplification via token passthrough reasoned Jones & Hardt (2012); Lodderstedt et al. (2025); MCP Working Group (2025b). |
| Credential Leakage & Data Exposure | Insecure Secret Management | E2 | Recognized secret-exposure weakness classes (CWE-312/532); MCP context extrapolated MITRE (2025a;b); Errico et al. (2025). |
| Credential Leakage & Data Exposure | Overprivileged & Persistent Tokens | E2 | OAuth standards plus documented real-world OAuth-application abuse persisting after credential resets Jones & Hardt (2012); Lodderstedt et al. (2013; 2025); Proofpoint Threat Research (2025). |

| Category | Threat | Tier | Basis |
|---|---|---|---|
| Credential Leakage & Data Exposure | Cross-Tool Data Flows | E2 | Documented indirect-prompt-injection attack exfiltrating conversation history to an attacker-controlled domain Palo Alto Networks Unit 42 (2025). |
| Credential Leakage & Data Exposure | Residual Data Exposure | E2 | Established media-sanitization and memory-dump facts (NIST 800-88); MCP residue extrapolated Kissel et al. (2014); VeraCrypt Project (2015). |
| Credential Leakage & Data Exposure | Trojanized MCP Servers | E2 | MCP proxy/exfiltration proof-of-concept plus cross-server trust-abuse work CyberArk Threat Research (2025); Croce & South (2025); Jones & Hardt (2012). |
| Authorization & Access Control | Binary Permissions | E2 | MCP authorization design characteristic documented in enterprise and practitioner authz guidance Narajala & Habler (2025); Nikhil (2025). |
| Authorization & Access Control | Lack of Risk-Aware Differentiation | E2 | MCP specification states tool annotations are advisory only and must not drive security decisions Anthropic (2025c); Narajala & Habler (2025). |
| Authorization & Access Control | No Context-Aware Revalidation | E2 | MCP spec token-reuse behavior plus secure-delegation research on bearer/JWT validity-until-expiry MCP Working Group (2025a); Goswami (2025). |
| Authorization & Access Control | Uneven RBAC Support | E2 | MCP defines no shared RBAC model; spec plus practitioner authorization analyses Model Context Protocol (2025n). |
| Authorization & Access Control | Orphaned Sessions and Credentials | E2 | OAuth token-revocation standard provides the mechanism; MCP teardown failure extrapolated Lodderstedt et al. (2013). |
| Authorization & Access Control | Decentralized & Inconsistent Policy Enforcement | E2 | MCP governance analysis identifies decentralized, inconsistent enforcement; single analytical source Errico et al. (2025). |
| Execution Env. & Sandbox | Prompt-Based Remote Code Execution | E2 | Reproducible MCP proof-of-concept (injected prompt installs a reverse shell via shell config) Radosevich & Halloran (2025). |
| Execution Env. & Sandbox | DoS and Resource Exhaustion | E2 | Research on MCP-compatible tool servers inducing stealthy multi-turn resource amplification Zhou et al. (2026b). |
| Execution Env. & Sandbox | Vulnerable Third-Party Dependencies | E2 | Confirmed dependency CVE (CVE-2022-1471, SnakeYAML RCE); MCP triggering via parse paths extrapolated GitHub Advisory Database (2022); National Vulnerability Database (2022a); Model Context Protocol Security Working Group (2025). |
| Execution Env. & Sandbox | Lateral Movement via Shared Infrastructure | E2 | Established lateral-movement mechanics (MITRE Valid Accounts, NIST 800-207) plus MCP cross-component analyses Maloyan & Namiot (2026); Li & Gao (2025); MITRE ATT&CK (2025); National Institute of Standards and Technology (2020). |

| Category | Threat | Tier | Basis |
|---|---|---|---|
| Workflow & Config. | Context Drift and State Inconsistency | E2 | Agent state-propagation failure research plus formal context-drift studies Chang & Geng (2025); Wu et al. (2025); Dongre et al. (2025). |
| Workflow & Config. | Transactional Integrity Gaps | E2 | MCP transport/JSON-RPC specs lack an atomic multi-call primitive; trustworthy-agent research motivates rollback Model Context Protocol (2025s); Morley (2010); Li et al. (2025a); Chang & Geng (2025). |
| Workflow & Config. | Patch Gaps and Vulnerable Instances | E2 | Documented MCP incident (SQLite MCP SQL injection, 5,000+ forks) reported by two independent sources Park (2025); Lyons (2025). |
| Workflow & Config. | Session Isolation in Multi-Agent Environments | E2 | MCP security best-practice guidance on session hijacking and leaky session stores Model Context Protocol (2025j). |
| Emergent Threats | Model Inversion Attacks | E2 | Peer-reviewed black-box model-inversion attack; MCP shared-backend applicability extrapolated Fredrikson et al. (2015). |
| Emergent Threats | Membership Inference | E2 | Peer-reviewed black-box membership-inference attack; MCP shared-model applicability extrapolated Shokri et al. (2017). |
| Emergent Threats | Pre-MCP Data Poisoning | E2 | Peer-reviewed data-poisoning / instruction-backdoor research; MCP inheritance of model behavior reasoned Anthropic (2024b); Xu et al. (2023). |
| Emergent Threats | Unverified Agent Handoffs | E2 | MCP authorization spec requires token-audience binding and forbids passthrough; vendor multi-agent guidance Model Context Protocol (2025l); Anthropic (2025b). |
| Emergent Threats | Malicious Agent Operators | E2 | MCP ecosystem analyses showing hosts act on tool metadata / model output without strong secondary validation Li & Gao (2025); Guo et al. (2025b). |
| Emergent Threats | Insider Threats and Shadow MCP Servers | E2 | OWASP MCP Top-10 plus MCP ecosystem study on ungoverned / shadow servers OWASP Foundation (2025b); Li & Gao (2025). |
| *— E3: Emerging —* | | | |
| Supply Chain & Code Integrity | Trojanized or Compromised Installers | E3 | Supported by installer documentation and conceptual reasoning; no confirmed MCP installer-compromise incident yet Smithery (2025); mcp (2025a;b). |
| Supply Chain & Code Integrity | Vulnerable Redeployments and Stale Manifests | E3 | Survey-level and analogy-based evidence (agent state-propagation failures); MCP-specific demonstration limited Ehtesham et al. (2025); Chang & Geng (2025). |
| Malicious / Poisoned Tool Servers | Preference Manipulation | E3 | Emerging; one preprint plus practitioner TTP write-ups, distinct from explicitly-validated poisoned-metadata attacks Hasan et al. (2026); Model Context Protocol Security (2025); Elastic Security Labs (2025). |

| Category | Threat | Tier | Basis |
|----------|--------|------|-------|
| Credential Leakage & Data Exposure | Emergent Aggregation Risks | E3 | Mosaic-effect / aggregation reasoning is conceptual and drawn from adjacent privacy literature, not MCP-specific measurement Gurung et al. (2026); Solove (2004). |
| Authorization & Access Control | Consent Fatigue and Automation Drift | E3 | Human-factors mechanism plus a single MCP abuse PoC; prevalence not independently quantified CyberArk Threat Research (2025). |
| Workflow & Config. | Unsafe Fallbacks and Error Handling | E3 | Practitioner implementation/error-handling guidance; no MCP-specific empirical study Barthelet (2025); Hora (2025). |
| Workflow & Config. | Operational Visibility and Logging Gaps | E3 | Absence-of-control argument grounded in practitioner observability guidance and enterprise framing Sakib (2025); Narajala & Habler (2025); Ithena (2025). |
| Workflow & Config. | Incomplete Tool De-Registration | E3 | Conceptual plus tooling/API documentation; no documented exploited MCP incident Lumer et al. (2025); OpenSearch Project (2025); Akın (2025); TrustGraph (2025). |
| Emergent Threats | Agent Collusion | E3 | Covert-channel/collusion mechanism from adjacent multi-agent research; MCP applicability reasoned via the Sampling surface Motwani et al. (2024); Model Context Protocol (2025n). |
| Emergent Threats | Orchestration Drift | E3 | Spec change-notifications plus vendor field reports; emergent and not independently quantified Model Context Protocol (2025q); Anthropic (2025b). |
| Emergent Threats | Agent Interface Social Engineering | E3 | Conceptual/human-factors framing from MCP security analysis; no controlled MCP study Guo et al. (2025b). |

## D   Machine-Checkable Controls

The control families above state *what* to enforce. Here we give five small, fail-closed artifacts that make a representative subset directly machine-checkable in a registry admission step, an authorization service, a logging pipeline, or a CI/CD gate. Table 19 summarizes them, and the listings that follow give a minimal example of each. Together they are the concrete encodings of the checks evaluated in Section 5: manifest admission and the CI/CD gate block poisoned or unapproved tools and post-approval drift (benchmark scenarios S3, S4, and S7); scope-linting catches over-broad authority (S1, S12); audience validation enforces per-request authorization (S5, S6); and the log signals confirm that every denial leaves a trace (S8, S10). These are illustrative reference implementations meant to be adapted to a deployment's own registry, identity provider, and telemetry stack.

*Artifact 1* admits a tool only if its manifest is signed and the built binary's hash matches, blocking unapproved tools and silent post-approval changes Bhatt et al. (2025); Sigstore (2022).

Listing 1: Signed manifest and fail-closed admission check.

```
# read_file.manifest.json
{ "name":"read_file", "version":"1.4.2",
  "sig":"read_file.sig",
  "sha256":"9f86d081...",
  "scopes":["fs:read:/srv/data"] }
# admission gate (non-zero exit => tool rejected)
```

Table 16: Qualitative comparison between the proposed dual-axis view, reduced ablations, and established threat-modeling frameworks.

| Framework / View | Captures | Not designed to capture directly |
|---|---|---|
| Lifecycle-only ablation | Lifecycle phase of the visible failure signal | Responsible MCP component and component-scoped evidence |
| Component-only ablation | MCP component associated with the visible failure signal | Lifecycle phase where the relevant control should fire |
| STRIDE | Classical security properties such as spoofing, tampering, repudiation, information disclosure, denial of service, and elevation of privilege | MCP-specific lifecycle phase, component placement, executable check, and audit artifact |
| MAESTRO | Agentic-AI system layers and AI-specific threat context | Scenario-level MCP control placement and runtime evidence generated by the benchmark |
| MITRE ATLAS | Adversarial tactics, techniques, and procedures for AI-enabled systems | MCP-specific component/lifecycle localization, expected enforcement check, and operator-facing audit evidence |
| Proposed dual-axis view | MCP component, lifecycle phase, selected executable check, and reviewable evidence artifact | Broad adversary campaign modeling or complete empirical validation of all taxonomy categories |

Table 17: Evidence-strength scale for threat annotations.

| Label | Name | Meaning |
|---|---|---|
| E1 | Established | Supported by formal specifications, CVEs, peer-reviewed empirical work, or independently reproduced evidence. |
| E2 | Supported | Supported by MCP-specific preprints, benchmarks, proof-of-concept demonstrations, vendor advisories, or established security mechanisms extrapolated to MCP. |
| E3 | Emerging | Supported mainly by single practitioner reports, early operational observations, or plausible MCP-specific risks with limited independent validation. |

```
cosign verify-blob \
  -key tools.pub \
  -signature read_file.sig \
  read_file.manifest.json
test "$(sha256sum tool.bin | cut -d' ' -f1)" \
  = "$(jq -r .sha256 read_file.manifest.json)"
```

*Artifact 2* lints the authority a tool requests at registration, denying wildcard scopes or a write-plus-egress combination that enables exfiltration Open Policy Agent Contributors (2025); Lodderstedt et al. (2025).

Listing 2: Scope-linting rules (Rego); any **deny** blocks registration.

```
package mcp.scopelint
import rego.v1
deny contains m if {
  contains(input.scopes[_], "*")
  m := "wildcard scope"
}
```

Table 19: The five machine-checkable artifacts, what each blocks, and the control family it implements.

| Artifact | What it blocks | Control family |
|---|---|---|
| Signed-manifest admission (Listing 1) | Unsigned, unapproved, or silently modified tools | Crypto. Signing & Registry |
| Scope-linting (Listing 2) | Wildcard scopes; write combined with network egress | Fine-Grained Auth & Policy |
| Token-audience validation (Listing 3) | Token passthrough and scope escalation (confused deputy) | Zero-Trust Architecture |
| Runtime log signal (Listing 4) | Undetected exfiltration; denials with no audit trace | Monitoring & Logging |
| CI/CD failure conditions (Listing 5) | Unsafe tool/server versions reaching deployment | Crypto. Signing; Containerization |

```
deny contains m if {
  startswith(input.scopes[i], "fs:write")
  count(input.egress_allow) > 0
  m := "write + egress (exfil risk)"
}
```

*Artifact 3* authorizes each call only if the token audience equals this server and its scopes cover the request, defending against confused-deputy passthrough MCP Working Group (2025b); Model Context Protocol (2025l).

Listing 3: Per-request token-audience validation; failed assertion returns HTTP 403.

```
claims = verify_jwt(token, IDP_JWKS)
assert claims.aud == SERVER_ID
assert set(req.scopes) < set(claims.scopes)
assert set(req.scopes) < tool_policy.allowed
```

*Artifact 4* emits one structured record per call and flags egress to a destination not seen recently; a denial with no matching record is itself a reviewable observability gap Sakib (2025); Errico et al. (2025).

Listing 4: Audit record (per call) and first-seen-egress detection query.

```
{ "ts":        "2026-06-07T12:00:00Z",
  "principal": "agent:planner",
  "tool":      "read_file",
  "decision":  "deny",
  "reason":    "scope_violation",
  "egress_dst": null }

SELECT  principal, tool, egress_dst
FROM    mcp_audit
WHERE   decision = 'allow'
  AND   egress_dst IS NOT NULL
  AND   egress_dst NOT IN (
    SELECT egress_dst FROM mcp_audit
    WHERE ts < now() - interval '30 days'
      AND egress_dst IS NOT NULL
  );
```

*Artifact 5* fails the pipeline on an unsigned artifact, any scope-lint denial, or a critical dependency vulnerability, blocking unsafe versions before deployment SLSA Community (2025); Souppaya et al. (2022b).

Listing 5: CI/CD gate; each step exits non-zero on violation and blocks the deploy.

```
# .ci/mcp_gate.yml
- run: cosign verify-blob
    - key tools.pub
```

```
      - signature manifest.sig
    manifest.json
- run: opa eval
      - input manifest.json
      - data scopelint.rego
    'data.mcp.scopelint.deny'
      - fail-defined
- run: osv-scanner
      - lockfile=package-lock.json
    # non-zero on any CRITICAL CVE
```

