# OpenReview forum: "Securing The Model Context Protocol (MCP): A Dual-Axis Survey with a Mitigation-Oriented Threat Taxonomy"
_TMLR — Decision pending for TMLR_

### Review · Reviewer_cCVD · 2026-05-16

**Summary Of Contributions:**

This paper surveys security risks in MCP-based agentic systems and proposes a dual-axis taxonomy that maps each threat by both component and lifecycle phase. I see the main contribution as turning a scattered set of MCP risks into a more operational framework: threats are linked to STRIDE/MAESTRO labels, concrete mitigations, verification checks, and runtime signals. The paper is broad and timely, and the taxonomy is useful for practitioners. Its main weakness is that the empirical evaluation is still quite small, so the claims about practical effectiveness should be framed more carefully.

**Audience:**

Yes

**Audience Explanation:**

Yes. MCP is becoming increasingly relevant for tool-using and agentic AI systems, and security is a major concern for these deployments. The paper will likely interest researchers working on LLM agents, tool use, AI safety, security, and trustworthy deployment. The practical mapping from threats to controls and evidence artifacts also makes the work useful beyond a purely academic audience.

**Broader Impact Concerns:**

I do not see major unaddressed broader impact concerns. The paper is mainly defensive and aims to improve security for MCP-based AI systems. The main concern is dual use: the taxonomy and attack descriptions could help attackers understand MCP weaknesses. This is mostly mitigated by the paper’s focus on controls, verification, and monitoring, but I would still recommend adding a brief broader impact statement that explicitly discusses dual-use risk and responsible disclosure.

**Claims And Evidence:**

Yes

**Claims Explanation:**

Overall, yes. The paper supports its taxonomy with a large review of prior work, incident reports, specifications, and practitioner sources, and it explains its coding process reasonably clearly. The small MCP benchmark is also helpful because it shows how the proposed dual-axis view can improve localization compared with single-axis baselines. That said, the benchmark is intentionally compact, so it supports feasibility more than broad generality. I would like the authors to be more explicit about this limitation when interpreting the results.

**Requested Changes:**

Critical: Clarify the scope of the benchmark and avoid overclaiming from it. The evaluation is useful, but it is small and should be presented mainly as a feasibility/validation study rather than strong empirical proof.

Critical: Make the taxonomy construction easier to audit. The paper describes the review protocol, but it would help to include clearer criteria for why particular threats were merged, split, or assigned to specific lifecycle/component cells.

Would strengthen the work: Reduce some repetition in the threat descriptions. Several sections repeat similar points about trust, authentication, and telemetry, and tightening this would improve readability.

Would strengthen the work: Add a concise comparison table against the closest existing MCP security surveys/frameworks, highlighting exactly what is new in this paper.

Would strengthen the work: Discuss how the proposed controls would scale in larger, multi-agent and multi-tenant MCP deployments, since the current testbed is much simpler than many real deployments.

---

> ### Author Response · Authors · 2026-06-14
>
> We appreciate the reviewer’s careful reading and the clear summary of where the paper is useful and where it needs tighter framing. In the revision, we addressed these points directly; the bullets below summarize the main changes and how each one improves the paper.
>
> **1. Benchmark scope:** We thank the reviewer for this point and agree. We have recalibrated the benchmark claims throughout the paper and now present it consistently as a compact proof-of-concept and feasibility study rather than strong empirical proof. We softened the wording wherever the benchmark is referenced and explicitly state that it does not validate the completeness of the taxonomy, the effectiveness of all mitigation categories, or general real-world detection performance. We also added a dedicated Limitations of the Benchmark subsection (§5.1), and we changed the reported result from a general accuracy claim to scenario-level localization agreement on curated scenarios, noting that high agreement should be read as an implementation sanity check rather than evidence of general performance. We also added a coverage table (Table 13) that makes the scope of each scenario explicit by mapping it to its taxonomy category, expected check, observed outcome, and the limitation of that curated case.
>
> **2. Make the taxonomy construction easier to audit:** We thank the reviewer for this observation. The revised §3 adds explicit construction criteria for exactly these decisions. The Deduplication procedure paragraph states the merge rule (mechanism identity, not surface) with two worked examples, and the split rule (separate entry only when a distinct mechanism needs different controls). The Coding paragraph and the Ambiguity resolution block state how cells are assigned, one primary category by dominant mechanism via the Table 2 boundary rules, with component and lifecycle tags for secondary effects. Furthermore, we have also included five representative coding decisions to illustrate the rules on near boundary threats.
>
>
> **3. Reduce some repetition in the threat descriptions:**  We thank the reviewer for this feedback. We have tightened the threat descriptions in §3 to reduce redundancy, focusing especially on the identity, credential leakage, workflow, and execution subsections. Across these subsections, we removed repeated material including overlapping explanations of bearer-token handling, a reused vulnerability reference with a duplicate bibliography entry, a restated PKI-versus-implicit-trust comparison, a repeated injection explanation, recurring late-detection phrasing in the workflow section, and repeated impact language. The revised §3 now keeps each threat’s structure same across categories while making the descriptions more concise.
>
> **4. Comparison table against the closest existing MCP security surveys/frameworks:** We thank the reviewer for this suggestion, which we have adopted. §1 now includes a comparison table, Table 1, that positions our work against the five closest MCP security surveys and frameworks. The table compares each work across seven capabilities: lifecycle-axis coverage, component-axis coverage, joint dual-axis mapping, STRIDE/MAESTRO alignment, controls tied to where and when they apply, verifiable checks with evidence, and an empirical study or benchmark. Each cell is marked as yes, partial, or no. This makes the distinction between our contribution and prior work more explicit.
>
> **5. Scalability of proposed controls:** We thank the reviewer for this suggestion, which we have adopted. We added a subsection titled “Scaling Considerations for Larger MCP Deployments” (§5.2) and expanded the benchmark limitations to note that the testbed does not model large multi host, multi agent, or multi-tenant settings. It explains how the proposed controls would operate as shared platform services at scale, covers multi-agent and multi-tenant isolation and delegation, and names the scaling risks left as future work.

---

### Review · Reviewer_c9Rf · 2026-05-29

**Summary Of Contributions:**

This survey paper first reviews literature relevant to, and proposes a two-axis taxonomy for methods securing the Model Context Protocol (MCP), a protocol that enables agentic AI systems to interface with external tools, servers, and data sources. In particular, these axes involve: (1) lifecycle-based categorization, which identifies when threats emerge, and (2) component-based categorization, which identifies where threats appear. The authors argue that existing efforts toward MCP security focus separately on the two axes without accounting for how threats can cascade across both the axes. The contributions of this paper include the following: (1) an understanding of MCP security concerns should account for threat cascading across lifecycle and components, (2) an enumeration and organization of MCP-specific threats in nine failure categories, such as identity/trust failure, supply chain vulnerability, poisoned tools, prompt injection, credential leakages, access control vulnerabilities, sandboxing failure, workflow drift, and multi-agent risk, (3) mapping of each threat to MCP components, lifecycles, and STRIDE/MAESTRO threat profiles, and (4) pairing each threat with a mitigation strategy and benchmark that demonstrates the feasibility of detecting and auditing such threats.

**Additional Comments:**

Some additional thoughts/recommendations:
- While this is more broadly a survey paper, there's not really a discussion of potential mitigations for some of the threats that are part of the discussed axes. It would be good if it were clearly identified how, for each part of the matrix, implementations could be created which mitigate the threats (both a at a systemic level, and at a local developer level).
- It would also be nice if the paper did a slightly better job of identifying areas for future work. Right now, it's not immediately clear what are the big problems remaining that need to be solve, and what kinds of research are required. A summary table for this would be quite helpful.

**Audience:**

Yes

**Audience Explanation:**

Understanding how non-deterministic generative models interact with deterministic, high-privilege enterprise environments is a true real-world problem, and it's clear to me that the application, and the findings here are particularly timely. Despite some of the issues with the paper, I think that the proposed axes do represent a reasonable categorization, and the taxonomy, though dense, is comprehensive enough to be useful to downstream readers, particularly anyone who is trying to learn about the security issues that could arise from the adoption of MCPs in their workflow. The survey paper does hit a very odd audience, however, since readers who are in-domain will likely prefer to read more targeted analyses of parts of the pipeline, and readers who are out of domain will likely prefer a more concise summary of the potential axes. Thus, while the information is relatively timely, and actually quite interesting, I'm a bit concerned that the potential audience will be fairly limited (a good example of who might be interested are security auditors who need a comprehensive view of the potential pipeline areas they should audit, and need to justify those potential issues to a downstream stakeholder).

**Broader Impact Concerns:**

There are no general broader impact concerns, since this is overall a survey paper, and does not directly address the ethical considerations of MCP research.

**Claims And Evidence:**

No

**Claims Explanation:**

One of the primary concerns that I have with this paper is the quality of the referenced data. A large number of the citations are secondary, or even tertiary sources for the research. For example, [1] is a blog post, and the paper should actually probably cite the relevant research [2], and similarly [3,4,5,6, etc.] are examples of blog posts that provide anecdotal evidence which is taken as fact in the paper. This can make it really challenging to trust the claims in the paper, since it's not clear if (1) these claims are actually validated by real research, and (2) if the overall takeaways are real, or just observed internally and reported anecdotally, which isn't exactly good science. I realize that a lot of security research is anecdotal in nature, but this should be clearly identified in the paper when it cannot be verified.

There are several other weaknesses:
- The paper claims of the experiments in Section 5 `If the taxonomy is useful operationally, then checks derived from it should be able to detect representative MCP failures, place those failures at the correct component and lifecycle phase, and leave behind evidence that an operator could inspect later.`, however it's not clear that this is actually true. The fact that the proposed method gets 100% classification accuracy on this benchmark seems to, instead, be an artifact of the design itself (i.e. the benchmark is designed to admit only an optimal classifier of the form discussed in the paper). I wouldn't expect that Real-world LLM orchestration would even closely resemble this setup, and if it does resemble this setup, I wouldn't expect a simple classifier to achieve such strong alignment.
- The baselines in Sec 5 are also somewhat weak, comparing against both the lifecycle-only and component-only baselines suggests that these are reasonable competitors to the method, however I don't see how any modern security framework would implement either of those (i.e. lifecycle without component awareness). It might be worth considering some baselines which are a bit more applicable (perhaps something like the MITRE ATLAS?).





[1] Ajeet Singh Raina. Mcp horror stories: The security issues threatening ai infrastructure, July 2025. URL https://www.docker.com/blog/mcp-security-issues-threatening-ai-infrastructure/. Blog post summarizing ecosystem-wide MCP security findings.

[2] Hasan, Mohammed Mehedi, et al. "Model context protocol (mcp) at first glance: Studying the security and maintainability of mcp servers." ACM Transactions on Software Engineering and Methodology (2025).

[3] Data Science Dojo Staff. The state of mcp security in 2025: Key risks, attack vectors, and case studies, September 2025. URL https://datasciencedojo.com/blog/mcp-security-risks-and-challenges/. Blog article.

[4] Melvin Lammerts. The ai protocol under siege: Mcp server vulnerabilities expose critical threats, July 2025. URL https://hadrian.io/blog/the-ai-protocol-under-siege-mcp-server-vulnerabilities-exp ose-critical-threats. Hadrian blog post.

[5]  Red Hat Product Security. Model context protocol (mcp): Understanding security risks and controls, 2025. URL https://www.redhat.com/en/blog/model-context-protocol-mcp-understanding-security-r isks-and-controls.

[6] Strobes Security Labs. Model context protocol and its critical vulnerabilities, 2025. URL https://strobe
s.co/blog/mcp-model-context-protocol-and-its-critical-vulnerabilities/.

**Requested Changes:**

My primary, and really only, requested change is that the the citations be thoroughly reviewed for quality - so many of these citations are blog posts, random powerpoints, or similar, and are secondary/tertiary resources for the actual research/findings. Until this happens, I don't believe that the survey paper is technically sound.

---

> ### Author Response · Authors · 2026-06-14
>
> We thank the reviewer for the careful and detailed feedback. We appreciate the reviewer’s positive comments on the timeliness of the problem and the usefulness of the proposed axes, as well as the additional suggestions that helped us improve the paper. We have revised the paper in response to the concerns raised, as summarized below.
>
>
> **1. Citation Quality:** We thank the reviewer for this important concern, and we agree that citation quality is central to a survey's soundness. We audited every citation in the paper, tracked in a public sheet, and primarily made three changes. First, we removed about 81 secondary and tertiary sources such as blogs, vendor posts, etc. and added 20 primary references including peer reviewed and arXiv papers, RFCs, CVEs, CWEs, NIST publications, and the official MCP specification. None of the removed sources remain active citations. Second, this was not just a citation swap. We also reworded about ~70 passages so that no claim now depends on a removed source, anchoring each one in primary sources or standards. Practitioner and single source reports are now used only as operational signals or motivation, never as sole support for prevalence, effectiveness, or generality claims, and they are now explicitly flagged directly in the prose, for example as practitioner guidance or as practitioner evidence rather than primary evidence. Third, we labeled every threat by evidence strength at one of three levels, namely E1 established, E2 supported, and E3 emerging, covering 6, 42, and 11 of the 59 threats respectively. The rubric is defined in Section 3 (Tables 4-6 now also have an additional evidence column), and a justification table for each threat appears in the appendix (Table 18).  These changes ensure the paper's claims rest on verifiable primary evidence, with any remaining operational evidence explicitly labeled and bounded.
>
> Link : https://docs.google.com/spreadsheets/d/1ndPB4WxHv0HFU8UKYZsrHlCFkZYdaZEn/edit?gid=336788616#gid=336788616
>
>
> **2. Calibrating the Section 5 benchmark and baseline claims:** We thank the reviewer for the helpful feedback. We agree that the original Section 5 framing overstated what the compact benchmark can establish. We have revised the paper to present the benchmark as a compact proof-of-concept and feasibility study rather than as broad empirical validation. The revised text now explicitly states that the benchmark does not validate the completeness of the taxonomy, the effectiveness of all mitigation categories, or general real-world detection performance.
>
> We also changed the reported result from a general accuracy claim to scenario-level localization agreement in curated scenarios, and added limitations (§5.1) explaining that high agreement should be interpreted as an implementation sanity check rather than evidence of general classifier performance. To make the benchmark less opaque, we added a coverage table (Table 13) mapping each scenario to its taxonomy category, expected check, observed outcome, and limitation.
>
> Finally, we agree that lifecycle-only and component-only variants should not be treated as realistic security-framework baselines. We now describe them as reduced ablations used only to isolate the contribution of the two-axis structure throughout the paper. We also added a qualitative comparison with STRIDE, MAESTRO, and MITRE ATLAS, positioning them as complementary reference frameworks rather than direct empirical baselines in appendix (B) .
>
>
> **3. Expanding the mitigation discussion across the taxonomy matrix:**  We thank the reviewer for this suggestion. We added Section~4.7, “Mitigation Responsibilities Across the Matrix,” to make the mitigation path more explicit. The new table maps each of the nine taxonomy categories, with its component and lifecycle position, to both a systemic mitigation and a local-developer mitigation. The systemic level covers operator/platform controls such as gateways, trusted registries, sandboxes, and policy engines, while the local level covers per-server or per-tool controls such as input validation, scoped tokens, secret handling, and structured logging. This revision clarifies that the two levels are complementary: operators can bound blast radius and enforce a uniform baseline across deployments, while developers can address root causes where they have the most context. Each mitigation also points to the relevant control family in Sections 4.1-4.6 or to the machine-checkable artifacts in Section 4.8.
>
> **4. Future-work roadmap for MCP security:** We thank the reviewer for this suggestion. We agree and have strengthened the future-work section. We added a summary table (Table 15) listing seven open problems for MCP security, each with why it remains open, the specific research required to close it, and the taxonomy region most affected. Each row expands into a dedicated subsection (Sections 7.1 to 7.7), and Section 7.6 and 7.7 are new areas of future work we added to the revised version.

---

### Review · Reviewer_nu6p · 2026-06-01

**Summary Of Contributions:**

This paper surveys security risks in the Model Context Protocol (MCP) and proposes a dual-axis taxonomy that maps threats both to MCP components and to lifecycle phases. The central motivation is that prior work often studies either the temporal dimension of MCP deployments or the architectural/component dimension, but not both jointly. The paper argues that a combined spatio-temporal view is necessary for understanding cross-component and cross-phase threat propagation in agentic workflows.
The paper’s main contributions are: (1) a taxonomy of MCP-specific threats organized into nine failure-mode categories and mapped to components, lifecycle stages, STRIDE, and MAESTRO; (2) mitigation guidance that connects threat categories to controls, verification checks, and runtime signals; and (3) a compact benchmark intended to demonstrate how selected controls can be checked in a real MCP deployment and tied to audit artifacts.
Strengths:
•	The paper addresses a timely and practically important topic, since MCP-style tool use is rapidly becoming part of real agentic AI deployments.
•	The dual-axis framing is useful: it makes clear not only what type of threat is present, but also where and when the threat should be detected or mitigated.
•	The paper is broad in coverage and attempts to connect taxonomy, mitigation, verification, and operational evidence rather than stopping at a descriptive survey.
•	The mapping to STRIDE and MAESTRO makes the work easier to relate to existing security and agentic-AI threat-modeling practices.
Weaknesses:
•	The empirical validation is relatively limited compared with the breadth of the taxonomy. The benchmark demonstrates feasibility but does not yet establish broad coverage across realistic MCP deployments.
•	Some parts of the taxonomy rely on a mixed evidence base including practitioner reports and emerging incidents. This is understandable for a fast-moving area, but the paper should more clearly distinguish mature evidence from early operational signals.
•	The paper would benefit from a clearer methodology for source selection, coding decisions, inter-rater agreement or disagreement resolution, and how threats were deduplicated or assigned to categories.
•	The mitigation guidance is useful, but some controls remain high-level. More concrete examples of deployable policies, verification scripts, failure cases, and expected audit artifacts would strengthen the paper.

**Additional Comments:**

NA

**Audience:**

Yes

**Audience Explanation:**

I believe this paper would interest a meaningful subset of the TMLR audience working on agentic AI systems, tool-use agents, LLM safety, applied security, and evaluation of AI systems interacting with external environments. MCP is becoming a practical integration layer for connecting language models to tools, data sources, and services, and security failures in this setting can directly affect deployed AI systems. A structured taxonomy that helps practitioners reason about where and when risks arise is therefore valuable.
The paper is especially relevant for researchers studying secure tool use, prompt injection, agent orchestration, multi-agent workflows, and AI system auditing. It may also be useful to practitioners building MCP servers or deploying MCP-enabled agents, because the mitigation-oriented mapping gives them a way to translate threat categories into controls and checks.
The paper’s practical relevance is high, although its contribution is more survey/taxonomy-oriented than algorithmic. Its value therefore depends on the clarity, rigor, and usability of the taxonomy and mitigation framework.

**Claims And Evidence:**

Yes

**Claims Explanation:**

for a dual-axis view of MCP security by explaining how lifecycle-only and component-only analyses can miss cross-phase threat cascades. The taxonomy is extensive and appears useful for organizing a broad range of MCP-specific security risks. The use of STRIDE and MAESTRO also helps situate the proposed taxonomy within established threat-modeling frameworks.
However, the evidentiary support is uneven. The paper combines peer-reviewed work, preprints, specifications, incident reports, and practitioner sources. This is appropriate given the novelty and fast movement of MCP security, but the paper should more explicitly separate claims supported by systematic empirical evidence from claims based on anecdotal or early operational reports. Some prevalence claims and category boundaries would be more convincing if the authors provided a clearer coding protocol, source inclusion/exclusion criteria, and examples of ambiguous cases.
The benchmark supports the claim that selected mitigation checks can be instantiated and audited, but it is too compact to support stronger claims about broad deployment coverage, completeness, or general effectiveness. The paper should avoid over-interpreting the benchmark and should state more clearly that it is a proof-of-concept evaluation rather than a comprehensive security evaluation of MCP ecosystems.
Overall, the claims are plausible and mostly well motivated, but the paper would be stronger if it tightened the empirical methodology and calibrated the scope of the benchmark claims.

**Requested Changes:**

1.	Clarify the survey methodology. The paper should more precisely describe how sources were collected, screened, and coded. In particular, it should explain the search strategy, inclusion/exclusion criteria, how many sources were considered at each stage, how duplicates were removed, and how ambiguous threats were assigned to categories.
2.	Better justify the taxonomy design. The paper argues for failure-mode-first grouping plus component/lifecycle tags, which is reasonable, but the category boundaries should be made more explicit. Some threats could plausibly fit multiple categories, so the authors should explain how they handled overlap and provide representative examples of difficult coding decisions.
3.	Calibrate claims about the benchmark. The benchmark should be described as a compact proof of concept unless broader empirical coverage is added. The authors should avoid implying that the benchmark validates the whole taxonomy or the effectiveness of all mitigation categories. A table mapping benchmark scenarios to taxonomy categories, components, lifecycle phases, expected checks, and observed outcomes would help.
4.	Strengthen the evidence quality discussion. Since the taxonomy relies on a mixture of archival papers, specifications, incident reports, and practitioner sources, the paper should distinguish between well-established threats and emerging or less independently validated threats. A confidence or evidence-strength column in the taxonomy tables would be very helpful.
5.	Improve actionability of the mitigation guidance. The paper should provide more concrete examples of enforceable policies and machine-checkable artifacts. For example, sample manifest checks, scope-linting rules, token-audience validation checks, runtime log signals, and CI/CD failure conditions would make the mitigation framework more useful.

---

> ### Author Response · Authors · 2026-06-14
>
> We thank the reviewer for the careful review. We appreciate the positive comments on the paper’s timeliness, dual-axis framing, and connection between taxonomy, mitigations, and verification. We also appreciate the constructive suggestions for improving the paper.
>
> **1. Survey Methodology :** We agree that the methodology needs more detail. §3 now includes a structured Review Protocol and Coding Procedure (before §3.1) with a stage-by-stage flow in Table 3. We searched arXiv, ACM, IEEE, Google/Semantic Scholar, the official MCP specs, advisories, and practitioner reports (Nov 2024 - Feb 2026) using MCP-specific queries plus backward citation tracing, giving 427 initial sources reduced to 257 after title/abstract and full-text screening, with explicit inclusion/exclusion criteria. From these we extracted 94 threat observations and deduplicated by mechanism identity (not surface) into 59 canonical entries, merging, for example, registry and plugin name-collision reports into one Tool Shadowing entry. Each entry was coded by component(s), lifecycle phase(s), STRIDE/MAESTRO, mitigation, and evidence label, with one primary failure-mode category assigned by dominant mechanism using the boundary rules in Table 2.
>
> **2. Better justify the taxonomy design :**  In the revised manuscript, we have added an explicit methodology block at the end of the Review Protocol and Coding Procedure (under §3, before §3.1) that addresses this in three parts. First, we provide a table of per-category boundary rules (Table 2), where each category is given a coding rule and a boundary clarification that distinguishes it from adjacent categories in terms of primary attack mechanism and pipeline stage rather than surface manifestation. The most ambiguous distinctions it draws include poisoned tools (§3.3) versus prompt injection (§3.4), supply chain (§3.2) versus malicious tools (§3.3), and credential leakage (§3.5) versus authorization (§3.6). Second, we give five representative coding decisions showing how those rules were applied to Pre-Invocation Code Injection (§3.3), Transitive-Trust Poisoning (§3.4), Unverified Package Sources and Automatic Invocation (§3.2), Lateral Movement via Shared Infrastructure (§3.7), and Overprivileged and Persistent Tokens (§3.5), each naming the competing categories, the deciding criterion, and the final assignment. Third, we provide two worked examples of the deduplication procedure showing how sources describing the same mechanism across different surfaces were merged into single canonical entries, with surface or evidence differences retained through component and lifecycle tags or supporting citations.
>
> **3. Calibrate claims about the benchmark:** We thank the reviewer for this observation. We have calibrated the benchmark claims throughout the paper. Wherever the benchmark is referenced, we softened the wording and now describe it consistently as a compact proof of concept rather than a comprehensive evaluation, and we explicitly state that it does not aim to validate the full taxonomy or assess all mitigation categories. We also added a dedicated Limitations of the Benchmark subsection (§5.1) and revised the surrounding text within the benchmark section to ground the claims more carefully. For the scenario-to-taxonomy mapping the reviewer requested, the existing Table 12 already records each scenario's component, lifecycle phase, and expected evidence, and we additionally added Table 13, which maps each scenario to the taxonomy category it exercises, the check expected to fire, the observed outcome, and the limitation of that curated case. Together, the two tables now cover all five dimensions the reviewer listed, namely taxonomy category, component, lifecycle phase, expected check, and observed outcome.

---

> > ### Author Response · Authors · 2026-06-14
> >
> > **4. Strengthen the evidence quality discussion:** We thank the reviewer for this suggestion, which we have adopted. We now make source-quality reasoning explicit at the level of each individual threat. Specifically,  We added an Evidence column to all three taxonomy tables (Tables 4–6), labeling every threat E1 (established), E2 (supported), or E3 (emerging).  We also define the three-tier rubric in §3, tied to the source categories already used in our Review Protocol. E1 requires direct MCP evidence (MCP-specific CVE/advisory, large-scale MCP measurement, or specification text naming the gap) or a confirmed CVE/peer-reviewed result that transfers to MCP without added assumptions; E2 covers MCP-specific preprints, benchmarks, proof-of-concept demonstrations, single documented MCP incidents, or well-established mechanisms extrapolated to MCP; E3 covers practitioner, conceptual, human-factors, or adjacent-ecosystem evidence with limited independent MCP validation. Of the 59 threats, 6 are E1, 42 are E2, and 11 are E3, so the taxonomy now clearly separates well-established threats (e.g., implicit trust in unauthenticated servers, confused-deputy/role leakage, execution-isolation and sandbox-escape risks, configuration drift, each backed by MCP CVEs, MCP specification text, or large-scale MCP measurements) from emerging ones (e.g., preference manipulation, emergent aggregation, agent collusion, incomplete tool de-registration).
> >
> > We also added an appendix table (Table 18) that justifies every evidence-strength label across all 59 threats against its sources. We believe this directly addresses the request to distinguish well-established from emerging or less independently validated threats.
> >
> >
> > **5. Improve actionability of the mitigation guidance:** We agree and have added a dedicated subsection (§4.8) along with an artifact catalog in Appendix (D) : (1) a signed tool-manifest schema and admission check that rejects unsigned, unapproved, or silently modified tools (S3/S4/S7); (2) scope-linting rules in Rego that deny wildcard scopes and write-plus-egress combinations before registration (S1/S12); (3) a token-audience validation routine that authorizes a call only if the token audience equals the server and its scopes cover the request, forbidding token passthrough per the MCP authorization spec (S5/S6); (4) runtime log signals comprising a canonical per-call audit record and a first-seen-egress detection query, so that a denial with no matching record is itself a reviewable observability gap (S8/S10); and (5) a CI/CD gate that fails the pipeline on unsigned artifacts, scope-lint violations, or critical dependency CVEs. Each artifact is fail-closed and emits a structured evidence record, so an enforced denial is also an auditable event, the property in our observability-gap scenario (S8) tests. Table 11 maps each artifact to the control family it implements, and the threat category it enforces.